# In situ analysis reveals the TRiC duty cycle and PDCD5 as an open-state cofactor

Huaipeng Xing[1,2], Remus R. E. Rosenkranz[1], Piere Rodriguez-Aliaga[3], Ting-Ting Lee[3], Tomáš Majtner[1], Stefanie Böhm[1], Beata Turoňová[1], Judith Frydman[3✉] & Martin Beck[1,4✉]

The ring-shaped chaperonin T-complex protein ring complex (TRiC; also known as chaperonin containing TCP-1, CCT) is an ATP-driven protein-folding machine that is essential for maintenance of cellular homeostasis[1,2]. Its dysfunction is related to cancer and neurodegenerative disease[3,4]. Despite its importance, how TRiC works in the cell remains unclear. Here we structurally analysed the architecture, conformational dynamics and spatial organization of the chaperonin TRiC in human cells using cryo-electron tomography. We resolved distinctive open, closed, substrate-bound and prefoldin-associated states of TRiC, and reconstructed its duty cycle in situ. The substrate-bound open and symmetrically closed TRiC states were equally abundant. Closed TRiC containing substrate forms distinctive clusters, indicative of spatial organization. Translation inhibition did not fundamentally change the distribution of duty cycle intermediates, but reduced substrate binding for all states as well as cluster formation. From our in-cell structures, we identified the programmed cell death protein 5 (PDCD5) as an interactor that specifically binds to almost all open but not closed TRiC, in a position that is compatible with both substrate and prefoldin binding. Our data support a model in which TRiC functions at near full occupancy to fold newly synthesized proteins inside cells. Defining the TRiC cycle and function inside cells lays the foundation to understand its dysfunction during cancer and neurodegeneration.

The chaperonin TRiC has a pivotal role in maintaining proteostasis in eukaryotic cells by folding approximately 10% of the proteome, including the cytoskeletal proteins actin and tubulin[5–8]. TRiC is a 1 MDa complex comprising two identical stacked rings, each with eight distinct, paralogous subunits arranged in a specific order (CCT3–CCT1–CCT4–CCT2–CCT5–CCT7–CCT8–CCT6)[9]. Each subunit is around 60 kDa in size and contains apical, intermediate and ATP-binding equatorial domains[10]. TRiC uses ATP to cycle between two main conformations in vitro: open and closed. In the open TRiC architecture, the eight apical domains exhibit diverse positions, conferring an asymmetric shape encompassing a chamber, where unfolded substrates can bind. In the closed TRiC state, the trigonal-bipyramidal transition state of ATP at hydrolysis causes a built-in lid in the apical domains to close[11], resulting in a pseudo-$D_8$ symmetrical complex that encapsulates folding substrates within the TRiC chamber[11,12]. The allosteric communication between the rings is not understood, in particular whether TRiC undergoes asymmetric closing—as is the case for bacterial chaperonins, in which one ring is in the open and the other in the closed state—or if both rings open and close in a concerted manner. Furthermore, as previous in vitro structural analyses had used ATP–AlFx to drive TRiC closure[13–16], it remains unclear how the open and closed states distribute inside cells during physiological ATP cycling.

In both the open and closed state, substrates can occupy different positions within the TRiC chamber[14–16]. Moreover, several co-chaperones, such as prefoldin (PFD) and phosducin-like proteins (PhLP), that are involved in TRiC-dependent protein folding were identified in previous studies[17,18]. Cryo-electron microscopy (cryo-EM) analysis of purified PFD and TRiC incubated in vitro showed that jellyfish-shaped PFD binds to the apical domain of the open TRiC conformation, and it is thought to deliver unstructured substrates into the TRiC chamber[17,19]. Recent in vitro studies reported binding of recombinant human PhLP2A, a co-chaperone, within the chamber of one ring of closed TRiC. Thereby, PhLP2A and the substrate asymmetrically bind to two opposing rings[16,20,21]. Furthermore, PhLP2A binds to open TRiC in a mutually exclusive manner with PFD[21]. Although in vitro studies have inarguably advanced our molecular understanding of TRiC[13–16], investigations defining TRiC composition, dynamics and its duty cycle within its native unperturbed environment are to date lacking.

## Open and closed TRiC are evenly abundant

To examine the chaperonin TRiC in its cellular environment, we analysed around 700 previously acquired tilt series from human embryonic kidney 293 (HEK293) cells[22] (Extended Data Fig. 1a). We applied template matching[23] to detect potential TRiC particles in reconstructed

[1]Department of Molecular Sociology, Max Planck Institute of Biophysics, Frankfurt, Germany. [2]Faculty of Biochemistry, Chemistry and Pharmacy, Goethe University Frankfurt am Main, Frankfurt, Germany. [3]Department of Biology and Genetics, Stanford University, Stanford, CA, USA. [4]Institute of Biochemistry, Goethe University Frankfurt, Frankfurt, Germany. ✉e-mail: jfrydman@stanford.edu; martin.beck@biophys.mpg.de

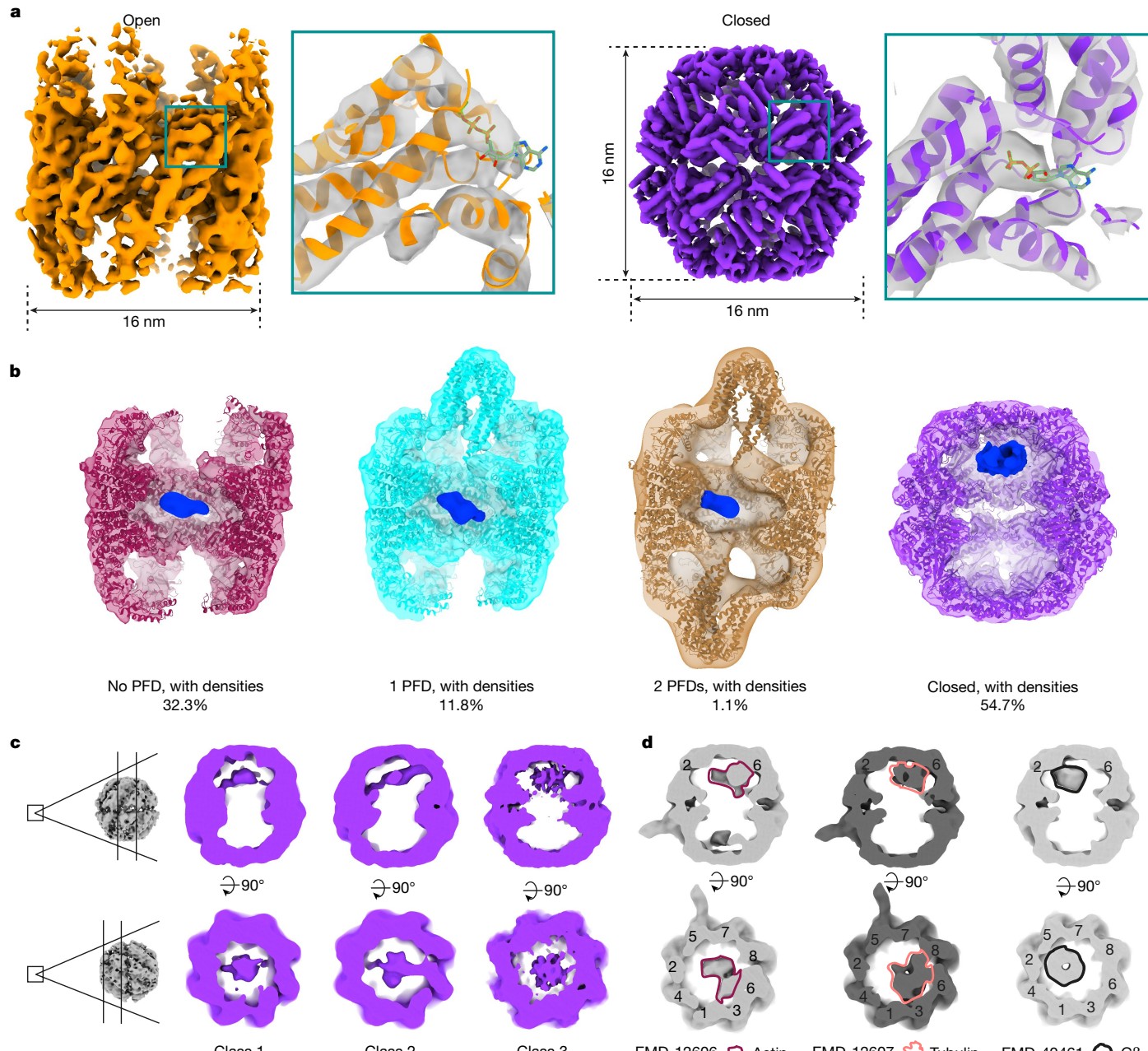

**Fig. 1 | TRiC structures inside human cells. a**, Cryo-EM maps of open and closed TRiC within cells. The square insets show the maps fitted with TRiC atomic models from PDB 7X3J (open) and 7NVN (closed). The resolution does not allow assignment of ATP or ADP to the cryo-ET map, but the secondary structure is well resolved. **b**, Classes of TRiC states obtained by subtomogram averaging within cells (densities in the substrate position are shown in blue). Both open and closed TRiC are substrate bound and abundant. **c**, Classification of closed TRiC according to substrate position. Individual CCT subunits cannot be assigned to our maps of closed TRiC at the given resolution. The three major classes are displayed in the same views as previously analysed in vitro structures, with defined substrates shown in **d**. **d**, The structures of closed TRiC associated with actin, tubulin and Gβ$_5$ (an isoform of G protein β). The substrate densities in **b** and **d** were Gaussian filtered (sDev = 4) for visualization.

tomograms (Extended Data Fig. 1b and Supplementary Video 1). In subsequent subtomogram classification and refinement (Extended Data Fig. 2, Supplementary Figs. 1 and 2 and Supplementary Table 1), we were able to resolve TRiC structures in both the open and closed conformations, achieving resolutions of up to 10 Å (Fig 1a, Extended Data Fig. 1c–g and Supplementary Video 2). The TRiC structures observed in the cellular context closely resembled the structures determined by in vitro studies[13,16] (Fig 1a and Extended Data Fig. 1c–g), except for additional densities observed in the open TRiC structure (Extended Data Fig. 1c; see below). Based on the rotationally asymmetric shape of open TRiC, individual CCT subunits of the in vitro atomic model

were assigned to our density map (Extended Data Fig. 1c and Supplementary Video 2). However, we were unable to resolve the individual CCT subunits of the closed TRiC (pseudo-$D_8$ symmetry), probably because of the similar conformation of all subunits (Extended Data Fig. 1d,e, Supplementary Video 2 and Methods). Our analysis revealed 45.3% of TRiC in a symmetrically open conformation and 54.7% in the symmetrically closed conformation (Extended Data Fig. 1h). To estimate the number of false-negative detections, we manually inspected and curated 20 tomograms and estimated that around 3.3% of putative TRiC remains undetected (Extended Data Fig. 1i). Thus, in-cell analysis of the open and closed chaperonin TRiC using cryo-electron

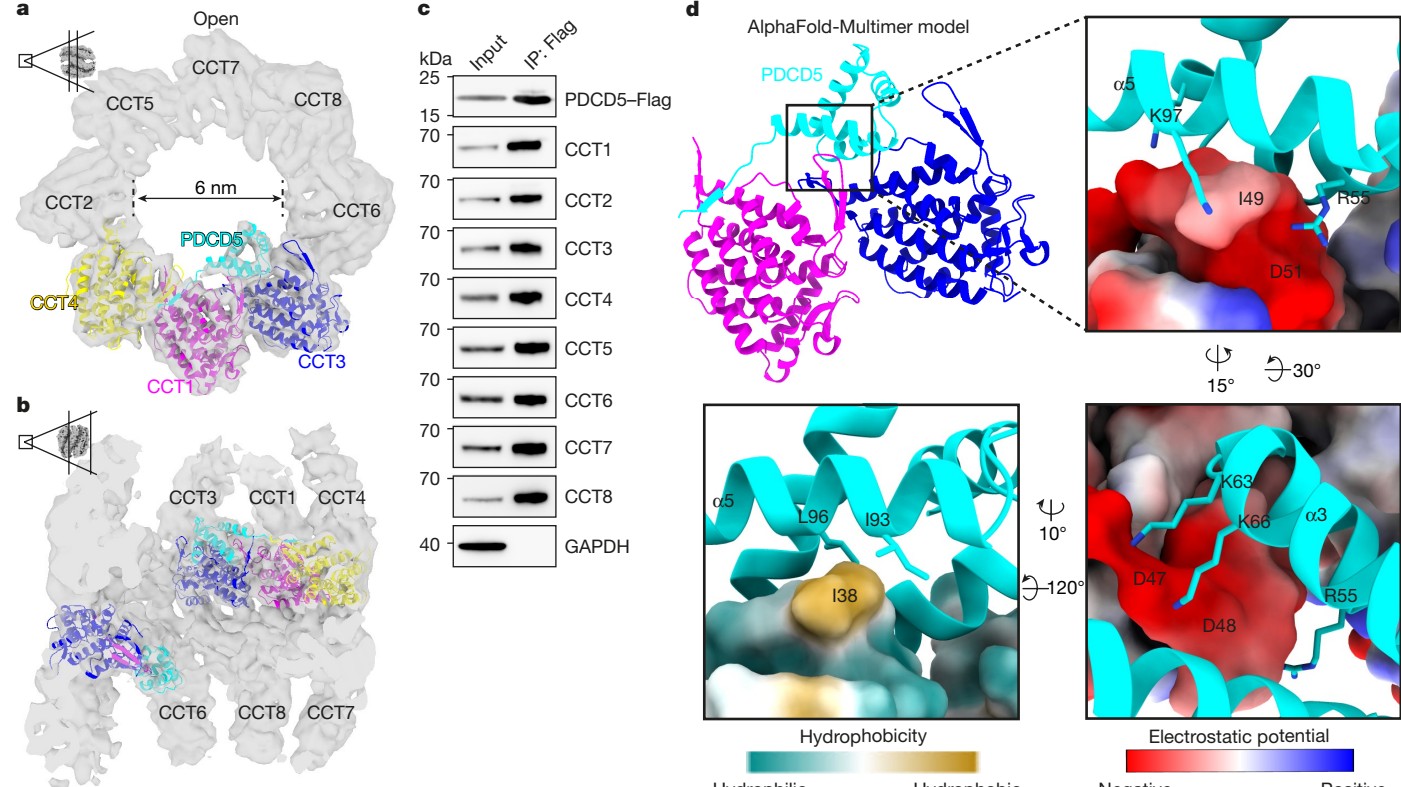

**Fig. 2 | PDCD5 binds to the open TRiC state. a**, A PDCD5–CCT3–CCT1–CCT4 model predicted by AlphaFold-Multimer was fitted into the open TRiC map and explains the additional density observed proximate to CCT1, CCT3 and CCT4 in situ. **b**, The side view of open TRiC fitted with the PDCD5–CCT3–CCT1–CCT4 model in **a**. **c**, PDCD5–Flag pull-down analysis of TRiC in HEK293F cells.

The experiment was repeated independently four times with similar results. **d**, The interface between PDCD5 and the stem–loop of CCT1 predicted by AlphaFold-Multimer. Insets: the CCT1 stem–loop coloured by electrostatic potential or hydrophobicity. The representative residues in PDCD5 and CCT1 that mediated the interaction were labelled in the predicted model.

tomography (cryo-ET) is feasible and indicates a relatively balanced distribution of the open and closed conformation in an unperturbed cell. Moreover, our analyses do not identify TRiC with one ring open and one ring closed inside cells, in contrast to what was observed for bacterial chaperonins[24].

We next investigated whether different functional states of TRiC during substrate folding can be observed in situ. We therefore conducted extensive classification of subtomograms of open TRiC with a mask focusing on the PFD-binding area (Fig. 1b, Methods, Extended Data Figs. 2 and 3, Supplementary Fig. 3 and Supplementary Table 1). For open TRiC, three classes were resolved. About one third of all TRiC was found without PFD, but contained potential substrate densities near the equatorial plane of the TRiC chamber[10,16] (Fig. 1b). This is analogous to previous in vitro structures (Extended Data Fig. 3d). Approximately 12% of all TRiC was associated with a single PFD and a very small fraction of particles (around 1%) was associated with two PFD molecules (Fig. 1b), consistent with the infrequent observation of two PFDs bound to TRiC in previous in vitro studies[25,26]. All of these open states exhibited densities in the substrate position (Extended Data Fig. 3 and Supplementary Fig. 4), indicating nearly ubiquitous substrate binding to open TRiC.

Classification of closed TRiC also resulted in three classes (Fig. 1c) with differentially shaped substrate densities, suggesting a heterogeneity of substrates and folding intermediates within the TRiC chamber. However, all substrate densities display a common feature—they localize asymmetrically underneath the lid of only one chamber[15,16] (Fig 1c,d and Supplementary Fig. 5). Notably, these analyses show that both open and closed TRiC particles almost always contain density at the substrate position in situ, arguing for a constant substrate turnover in cells, with few idle empty TRiC complexes.

## PDCD5 binds to the open TRiC

A comparison of our open TRiC structure to the published in vitro TRiC structure[13] revealed additional densities near the equatorial domains of CCT3, CCT1 and CCT4 (Extended Data Fig. 1c). In an approach to determine the identity of this density, we first analysed mass spectrometry data of a previous study[16] that identified TRiC-associated proteins in HEK293T cells. In this dataset, the protein PDCD5 was considerably enriched in the CCT5–Flag pull-down samples compared with in the control samples[16], surpassing even CCT subunits, PFD, tubulin and actin in effect size. Notably, the yeast homologue of PDCD5, SDD2, also interacts with yeast TRiC in co-immunoprecipitation (co-IP) experiments[27]. Human PDCD5 is a 125 amino acid protein that exhibits a compact core structure (from Glu38 to Ser100) with limited flexibility, featuring two α-helices in the N-terminal region and an unstructured C-terminal region[28] (Extended Data Fig. 4a,b). The core of the PDCD5 structure that was previously determined (Protein Data Bank (PDB): 2K6B) aligned well with the respective AlphaFold model in the database[29,30] (Extended Data Fig. 4a–c). Importantly, the secondary structure of the PDCD5 core (from Glu38 to Ser100) closely matched the additional density that we observed in our map (Extended Data Fig. 4d), although the flexible C terminus with low model confidence was not accommodated.

We next used AlphaFold-Multimer[31] to assess the interaction between PDCD5 and the neighbouring subunits that we had identified in our structure: CCT3, CCT1 and CCT4. The predicted complex exhibited high-confidence interactions and is highly similar to experimentally determined atomic models of PDCD5[28] and TRiC[16] (Extended Data Fig. 5a–d). The predicted structure of the PDCD5–CCT3–CCT1–CCT4 complex fitted well with our cryo-ET map (Fig. 2a,b, Extended Data Fig. 5a,e and Supplementary Video 3), in contrast to other known

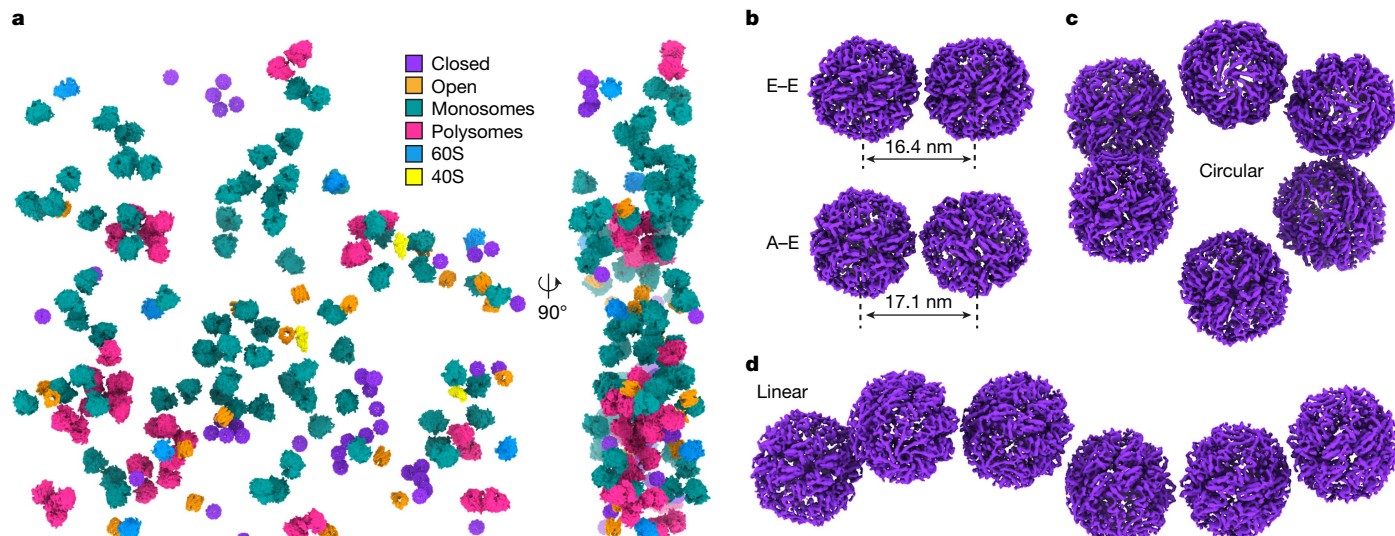

**Fig. 3 | Closed TRiC forms linearly organized clusters in situ. a**, The three-dimensional map from an exemplifying tomogram showing the spatial distribution of closed TRiC and open TRiC (this study), in comparison to mono-ribosomes, polysomes, 60S and 40S[22]. **b**, Two exemplifying closed TRiC pairs exhibiting distinct assembly patterns. **c,d**, Representative clusters of closed TRiC in a circular (**c**) or linear (**d**) arrangement.

TRiC interactors that we analysed (Supplementary Fig. 6). Consistent with our structural data, we found that PDCD5 interacts with TRiC (Fig. 2c, Extended Data Fig. 7g and Supplementary Fig. 7a,b). We also conducted AlphaFold-Multimer prediction for the seven other possible combinations of the three neighbouring CCT subunits and PDCD5. This showed that the core region of PDCD5 was specifically associated with the stem–loop of CCT1 and neither of the other subunits (Extended Data Fig. 5f). The predominant interactions were mediated by helices α3 (from Gln50 to Val62) and α5 (from Glu89 to Ser100) in PDCD5 through a combination of hydrophobic and electrostatic interactions with CCT1 (Fig. 2d) in the predicted model. In particular, the conserved residues Ile93 and Leu96 in PDCD5 form the hydrophobic interaction with CCT1 in the predicted model (Fig. 2d and Extended Data Fig. 6a,b). Confirming this interface in the predicted model, expression of PDCD5(I93G/L96G) reduced its interaction with TRiC in pull-down experiments (Extended Data Fig. 6f,g) and affected the binding kinetics of purified recombinant PDCD5(I93G/L96G) to TRiC (Extended Data Fig. 6i,j). Four positively charged residues of PDCD5 (Arg55, Lys63, Lys66 and Lys97) interacted with the negatively changed stem–loop of CCT1 in the predicted model (Fig. 2d). Arg55, Lys63 and Lys66 of PDCD5 are highly conserved across organisms (Extended Data Fig. 6a–c). Abolishing the charge of these residues (R55A/K63A/K66A, hereafter, PDCD5(RKK)) substantially reduced the expression levels in cells, precluding any conclusions on the role of these residues for the PDCD5–TRiC interaction by co-IP (Extended Data Fig. 6f). We therefore performed native gel analyses using recombinant human TRiC complex and PDCD5. In these experiments, PDCD5(RKK) binding to TRiC was severely impaired as compared to wild-type (WT) PDCD5, suggesting that these RKK residues mediate the PDCD5 interaction with TRiC (Extended Data Fig. 6h). Consistent with this observation, using sequence alignment and electrostatic potential analysis of CCT stem–loops, we observed a higher negative charge of the CCT1 stem–loop than that of other CCTs[16] (Extended Data Fig. 6d,e), providing a possible explanation for the specific binding of PDCD5 to CCT1.

In our experimental data, the PDCD5 density was observed within both rings of the open TRiC map (Extended Data Fig. 7a). Even after extensive classification with masks focusing on the PDCD5 region, we were unable to identify open TRiC without PDCD5. Our structures further show density in the substrate position in the central region of the open TRiC–PDCD5 complex, indicating that PDCD5 does not prevent substrate binding to the open form. Notably, the density observed probably corresponds to the substrate because of its position within the TRiC chamber. We cannot ultimately exclude binding of cofactors, cochaperones or mixtures thereof in this position. However, we find that simultaneous PDCD5 and PFD binding to TRiC is compatible, as we observe both densities in the open TRiC complex (Extended Data Fig. 3e–g).

By contrast, the PDCD5 density was absent in both rings of closed TRiC (Extended Data Fig. 7b). Consistently, structural superimposition shows that the closed conformation leads to a clash between the C-terminal region of PDCD5 and the CCT4 helix (Asn394 to Val416), whereas it would accommodate PDCD5 binding in the open TRiC conformation[13,16] (Extended Data Fig. 7c,d). Furthermore, the actin- and tubulin-binding sites[16] within the closed TRiC cavity partly overlap with the region occupied by PDCD5 (Extended Data Fig. 7e,f). The absence of a PDCD5 density in the structure of closed TRiC could, in principle, also be explained by PDCD5 becoming too flexible and therefore too heterogenous for structural detection. To differentiate between flexible PDCD5 remaining in the closed TRiC chamber and its dissociation during TRiC closure, we expressed Flag-tagged PDCD5 in HEK cells and assessed its interaction with TRiC using co-IP followed by immunoblotting. We immunoprecipitated the TRiC–PDCD5–Flag complex in buffer without ATP–AlFx (Methods) and subsequently induced TRiC closure by incubating the precipitated complex on beads with ATP–AlFx (1 h, 37 °C)[13,14,16,21]. After closure, we observed CCT1 enrichment (TRiC) in the supernatant, indicating a release of TRiC from bead-bound PDCD5 (Extended Data Fig. 7h–j). Consistently, native gel analysis of the interaction between purified PDCD5 and TRiC also showed that PDCD5 bound to open TRiC (Extended Data Fig. 7k). This interaction was disrupted by ATP–AlFx, which induces TRiC closure, but not by other nucleotide analogues. Taken together, our data suggest that the closed TRiC complex does not associate with PDCD5, whereas the open TRiC complex does. Although previous mass spectrometry studies have listed PDCD5 as a candidate CCT interactor in different cell types and organisms (HEK293, HeLa and *Saccharomyces cerevisiae*)[16,27,32,33], the importance and role of this interaction was unclear and no high-resolution structure of PDCD5-bound TRiC isolated from cells has been reported to date. This may indicate that PDCD5 associated with TRiC in cells dissociates during purification.

Notably, PDCD5 is thought to have an early and universal role promoting apoptosis[34]. To gain insights into the functional link between PDCD5 and TRiC, we used the Dependency Map (DepMap) portal[35] and the *Saccharomyces* Genome Database[36] to assess genetic co-dependencies in human and yeast, respectively. Both analyses revealed strong genetic

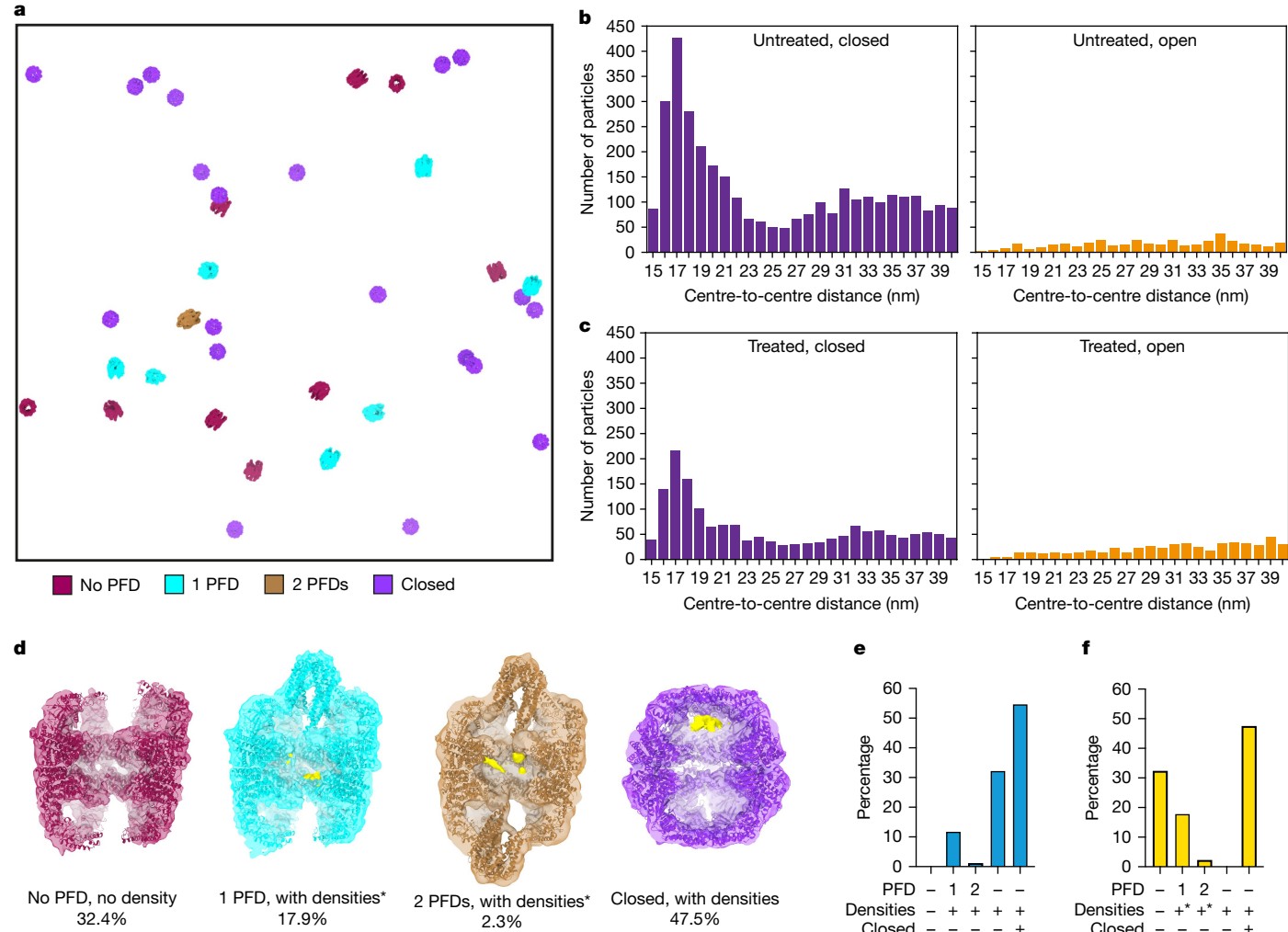

**Fig. 4 | Translation inhibition diminishes substrate in all states and clusters of closed TRiC reduce in HHT-treated cells. a**, TRiC states mapped into a tomogram of translation-inhibited (HHT treated) cells. Clusters are not apparent. **b**, Quantification of TRiC complexes belonging to clusters at thresholds ranging from 15 to 40 nm (centre-to-centre distance) in untreated cells. The analyses were performed for 4,054 closed TRiC (left) and 3,353 open TRiC (right) particles from 360 tomograms. **c**, The counts of detected TRiC clusters within 15 to 40 nm (centre-to-centre distance) in treated cells. The analyses were performed for 3,418 closed TRiC (left) and 3,785 open TRiC (right) particles from 352 tomograms. **d**, Classes of TRiC states obtained by subtomogram averaging in HHT-treated (translation inhibited) cells. Densities in the substrate position are shown in yellow. The substrate densities were Gaussian filtered (sDev = 4) for visualization. **e,f**, Quantification of the TRiC state distribution under untreated (**e**) and HHT-treated (**f**) conditions. Asterisks in **d**–**f** indicate reduced densities in the chamber of TRiC.

interactions between PDCD5 and the TRiC folding network. In DepMap, the top six PDCD5 co-regulated genes were directly involved in TRiC folding, including several PFDs and CCT1 (Supplementary Fig. 7c). Similarly, SDD2 (the homologue of PDCD5 in yeast) exhibits negative genetic interactions with genes associated with TRiC (Supplementary Fig. 7d). Furthermore, we analysed the aggregation propensity of two prominent TRiC substrates, actin and tubulin, in commercial CRISPR-mediated *PDCD5*-knockout cells using thermal profiling. Cells were thereby exposed to different temperatures to induce the aggregation of proteins to probe their stability. These experiments revealed substrate stabilization in *PDCD5*-knockout cells (Supplementary Fig. 7e–h). Although we cannot rule out indirect effects, these results are consistent with a model in which PDCD5 is functionally associated with TRiC activity.

## Closed, but not open, TRiC forms clusters

Analysis of the spatial distribution of the TRiC states in native untreated cells indicated that closed TRiC formed clusters of two or more particles (Fig. 3a–d, Extended Data Fig. 8a–d and Supplementary Videos 4 and 5).

By contrast, open TRiC particles were randomly distributed across the cytosol (Fig. 3a and Supplementary Video 4). The clusters were not specific to any closed TRiC class (Extended Data Fig. 8b,c). TRiC pairs with different arrangement patterns were observed (Fig. 3b), including ones for which the equatorial domain was close to its neighbour's equatorial domain (denoted E–E) or the apical domain towards the neighbour's equatorial domain (denoted A–E). Angular analysis revealed a random orientation of TRiC to its nearest neighbours in the cluster (Extended Data Fig. 8d), suggesting the absence of specific interfaces. However, the cluster did not form globular arrangements. Instead, they adopted distinctive topologies, such as circular and linear organizations (Fig. 3c,d), indicating an inherent flexibility.

## TRiC clusters are unrelated to polysomes

To further understand the observed cluster organization, we assessed the cellular environment surrounding TRiC clusters in our tomograms (Extended Data Fig. 8e–g and Supplementary Videos 5 and 6). There were no discernible associations with specific organelles, including

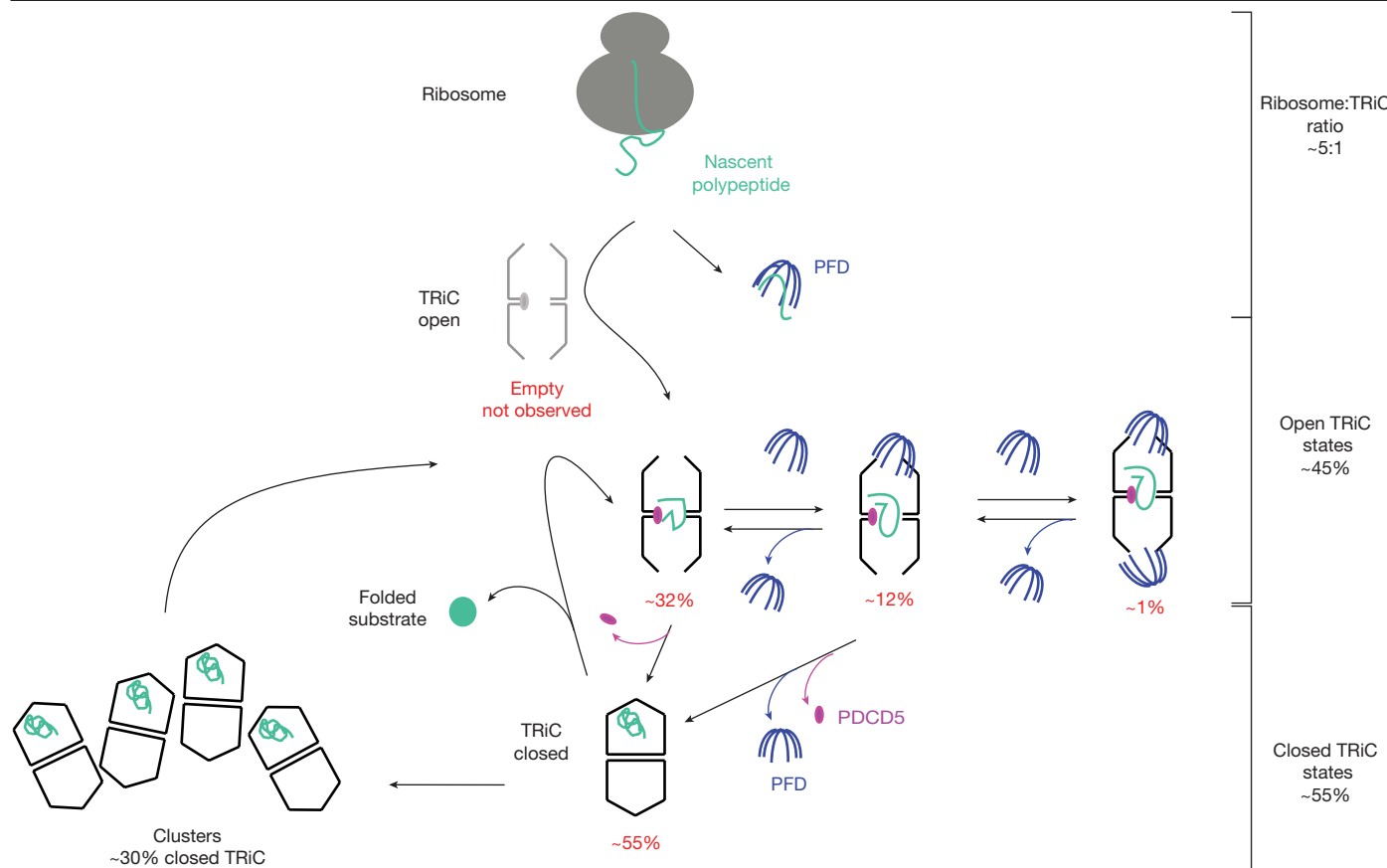

**Fig. 5 | The in vivo TRiC folding cycle in human cells.** Schematic of the proposed TRiC duty cycle inside human cells. Substrates are delivered to TRiC by PFD or directly enter the chamber of open TRiC. Open TRiC is abundant and not observed without PDCD5 and substrate (for the latter unless translation is inhibited). ATP hydrolysis drives TRiC closure, PDCD5 dissociates and closed TRiC tends to form linear clusters. Once substrates are folded, TRiC opens, and the substrates are released. PDCD5 binds to the open TRiC complex to initiate a new cycle. The abundance of TRiC states as observed in our analysis is listed for each step of the cycle.

mitochondria, the endoplasmic reticulum or the Golgi apparatus, nor were there any apparent connections with membranes in the proximity of TRiC clusters. About 37% of TRiC clusters were spatially close to actin filaments (Methods). However, given the high abundance of actin filaments in the cytosol, this could well be a stochastic event.

Previous research has demonstrated that TRiC can interact with the nascent protein chain to support cotranslational folding[37,38]; however, this has not yet been visualized inside cells. To investigate this in situ, we analysed the spatial interplay between ribosomes and TRiC (Methods) and found that both open and closed TRiC were slightly enriched at the exit tunnel side (ETS) of the ribosome (Extended Data Fig. 9a–c and Supplementary Fig. 8a–c). In light of the linear topology observed for the clusters, we wondered whether polysomes may function as primers for their formation. Our data showed that both TRiC clusters and individual closed TRiC complexes were similarly abundant in the vicinity of the ribosome (Supplementary Fig. 8d,e and Supplementary Video 7). TRiC clusters were only occasionally found close to the ribosome exit tunnels in polysomes (Supplementary Fig. 8d,f and Supplementary Video 8), arguing against polysomes as primary organizers of the TRiC clusters. The exact function and emergence of these clusters, suggestive of spatial organization of closed TRiC states, therefore remains to be further investigated in the future (Discussion).

## Translation inhibition fades substrate

If TRiC is indeed important for the folding of newly synthesized proteins[39], we wondered whether the presence of substrates within the TRiC chamber and the occurrence of the functional states observed in our study were dependent on active translation. Homoharringtonine (HHT) inhibits translational activity by targeting the peptidyl transferase centre of the ribosome[22,40,41]. While, in translation-inhibited cells, closed TRiC also formed clusters, their abundance (~19%) was reduced compared with in untreated cells (~30%) (Fig. 4a–c and Extended Data Fig. 9d–g). Moreover, the cluster length (particle number per cluster) decreased after HHT treatment (Extended Data Fig. 9d,e), although the distance between TRiC neighbours remained similar under both conditions (Extended Data Fig. 9h,i). Notably, the slight enrichment of TRiC around the ETS of the ribosome was diminished in translation-inhibited cells (Extended Data Fig. 9j,k). These analyses indicate that the cluster number and length, as well as the distribution of TRiC around the ribosome, are correlated with translational activity.

To further examine how translational activity impacts the dynamics of TRiC in the cell, we classified subtomograms of open TRiC from translation-inhibited cells (Supplementary Figs. 1 and 2d–f and Methods). Notably, the duty cycle of TRiC is almost unaffected by inhibiting translation. The abundance of open TRiC without PFD was comparable to that in untreated cells (Fig. 4d–f, Extended Data Fig. 10, Supplementary Fig. 9 and Supplementary Table 1). The numbers of single-PFD-bound (~18%) or double-PFD-bound (~2%) TRiC were slightly increased compared with in untreated cells (Fig. 4d–f and Supplementary Video 9). The overall architectures of closed TRiC were indistinguishable in untreated and translation-inhibited cells. Importantly, less density was observed in the substrate position for all the TRiC states in translation-inhibited cells (Figs. 1b and 4d

and Extended Data Figs. 3 and 10). These analyses indicate that substrate binding to TRiC is correlated with translational activity in cells, whereby the distribution of functional states of TRiC is not fundamentally changed.

## Discussion

Taken together, based on our results, we propose the following model for the in vivo TRiC folding cycle (Fig. 5). Initially, substrates, probably originating from translation, are delivered either by binding of PFD—in some cases by two PFDs on either side of the TRiC—or possibly also by directly binding to the cavity of the open TRiC, close to the equatorial plain. Such PFD-independent binding was previously speculated on[14,17], and may involve additional chaperones that were not captured by our analysis. Next, ATP hydrolysis[2] leads to symmetric closure of both TRiC rings accompanied by PDCD5 disassociation from the stem–loop of CCT1. After closure, the substrates translocate from the equator towards the lid region within the closed chamber. The substrates appear to bind to slightly different regions in the chamber (Fig. 1c,d), as seen with in vitro studies of actin, tubulin and Gβ5 bound to TRiC[15,16]. Closed TRiC is abundant in actively translating cells, making up about half of all particles. When TRiC reopens during ATPase cycling, the folded substrates are released. We propose that PDCD5 then rebinds to CCT1 of the open TRiC conformation and a new folding cycle is initiated subsequently. After HHT treatment, we observed an absence of substrate densities in PFD-free open TRiC (Fig. 4e,f). This finding may indicate that substrates in the chamber of open PFD-free TRiC reflect newly translated proteins, which would be absent after translation inhibition.

Our finding that nearly all detected TRiC, both open and closed, is engaged with substrates indicates that its idle capacity in proliferating and actively translating cells is low. This result would probably not have been expected for a chaperone that should be 'ready' to rapidly act in response to stress, but is consistent with previous reports that TRiC is not induced by heat stress, in contrast to many other chaperones. This finding contrasts the recent in situ analysis of the prokaryotic chaperonin GroEL-ES that becomes functionally important under stress conditions[42]. Our model that TRiC substrates are probably predominantly newly translated proteins may provide an explanation to this conundrum, as protein synthesis is inhibited under stress conditions and substrates for TRiC would become sparse.

We observed that closed TRiC can form beads-on-a-string-like clusters, with varying angles and lengths. Although it remains unclear how these clusters form, their presence appears to be associated with translational activity, given that they are less frequent in HHT-treated cells. One may speculate that large substrate proteins act as a bridge between closed TRiC complexes. The TRiC cavity is only large enough to enclose proteins of up to approximately 70 kDa[43,44] yet, in mammalian cells, many larger proteins are reported to be interactors of TRiC[44,45]. Closed TRiC has several holes with diameters of around 5 Å in the top lids or between subunits[14] that could accommodate extended linker sequences connecting domains in multidomain proteins. Thus, a co-folding model whereby multiple TRiC entities fold one large substrate could be conceivable. Alternatively, cluster formation may be mediated by closed-state-specific TRiC interactors that are not resolved in our analysis. If and how the cell benefits from cluster formation remains unclear. One conceivable model could be that spatial proximity between folding-active TRiC complexes facilitates rebinding throughout iterative TRiC-mediated folding cycles, and prevents aggregation of newly synthesized proteins. Notably, localized accumulation of TRiC complexes was observed at the periphery of poly(GA) aggregates in rat neurons expressing (GA)$_{175}$–GFP[46], while our findings strongly suggest that the formation of TRiC clusters occurs as a more general phenomenon, probably related to the presence of substrate.

Our study defines the structural states and spatial organization of the chaperonin TRiC in human cells. On the basis of our cryo-ET results, AlphaFold-Multimer predictions, biochemical analyses, and previously reported mass spectrometry and genetic interaction data[16,32], we report that PDCD5 is a major TRiC interactor specifically bound to the open conformation. The interaction is mediated by the helices α3 and α5 in PDCD5 and the stem–loop of CCT1. Our experimental data underscore the physical and functional association of PDCD5 as a co-factor of open TRiC. Although we do not yet elucidate the precise molecular effect that PDCD5 binding has on TRiC function, the apparently exclusive interaction with the large majority of the cellular open TRiC suggests a role in modulating TRiC substrate recruitment and the folding cycle. This would be supported by the observation that actin and tubulin are stabilized in *PDCD5*-knockout cells. However, we did not observe that PDCD5 affects TRiC ATPase activity in vitro (Supplementary Fig. 10). Importantly, while TRiC function is essential, *PDCD5*-knockout cells are viable. Thus, PDCD5 may have a regulatory function. Given its reported role in initiating apoptosis[34], it is also tempting to speculate that PDCD5 could potentially act as a sensor for TRiC activity, where the abundance of free PDCD5 would indicate the level of closed and, therefore, occupied TRiC in a cell.

A very recent structural report that reconstituted the complex of the co-chaperone PhLP2A and TRiC in vitro[21] showed that PhLP2A binds to a similar, yet not identical, interface of TRiC as PDCD5 does in situ. However, in clear contrast to PDCD5, PhLP2A binds to both open and closed TRiC conformations, in a mutually exclusive manner with PFD, and does not localize to the same TRiC chamber as the substrate[21]. Notably, we did not observe TRiC with densities that would correspond to PhLP2A in our structures and it is therefore unlikely that PhLP2A is constitutively or abundantly associated with TRiC inside cells, in stark contrast to PDCD5. This is consistent with overall steady-state protein abundance information in public databases that lists PDCD5 as similarly abundant to TRiC, contrasting with a low abundance of PhLP2A within cells[47].

In summary, we pushed the boundaries of in situ structural analysis to gain, in combination with in vitro experiments, a deeper understanding of the TRiC folding machine operating in its native environment inside human cells. We propose a model for its folding cycle that identifies both open and closed TRiC in their symmetric conformations as equally abundant inside cells, where PDCD5 is an open-state specific cofactor. We show that the TRiC chamber is constitutively occupied in unperturbed human cells, while this is diminished when translation is inhibited. These findings underline the importance of TRiC in processing newly synthesized proteins, thereby contrasting prokaryotic chaperonins.

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

## Methods

### Cell culture

HEK (HEK Flp-In T-Rex 293, Invitrogen) cells were cultured in DMEM (Sigma-Aldrich) supplemented with 10% fetal bovine serum (Gibco) under standard tissue culture conditions (37 °C, 5% $CO_2$). HEK293F (Thermo Fisher Scientific) cells were cultured in Freestyle medium (Thermo Fisher Scientific) at 37 °C, 8% $CO_2$ and 120 rpm. Cells were negative for mycoplasma contamination.

### Native PAGE

For immunoblotting, when HEK cells were at about 80% confluency, they were washed twice with ice-cold PBS and scraped in PBS, pelleted by centrifugation for 5 min at 1,000$g$, 4 °C and resuspended in modified native lysis buffer (50 mM HEPES pH 7.4, 50 mM KCl, 1.5 mM MgCl$_2$, 10% glycerol, 0.1% NP-40, 1 mM PMSF, complete EDTA-free protease inhibitor cocktail and 1 mM DTT). Lysis buffer was also supplemented with 30 U ml$^{-1}$ benzonase to remove DNA. Lysis was performed on ice for 20 min and the lysates were clarified by centrifugation for 10 min at 12,000$g$ at 4 °C. The protein concentration was determined using a BCA assay (Thermo Fisher Scientific). 4× NativePAGE sample buffer (Thermo Fisher Scientific) was added to a final concentration of 1×. Then, 15 μg of each sample was resolved on 3–12% Bis-Tris NativePAGE gels (Thermo Fisher Scientific). NativePAGE was soaked in 0.1% SDS buffer for 15 min, then transferred to 0.45 μM PVDF membranes presoaked in methanol for 30 s. The membranes were blocked with 5% molecular biology grade BSA (Millipore Sigma) in Tris-buffered saline supplemented with 0.1% Tween-20 (TBST) for 1 h at room temperature, then probed with specific primary antibodies 4 °C for overnight. Primary antibodies was diluted in 1% BSA/TBST as follows: 1:10,000 rabbit anti-CCT5 (Abcam, ab129016). The secondary antibody was diluted 1:10,000 in TBST. Total protein was detected with Revert total protein stain. Fluorescence signal detection was performed using Li-Cor Odyssey infrared imager.

### *PDCD5* knockdown

HEK cells ($5 × 10^5$) were seeded into six-well plates. Then, 24 h after plating, 25 pmol siRNA (Thermo Fisher Scientific, s17467) were added with Lipofectamine RNAiMAX Transfection Reagent (Invitrogen). Cells were collected with ice-cold PBS after 48 h and then immunoblotting was run for further analysis.

### Expression and purification of recombinant PDCD5 and its mutants

PDCD5 mutants were obtained using site-directed mutagenesis. A 6× His-tag was added to the C terminus of PDCD5. Plasmids containing WT and mutant PDCD5 were transformed into *Escherichia coli* Rosetta DE3 competent cells for expression. PDCD5 was expressed and purified as previously reported[33]. In brief, cell lysates were first passed through a nickel column, then PDCD5 bound to the nickel resin was eluted in high imidazole buffer, and pure PDCD5 was obtained by passing the elution twice through a Superdex 200 size-exclusion column. Proteins were concentrated by centrifugation and then quantified using the BCA colorimetric assay.

### TRiC ATPase activity

The assay was performed as previously described[48]. In brief, stock solutions of 0.05% (w/v) quinaldine red, 2.32% (w/v) polyvinyl alcohol, 5.72% (w/v) ammonium heptamolybdate tetrahydrate in 6 M HCl and water were mixed in a 2:1:1:2 ratio to prepare the quinaldine red reagent fresh before each experiment. Then, 300 nM TRiC was diluted in ATPase buffer (50 mM Tris-HCl pH 7.4, 100 mM KCl, 5 mM MgCl$_2$, 10% glycerol, 1 mM TCEP; 30 μl total reaction volume), preheated to 37 °C and added to 3 μl water or 10 mM ATP to start the reaction, then incubated for the indicated durations in the presence or absence of 3 μM PDCD5. The reactions were stopped by the addition of 5 μl of 60 mM EDTA in a

Corning 96-well opaque non-sterile polystyrene plate (Sigma-Aldrich, CLS3992) on ice. After samples at all timepoints were collected, the reactions were developed by adding 80 μl quinaldine red reagent for 10 min, then quenched by adding 10 μl 32% (w/v) sodium citrate. The fluorescence intensity was measured (excitation, 430 nm; emission, 530 nm) using the CLARIOstar plate reader (BMG Labtech). Analysis was performed by fitting a phosphate standard curve with a one-phase decay function, and we derived the parameters for calculating the amount of phosphate released from CCT complexes.

### PDCD5 binding to TRiC

To probe the binding affinity of PDCD5 for TRiC, increasing amounts of recombinant PDCD5 variants were incubated with a fixed concentration of TRiC (300 nM) for 20 min at 25 °C in ATPase buffer (50 mM Tris-HCl pH 7.4, 100 mM KCl, 5 mM MgCl$_2$, 10% glycerol, 1 mM TCEP), in the absence of ATP. The reactions were run in native gels and immunoblotted using PDCD5 or CCT8 antibodies, as described above. To test whether PDCD5 binds to the TRiC open or closed conformations, 3 μM of WT or mutant PDCD5 was incubated with 300 nM TRiC for 20 min at 25 °C in ATPase buffer containing 1 mM of different nucleotides and ATP analogues. The reactions were run in native gels and immunoblotted using PDCD5 or CCT8 antibodies, as described above. To obtain insights about the binding kinetics of PDCD5 variants to TRiC, 3 μM of WT or mutant PDCD5 was incubated with 300 nM TRiC in ATPase buffer at 25 °C for 10, 15, 20 and 30 min. The reactions were run in native gels and immunoblotted using PDCD5 (Proteintech, 12456-1-AP, 1:1,000) and CCT8 (Santa Cruz Biotechnology, sc-377261, 1:250) antibodies, as described above.

### Co-IP

For PDCD5–Flag co-IP, PDCD5-Flag constructs (GenScript) were transiently expressed in HEK293F for 48 h after transfection. Cells were washed with PBS before collection by centrifugation and frozen in liquid nitrogen. HEK293F cells were lysed in lysis buffer (PBS pH 7.4, 0.1% IGEPAL CA-630, 5 mM MgCl$_2$, freshly added 0.6 mM phenylmethylsulphonyl fluoride and protease inhibitors), triturated through a 24-gauge needle ten times and incubated on ice for 5 min. After lysate clearing by centrifugation, 500 μg clarified protein extract was mixed with 20 μl packed anti-Flag M2 beads (Sigma-Aldrich) and incubated for 1 h at 4 °C. After three washes with lysis buffer, bound proteins were eluted by boiling in LDS sample buffer (Invitrogen). For western blotting, input and eluate (IP) samples were loaded onto 4–12% Bis-Tris gels (Invitrogen) and subsequently transferred to nitrocellulose membranes (Bio-Rad).

CCT3 co-IP was performed with non-transfected HEK293F cells subjected to in vivo cross-linking with 1.5 mM dithiobis(succinimidyl propionate) (DSP; Thermo Fisher Scientific) at 37 °C for 10 min. The cross-linking reaction was quenched by the addition of Tris (pH 8.0) to a final concentration of 160 mM and cells were collected and lysed as described above. Then, 2 mg of clarified protein extract was mixed with 10 μg rabbit anti-CCT3 antibody (Proteintech, 10571-1-AP) or rabbit control IgG (Proteintech, 30000-0-AP) as mock IP for 1 h at 4 °C, followed by addition of 50 μl equilibrated Protein G Magnetic Beads (Thermo Fisher Scientific) and incubation for 1 h at 4 °C. The samples were washed, eluted and evaluated using SDS–PAGE as described above.

The percentage of IP efficiency was calculated by normalizing the measured intensities and the respective dilution factor of the loaded sample for western blotting (1% for the input sample and 5% for the IP sample), followed by IP/input. For the quantification, the mean ± s.d. values were as follows: PDCD5–flag (42.70 ± 16.16), CCT1 (86.66 ± 41.01), CCT2 (45.54 ± 15.25), CCT3 (45.57 ± 12.47), CCT4 (61.12 ± 15.08), CCT5 (98.98 ± 27.74), CCT6 (53.74 ± 21.34), CCT7 (65.99 ± 38.51), CCT8 (135.49 ± 64.48) and GAPDH (0.03 ± 0.06), with $n$ representing the number of biologically independent experiments ($n = 4$). For the quantification of PDCD5 mutation experiments, the mean ± s.d. values were as follows: WT (100 ± 0), RKK (133.65 ± 59.63) and IL (11.04 ± 9.68), with

$n$ representing the number of biologically independent experiments ($n = 4$).

To induce TRiC closure during co-IP, beads bound with TRiC–PDCD5–Flag (from co-IP, see above) were incubated in ATP/AlFx buffer (lysis buffer supplemented with 5 mM Al(NO$_3$)$_3$, 30 mM NaF and 1 mM ATP) for 1 h at 37 °C, followed by three washes with ATP/AlFx buffer. As a control, the beads bound with TRiC–PDCD5–Flag (from co-IP, see above) were incubated and washed in lysis buffer without the ATP/AlFx. For western blotting, 1% of input, 25% of released proteins after ATP/AlFx incubation and 25% of eluates (denoted as beads) were loaded.

Without adding ATP in the TRiC sample before plunge freezing, around 100% TRiC particles are at open conformation based on the single-particle analysis[13,14,19]. With extra ATP/AlFx in TRiC solution before plunge freezing, a portion of TRiC particles were closed, although different papers show different closed/open ratios with ATP/AlFx at different conditions. Closed/open ratio: ~1.7 in buffer (1 mM ATP, 5 mM MgCl$_2$, 5 mM Al (NO$_3$)$_3$ and 30 mM NaF) from ref. 13; ~5.1 in buffer (1 mM ATP, 1 mM Al$_3$(NO$_3$)$_3$, 6 mM NaF, 10 mM MgCl$_2$ 50 mM KCl) from ref. 21; ~0.6 in buffer with ATP-AlFx from ref. 14; and ~2.2 in buffer (1 mM ATP, 5 mM MgCl$_2$ and AlFx (5 mM Al(NO$_3$)$_3$ and 30 mM NaF) from ref. 16. In our experimental settings (Extended Data Fig. 7), we used the conditions from ref. 13 (1 mM ATP, 5 mM MgCl$_2$, 5 mM Al (NO$_3$)$_3$ and 30 mM NaF).

For the quantification in Extended Data Fig. 7, the mean ± s.d. values were as follows: PDCD5 (ATP/AlFx) (0.09 ± 0.05); PDCD5 (control) (0.10 ± 0.04); CCT1 (ATP/AlFx) (1.53 ± 0.51); and CCT1 (control) (0.38 ± 0.06); with $n$ representing the number of biologically independent experiments ($n = 4$).

## Thermal protein profiling (heat-shock treatment of cells)

WT (Abcam, ab255449) and *PDCD5*-knockout HEK293T cells (Abcam, ab266229) were used for the heart-shock assay and cultured in DMEM (Sigma-Aldrich) supplemented with 10% fetal bovine serum (Gibco) at 37 °C with 5% CO$_2$. The experiment was conducted as described previously[49,50]. In brief, cells were collected and resuspended in PBS. Five aliquots were prepared and distributed into PCR tubes, each of the tubes containing 5 × 10$^5$ cells. Each tube was incubated for 3 min at various temperatures (37.0, 44.1, 49.9, 55.5 and 62.0 °C; or 56.8, 58.3, 59.5, 60.7 and 62.1 °C). The cells were then lysed in a buffer containing 1.5 Mm MgCl$_2$, 0.8% NP-40, 0.4U µl$^{-1}$ benzonase and protease inhibitor for 40 min at 4 °C. Protein aggregations were removed, and the soluble fraction was used for western blotting. For quantification of the western blotting of thermal protein profiling, the mean ± s.d. values of actin in WT cells at 37.0 °C to 62.0 °C were as follows: 100.0 ± 0.0, 85.3 ± 5.2, 73.8 ± 7.7, 46.3 ± 2.9 and 26.3 ± 9.4; the mean ± s.d. values of actin in *PDCD5*-knockout cells at 37.0 °C to 62.0 °C were as follows: 100.0 ± 0.0, 100.3 ± 7.0, 109.0 ± 9.7, 83.0 ± 2.0 and 57.6 ± 9.4; the mean ± s.d. values of tubulin in WT cells at 56.8 °C to 62.1 °C were as follows: 100.0 ± 0.0, 78.2 ± 4.2, 49.3 ± 5.5, 20.0 ± 4.9 and 5.4 ± 3.8; and the mean ± s.d. values of tubulin in *PDCD5*-knockout cells at 56.8 °C to 62.1 °C were as follows: 138.0 ± 22.3, 99.7 ± 6.4, 63.9 ± 15.9, 34.8 ± 0.4 and 8.3 ± 4.7.

## Antibodies

Membranes from western blotting were incubated with primary antibodies (mouse anti-Flag M2 (Sigma-Aldrich, F1804, 1:2,000), rabbit anti-PDCD5 (Abcam, ab126213, 1:1,000), rabbit anti-CCT1 (Abcam, ab240903, 1:10,000), rabbit anti-CCT2 (Abcam, ab92746, 1:10,000), rabbit anti-CCT3 (Proteintech, 10571-1-AP, 1:30,000), rabbit anti-CCT4 (Proteintech, 21524-1-AP, 1:5,000), rabbit anti-CCT5 (Proteintech, 11603-1-AP, 1:3,000), rabbit anti-CCT6 (Proteintech, 19793-1-AP, 1:1,000), rabbit anti-CCT7 (Abcam, ab240566, 1:30,000), rabbit anti-CCT8 (Proteintech, 12263-1-AP, 1:2,000), rabbit anti-GAPDH (Proteintech, 10494-1-AP, 1:15,000), mouse anti-actin (Invitrogen, AM4302, 1:3,000), mouse anti-tubulin (Sigma-Aldrich, T5168, 1:3,000)), followed by incubation with HRP-conjugated secondary antibodies (anti-rabbit IgG (Cell Signaling, 7074, 1:10,000), anti-mouse IgG + IgM (Jackson ImmunoResearch, 115-035-044, 1:10,000)). Uncropped western blots are provided as Source Data.

## Grid preparation, data acquisition and tomogram reconstruction

Cryo-ET sample preparation, data collection and tomogram reconstruction were performed essentially as described previously[22]. In brief, R2/2 gold grids with 200 mesh (Quantifoil) were glow discharged for 90 s and were positioned in 3.5 cm cell culture dishes (MatTek). Then, 2 ml HEK Flp-In T-Rex 293 cell suspension, with a concentration of 175,000 cells per ml, was added to the dish. For untreated samples, cells were cultured for 5 h before plunge-freezing. For HHT-treated samples, cells were cultured without HHT for 3 h and subsequently exposed to HHT (Santa Cruz Biotechnology) at a final concentration of 100 µM for 2 h before the plunge-freezing process. The grids were blotted from the backside for 6 s using the Leica EM GP2 plunger under 70% humidity and 37 °C. The grids were rapidly plunged into liquid ethane and stored in liquid nitrogen. Grids were FIB-milled using Aquilos FIB-SEM (Thermo Fisher Scientific). The samples were sputter-coated with an organometallic protective platinum layer using the gas injection system for 15 s. Lamella preparation was performed through a stepwise milling process with gallium ion-beam currents decreasing from 0.5 nA to 30 pA.

The data acquisition area was focused on the cytoplasmic region within the cell. Tilt series were acquired on a Titan Krios G4 (Thermo Fisher Scientific) operated at 300 kV, and equipped with Selectris X imaging filter and Falcon 4 direct electron detector, at 4,000 × 4,000 pixel dimensions, pixel size of 1.188 Å, a total dose of 120 to 150 e Å$^{-2}$ per tilt series, 2° tilt increment, tilt range of −60° to 60° and target defocus of −1.5 to −4.5 µm, using SerialEM software[51]. Tilt series were aligned automatically using the IMOD package[52]. The alignment files generated from IMOD were used for tomogram reconstruction in Warp[53] v.1.0.9.

## Particle localization and refinement

Template matching was performed similarly to previous studies[22,54]. For this work, the parameters were set as follows: 5° angular scanning step, low-pass filter radius=20, high-pass filter radius=1, apply_laplacian=0, noise_corelation=1 and calc_ctf=1. The cryo-EM map (EMD-32822)[14] of TRiC downloaded from the Electron Microscopy Data Bank (EMDB) was used as the template covered by a sphere mask. The above optimized setting produced distinguished peaks visualized in napari[55] (Extended Data Fig. 1b and Supplementary Video 1). To analyse all potential TRiC complexes within the datasets, we extracted the top 1,000 peaks per tomogram. The selection was based on the constrained cross-correlation (CCC) value from template matching, and these chosen coordinates were subsequently extracted as subtomograms in Warp. In total, 360,000 untreated and 352,000 treated subtomograms were extracted. 3D classifications (classes = 4, T = 0.5, iterations = 30, without mask) and refinements ($C_1$ symmetry) were performed in RELION[56] v.3.1. In total, 3,353 open TRiC particles and 4,054 closed TRiC particles in the untreated dataset, and 3,785 and 3,418 in the treated dataset were identified. Open TRiC particles from untreated and treated datasets were combined and refined to improve map resolution. Closed TRiC particles were merged from untreated and treated datasets and refined with $C_1$ or $D_8$ symmetry. Actin filaments were manually picked in ten tomograms. In total, 1,490 subtomograms were extracted and refined at bin4. Atomic models obtained from the PDB (7X3J, 7NVN, 7NVO, 7NVL, 7NVM and 8F8P)[13,16,57] were fitted into our maps. ChimeraX[58,59] was used to visualize EM maps and models.

## Subtomogram classification of TRiC states

For 3,353 open TRiC particles in the untreated dataset, classification with a sphere mask covering the potential PFD region (classes = 3, T = 3, iterations = 50, $C_1$ symmetry) of one ring (denoted ring1) was performed (Extended Data Fig. 2a), which generated 2,874 particles without PFD and 479 particles with PFD of ring1. Independently, the

same classification was performed with a mask focused on the other ring (denoted ring2), which produced 2,791 particles without PFD and 562 particles with PFD of ring2. In total, 2,395 particles without PFD, 875 particles with 1 PFD and 83 particles with 2 PFD were identified by sorting particles based on the above two classifications. The same classification strategy was applied to 3,785 open TRiC particles in the treated dataset, resulting in 2,334 particles without PFD, 1,287 particles with 1 PFD and 164 particles with 2 PFD. The atomic model (PDB: 7WU7)[14] was fitted into the maps with PFD. Different classification parameters were evaluated in attempts to resolve the density in the chamber of TRiC, but this did not result in meaningful insights. The densities inside the TRiC chamber were Gaussian filtered (sDev = 2 or 4) for visualization in Figs. 1b and 4d and Extended Data Figs. 3 and 10. For closed TRiC, 3D classification (classes = 4, T = 3, iterations = 35, $C_1$ symmetry) was performed in untreated and treated datasets independently in RELION 3.1, which revealed several classes with different densities occupied in the chamber of the closed TRiC. Further classification with a mask focusing on the substrate position did not produce meaningful results (Supplementary Figs. 4 and 5). Fourier shell correlation (FSC) was calculated in RELION 3.1.

### AlphaFold-Multimer model of the CCT3–CCT1–CCT4–PDCD5 complex

The structure of human PDCD5 in a complex with human CCT3, CCT1 and CCT4 was predicted using AlphaFold-Multimer[31] (v.2.2.0). The prediction was executed using the default setting with AMBER relaxation, and 15 models were generated for each prediction. The same prediction setting was used for PDCD5 with the other CCT combinations. The full-length amino acid sequences of PDCD5 (UniProt: O14737)[60] and the equatorial domain of CCT1–CCT8 (the sequences were the same as PDB 7NVO) were used for the above prediction. The monomeric model of PDCD5 (AF-O14737-F1) was downloaded from the AlphaFold Protein Structure Database[30].

### Sequence alignment

Sequence alignment of CCT1–CCT8 (UniProt: P17987, P78371, P49368, P50991, P48643, P40227, Q99832 and P50990) was executed through Clustal Omega[61]. Sequence alignment of PDCD5 (UniProt: *M. maripaludis*, A9A8D7; *S. pombe*, O13929; *C. elegans*, Q93408; mouse, P56812; bovine, Q2HJH9; and human, O14737) and CCT1 (UniProt: *H. volcanii*, O30561; *S. pombe*, O94501; *C. elegans*, P41988; mouse, P11983; bovine, Q32L40; and human, P17987) were performed with ClustalO in Jalview[62]. The sequence conservation score of PDCD5 was calculated using the ConSurf server[63].

### Spatial analysis of TRiC in situ

The distance and angle examination of TRiC was performed similarly to as in previous studies[22,64,65]. For TRiC cluster tracing, the coordinates of TRiC determined by subtomogram averaging were used to localize the particles in the tomograms. The TRiC cluster (containing ≥2 TRiC particles) was defined by the distance between the coordinates of one TRiC and that of its nearest neighbour using a distance cut-off of 20 nm (centre-to-centre distance). As the coordinate represents the centre of the structure, the rotation of the particles would not affect the distance measurement. The particle closest to the previous particle in terms of Euclidean distance was selected as the trailing TRiC within the cluster, provided that it fell within the permitted distance threshold. Various distance thresholds ranging from 15 nm (the minimum centre-to-centre distance between two TRiC) to 40 nm were investigated (Fig. 4b,c). For each specific distance, the threshold was confined within a range of ±0.5 nm (for example, for 17 nm, the permissible distance ranged from 16.5 nm to 17.5 nm). A distance threshold of 20 nm was used to define whether TRiC belongs to the same cluster in this study.

For the distance of TRiC pair analysis in Extended Data Fig. 9h,i, the number and the mean ± s.d. values were $n_2$ (cluster length = 2) = 326

$(17.35 \pm 1.18)$; $n_3 = 218$ $(17.44 \pm 1.27)$, $n_4 = 74$ $(17.01 \pm 1.17)$, $n_5 = 35$ $(17.05 \pm 1.16)$, $n_6 = 16$ $(16.87 \pm 1.01)$ and $n_7 = 4$ $(17.33 \pm 0.89)$, respectively, in the untreated dataset. The number and the mean ± s.d. were $n_2 = 195$ $(17.04 \pm 1.28)$, $n_3 = 116$ $(17.42 \pm 1.18)$, $n_4 = 27$ $(16.87 \pm 0.96)$, $n_5 = 7$ $(17.09 \pm 1.25)$ and $n_6 = 4$ $(16.65 \pm 1.88)$, respectively, in the treated dataset. TRiC pairs with distances between 15 and 20 nm were analysed.

The angle between TRiC and its closest neighbouring TRiC was investigated for particles within clusters in the untreated dataset (Extended Data Fig. 8d). The divided area of the hemisphere contains all points denoting cone rotation, described by Euler angles $\theta$ and $\psi$, of a vector (0, 0, 1). These rotations are projected onto the northern hemisphere (for vectors rotated with a $z$-coordinate greater than 0) and the southern hemisphere (for vectors rotated with a $z$-coordinate less than or equal to 0) using stereographic projection. The north pole corresponds to zero rotation, signifying a vector (0, 0, 1). The rotations of the neighbour TRiC were multiplied by the inverse rotations of the respective neighbour particles.

To calculate the percentage of TRiC clusters with neighbouring actin filaments. The particles from the subtomogram averaging of TRiC and actin filaments were mapped back to tomograms for analysis. The threshold of the neighbouring distance (TRiC centre to the centre of actin dimer) was set to 20 nm.

### Spatial relation between ribosomes and TRiC in cells

The spatial distribution of TRiC near the ribosome exit tunnel was investigated. The coordinates of ribosome, 60S and 40S determined by subtomogram averaging were used to localize the particles in the tomograms[22]. The ribosome was rotated to a reference position (zero rotation) through an inverse rotation, which means it was rotated by $(-\psi, -\theta, -\varphi)_{ribosome}$. Subsequently, TRiC underwent rotation by its respective angles $(\varphi, \theta, \psi)_{TRiC}$, followed by another rotation of $(-\psi, -\theta, -\varphi)_{ribosome}$, therefore aligning the ribosome–TRiC within a standard rotation frame (zero rotation of the ribosome), while maintaining their original angular relationship. The coordinates of the ribosome exit tunnel were subtracted from both the ribosome exit tunnel coordinates (setting it to zero) and TRiC coordinates. The new TRiC coordinates were rotated by $(-\psi, -\theta, -\varphi)_{ribosome}$ to illustrate their positioning relative to the zero rotation of the ribosome. For the spatial analysis of ribosome and TRiC, ribosome particles were more abundant than TRiC particles. As a result, the same TRiC can be the nearest neighbour of several ribosomes. Our analysis focused on the ribosomes that acted as the nearest neighbours of TRiC. The mean ± s.d. in Extended Data Fig. 9c,k were as follows: untreated open TRiC in the ribosome ETS (55.1 ± 0.8%); untreated closed TRiC in the ETS (55.3 ± 0.3%); untreated open TRiC in the non-ETS (44.9 ± 0.8%); untreated closed TRiC in the non-ETS (44.7 ± 0.3%); treated open TRiC in the ETS (50.4 ± 0.4%); treated closed TRiC in the ETS (49.7 ± 1.0%); treated open TRiC in the non-ETS (49.6 ± 0.4%); and treated closed TRiC in the non-ETS (50.3 ± 1.0%). Data plotting and statistical analysis were performed using GraphPad Prism (v.10, GraphPad Software).

### Reporting summary

Further information on research design is available in the Nature Portfolio Reporting Summary linked to this article.

## Data availability

Cryo-ET maps have been deposited at the Electron Microscopy Data Bank (EMDB) under accession numbers EMD-18921 (open TRiC in untreated and HHT-treated cells), EMD-18913 (closed TRiC in untreated and HHT-treated cells, $C_1$ symmetry), EMD-18914 (closed TRiC in untreated and HHT-treated cells, $D_8$ symmetry), EMD-18922 (open TRiC in untreated cells), EMD-18923 (open TRiC without PFD in untreated cells), EMD-18924 (open TRiC with one PFD in untreated cells), EMD-18925 (open TRiC with two PFDs in untreated cells), EMD-18926 (closed

TRiC in untreated cells, $C_1$ symmetry), EMD-18927 (closed TRiC in untreated cells, $D_8$ symmetry), EMD-18928 (closed TRiC-class 1 in untreated cells), EMD-18929 (closed TRiC-class 2 in untreated cells), EMD-18930 (closed TRiC-class 3 in untreated cells), EMD-18931 (open TRiC in HHT-treated cells), EMD-18932 (open TRiC without PFD in HHT-treated cells), EMD-18933 (open TRiC with one PFD in HHT-treated cells), EMD-18934 (open TRiC with two PFDs in HHT-treated cells), EMD-18936 (closed TRiC in HHT-treated cells, $C_1$ symmetry), EMD-18937 (closed TRiC in HHT-treated cells, $D_8$ symmetry), EMD-18938 (closed TRiC-class 1 in HHT-treated cells), EMD-18939 (closed TRiC-class 2 in HHT-treated cells) and EMD-18940 (closed TRiC-class 3 in HHT-treated cells). Maps and atomic models used from previous studies were downloaded from the EMDB (EMD-12606, EMD-12607 and EMD-40461) and the PDB (2K6B, 7X3J, 7NVN, 7NVO, 7NVL, 7NVM, 8F8P and 7WU7). The model of PDCD5 was from the AlphaFold Protein Structure Database (AF-O14737-F1). The *Saccharomyces* Genome Database is available at https://www.yeastgenome.org/. Protein sequences were from UniProt: CCT1–CCT8 (P17987, P78371, P49368, P50991, P48643, P40227, Q99832 and P50990), PDCD5 (*M. maripaludis*, A9A8D7; *S. pombe*, O13929; *C. elegans*, Q93408; mouse, P56812; bovine, Q2HJH9; human, O14737) and CCT1 (*H. volcanii*, O30561; *S. pombe*, O94501; *C. elegans*, P41988; mouse, P11983; bovine, Q32L40; human, P17987). Source data are provided with this paper.

## Code availability

The contextual analysis tool for cryo-ET is available at GitHub (https://github.com/turonova/cryoCAT).

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

**Acknowledgements** We thank the members from the Department of Molecular Sociology and the Central Electron Microscopy facility at the Max Planck Institute of Biophysics, E. Schuman, A. Schwarz, K. H. Bui and the staff at the Max Planck Computing and Data Facility for support and discussions. The work was supported by the Max Planck Society, the Hessian Ministry of Higher Education, Research, Science and the Arts (HMWK) as part of the cluster project EnABLE and the Chan Zuckerberg Initiative for Visual Proteomics Imaging (grant 2021-234666).

**Author contributions** H.X., J.F. and M.B. conceptualized the project. H.X. prepared the cryo-ET samples, collected data, performed the image processing, analysed data and performed AlphaFold-Multimer. P.R.-A. cloned and purified recombinant WT and mutant PDCD5. P.R.-A. and T.-T.L. performed, with input from J.F., native gel analysis of the affinity and kinetics of PDCD5 binding to TRiC. P.R.-A. performed the ATPase assays with input from J.F. Using native gels, T.-T.L. analysed the effect of knocking down *PDCD5*, and the binding of PDCD5 to TRiC from cell lysates, with input from P.R.-A. and J.F.; H.X. and R.R.E.R. performed the cell culture and co-IP experiments. H.X., T.M. and B.T. were involved in the spatial analysis of TRiC. H.X., S.B., P.R.-A., J.F. and M.B. prepared figures and wrote the manuscript with input from all of the authors.

**Funding** Open access funding provided by Max Planck Society.

**Competing interests** The authors declare no competing interests.

**Additional information**
**Correspondence and requests for materials** should be addressed to Judith Frydman or Martin Beck.

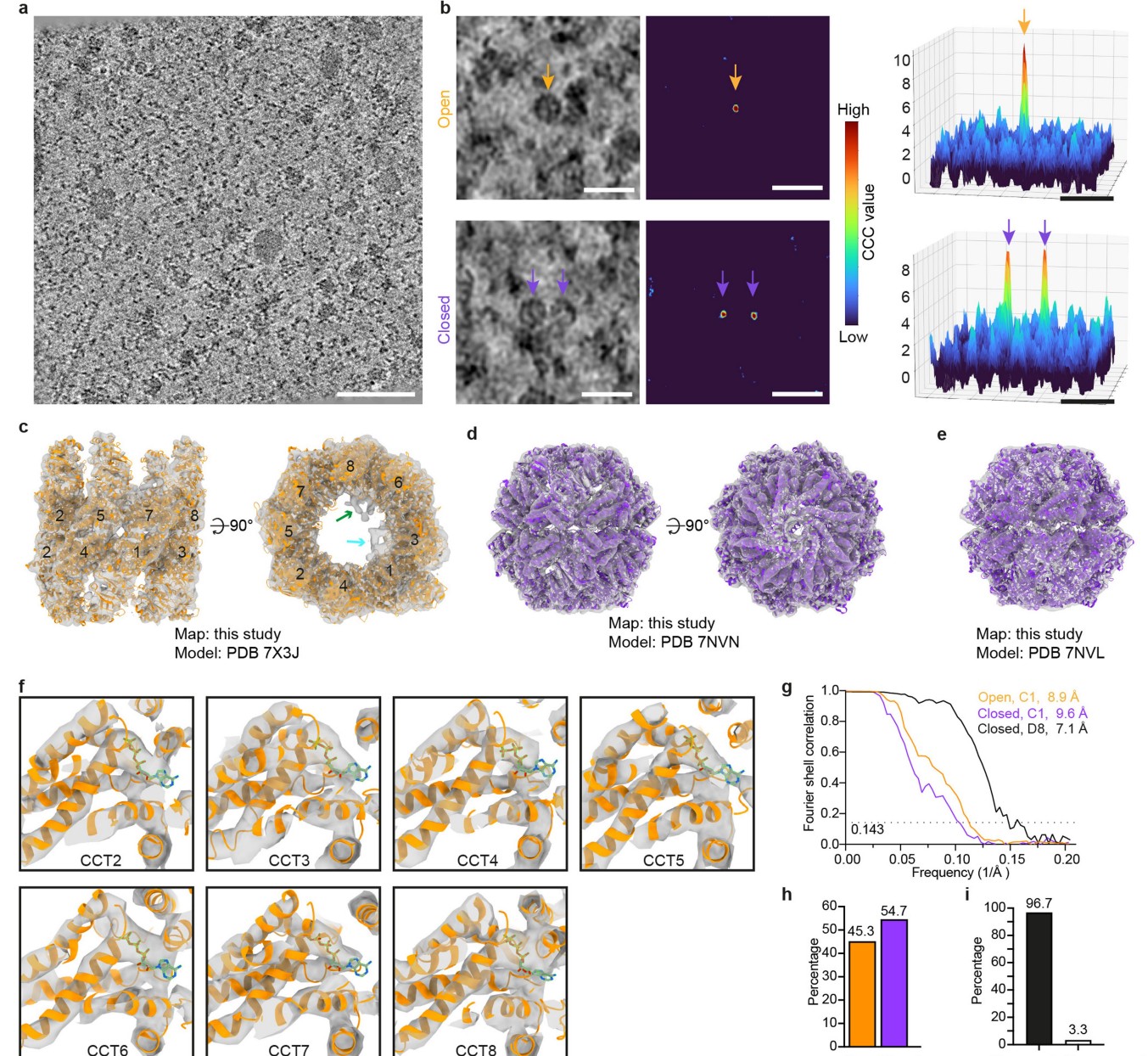

**Extended Data Fig. 1 | Open and closed TRiC structures in human cells.**
**a**, A tomogram slice from an untreated cell. Scale bar, 100 nm. The micrograph
is representative of 360 similar tomograms. **b**, TRiC densities in the tomogram
slices and corresponding template matching peaks colour-coded by constrained
cross-correlation (CCC). Scale bar, 30 nm. Potential TRiC particles were
extracted based on the CCC value from template matching. The micrograph
is representative of 360 similar tomograms. **c**, Open TRiC map (C1 symmetry)
fitted with PDB 7X3J. Arrows point to the additional density: ring 1 (green);
ring 2 (cyan). **d,e**, Closed TRiC maps with D8 symmetry (**d**) and C1 symmetry (**e**)
were fitted with atomic models from PDB. **f**, The open TRiC map fitted with

PDB 7X3J. The resolution does not allow us to assign ATP or ADP to the map but
secondary structure is resolved. **g**, Fourier shell correlation (FSC) curves of
TRiC maps (**c**-**e**). **h**, Percentages of open and closed TRiC in cells growing in the
normal condition. **i**, Evaluation of processed and unprocessed potential TRiC
particles in this study. Particle coordinates from the subtomogram averaging
were mapped back into the tomogram, and the TRiC-like (based on size and
shape) particles that were not processed were manually counted in twenty
tomograms. In total, 17 TRiC-like particles were not processed, while 515
particles were processed.

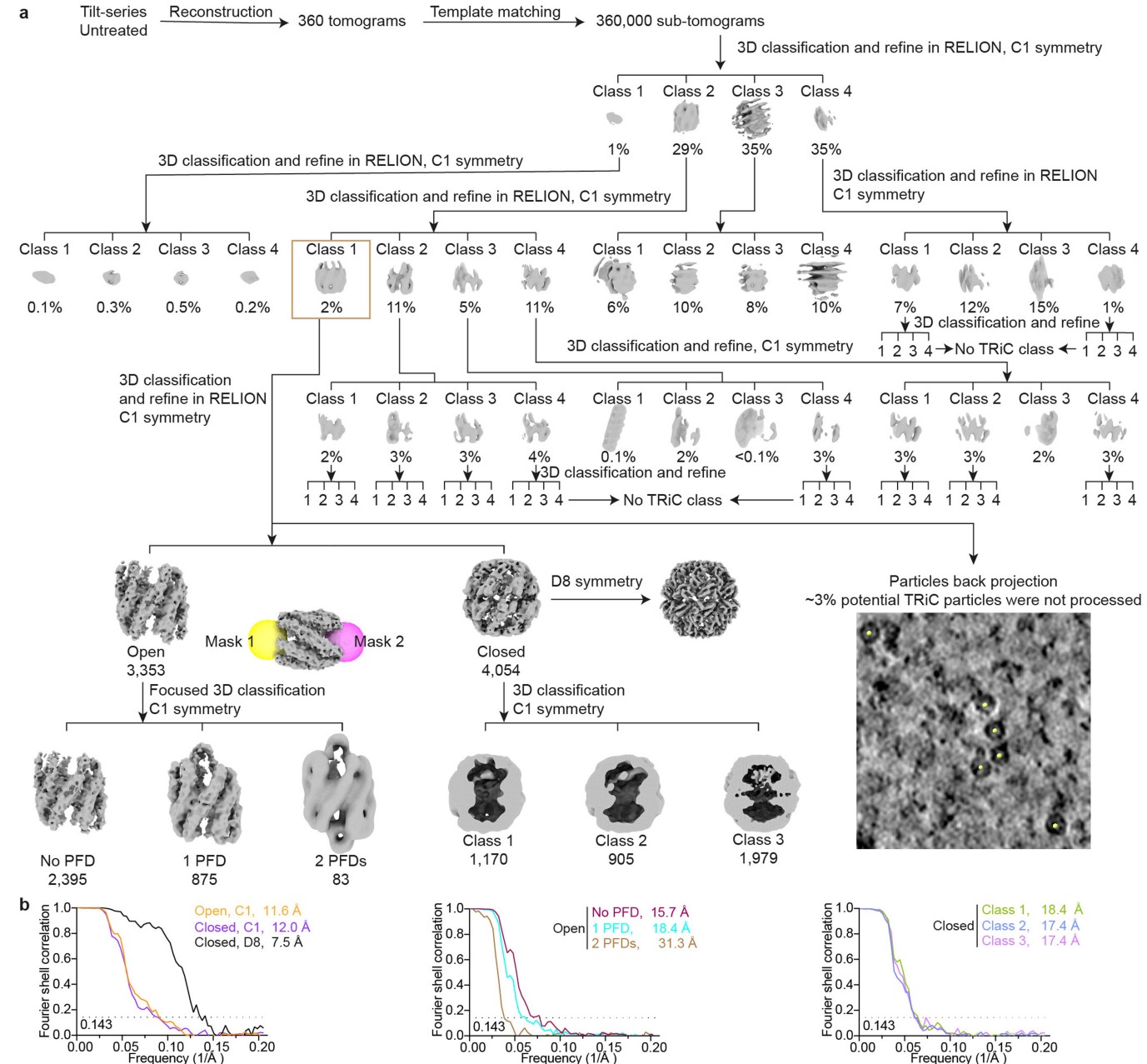

**Extended Data Fig. 2 | Data processing workflow of TRiC in untreated cells.**
**a**, Diagrams of TRiC image-processing in the untreated dataset. Tomograms were reconstructed with IMOD at bin4. Initial TRiC candidates were generated through template matching using STOPGAP. Subtomogram extraction was carried out in Warp. 3D classifications were executed to remove false positive particles and identify TRiC particles in RELION 3.1 (Methods). TRiC particles were mapped back into the tomogram for assessing the workflow (Extended Data Fig. 1i). Further classification and refinement allowed us to determine different open and closed TRiC states. ~2% TRiC particles were highlighted in brown square and all percentages in the workflow were shown as the percent of 360,000 initial particles for clarity. **b**, FSC curves of corresponding TRiC states and the resolution were displayed (FSC = 0.143).

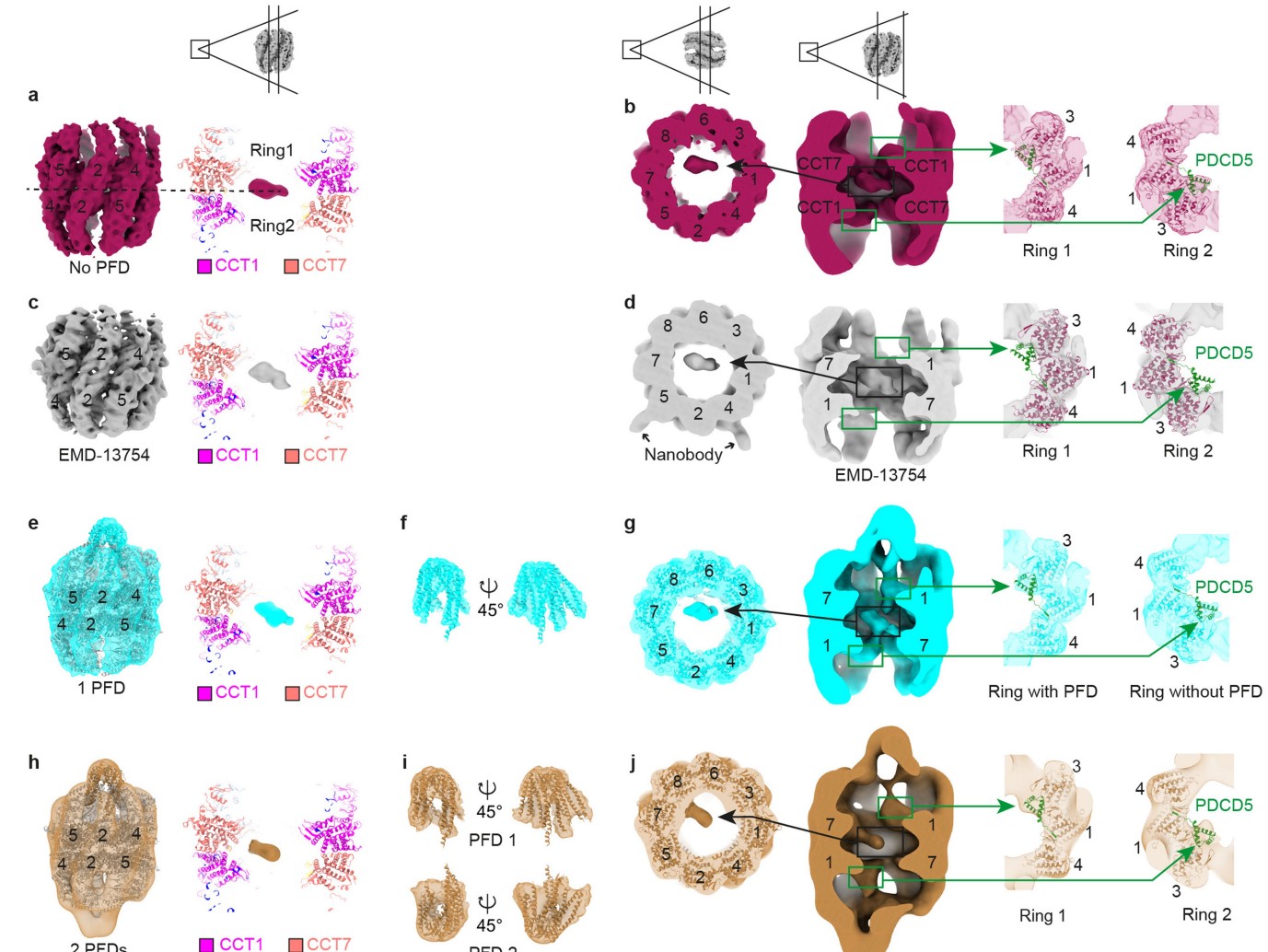

**Extended Data Fig. 3 | TRiC states fitted with atomic models in untreated cells. a**, The cryo-ET map of open TRiC without PFD bound in untreated cells. The densities within the TRiC cavity were Gaussian-filtered (sDev = 4). **b**, Different views of the map in (**a**) showing both PDCD5 and potential substrate densities. The predicted model of PDCD5-CCT3-CCT1-CCT4 was fitted into the map in (**a**). **c**, The open TRiC structure from EMDB (EMD-13754). The map was Gaussian-filtered (sDev = 4) for visualization. **d**, Different views of EMD-13754 showing potential substrate densities but no PDCD5 density. The AlphaFold-Multimer predicted model of PDCD5-CCT3-CCT1-CCT4 was fitted into the open TRiC

map in (**c**). **e**, Atomic model from PDB 7WU7 was fitted into the open TRiC structure associated with 1 PFD. The chamber densities were Gaussian-filtered (sDev = 4). **f**, PFD (PDB 7WU7) was fitted into the corresponding densities segmented from (**e**). **g**, Different views of the map in (**e**). The predicted model of PDCD5-CCT3-1-4 was fitted into the structure in (**e**). **h**, The open TRiC bound with 2 PFDs was fitted with PDB 7WU7. **i**, PFD densities segmented from (**h**) were fitted with PDB 7WU7. **j**, Different views of the map in (**h**). The predicted model of PDCD5-CCT3-CCT1-CCT4 was fitted into the map in (**h**).

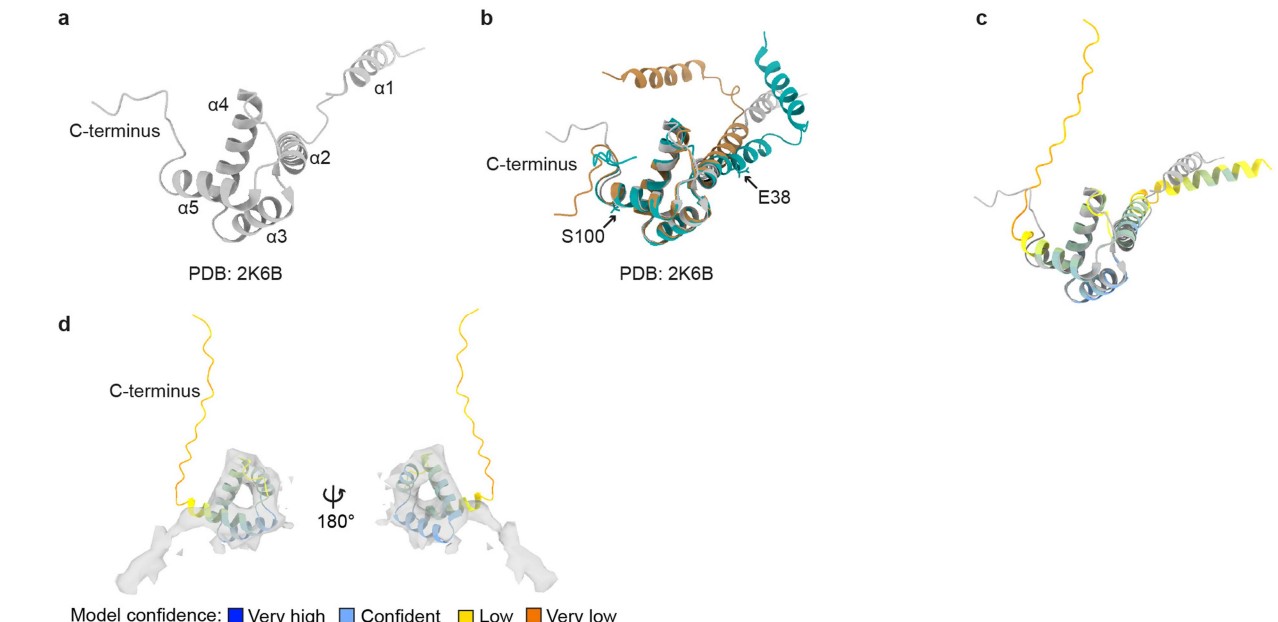

Model confidence: ■ Very high ■ Confident ■ Low ■ Very low

**Extended Data Fig. 4 | PDCD5 structures and densities of the open TRiC map. a**, NMR structure of PDCD5 (1–112 a.a.). The five helices of PDCD 5 were labelled as α1 to α5. **b**, Overlay of PDCD5 at different conformations (PDB 2K6B). **c**, Overlay of the AlphaFold predicted model of PDCD5 (PDCD5, AF-O14737-F1, full length, 1–125 a.a.) with PDB 2K6B coloured in grey. AlphaFold produces a per-residue model confidence score (predicted local distance difference test, pLDDT) between 0 and 100. Very high (blue, pLDDT > 90), High (light blue, 90 > pLDDT > 70), Low (yellow, 70 > pLDDT > 50), Very low (orange, pLDDT < 50). **d**, The additional density that was not modelled by in vitro studies of TRiC. The density was fitted with the AlphaFold predicted model of PDCD5.

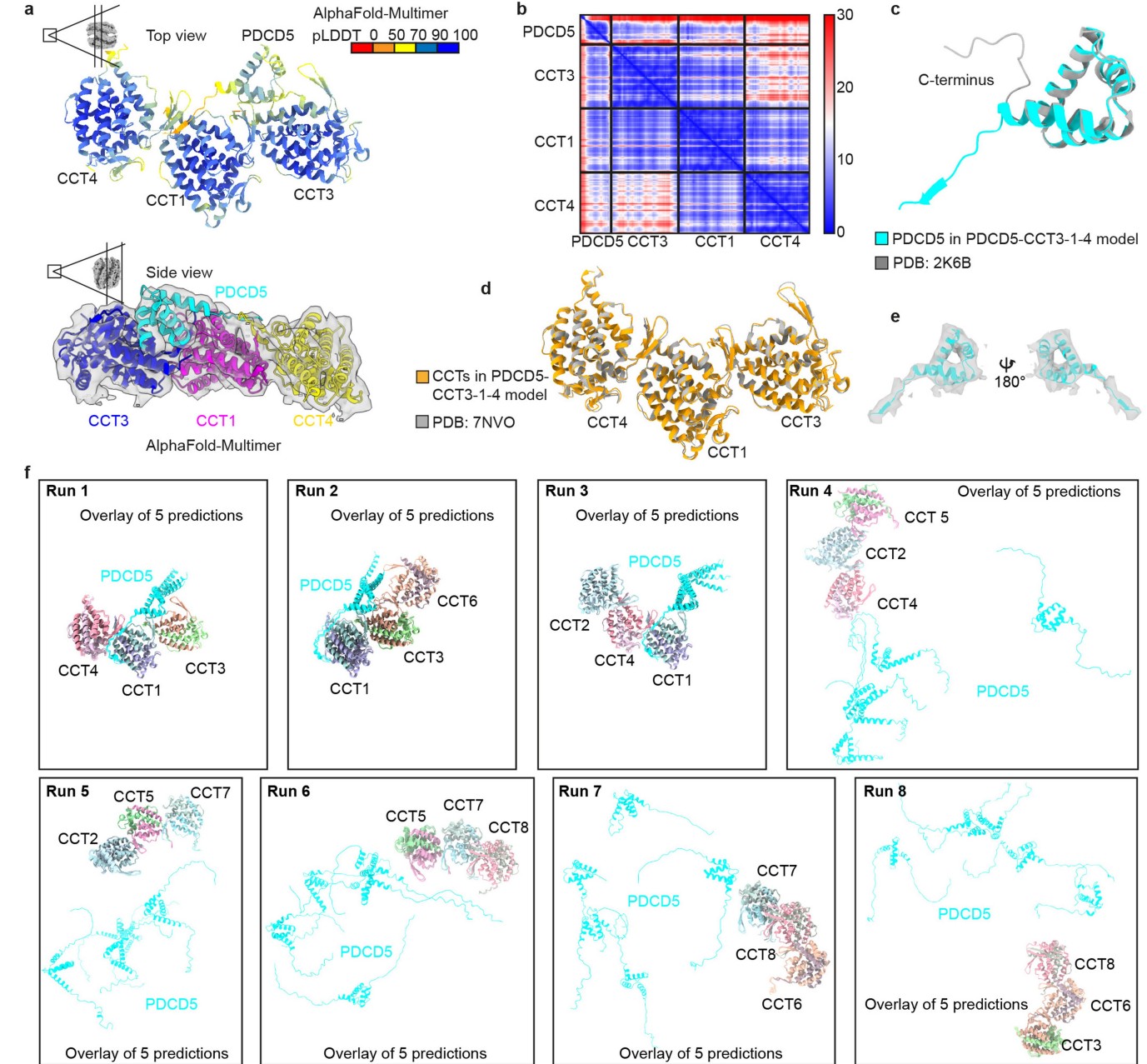

**Extended Data Fig. 5 | AlphaFold-Multimer predicted models of PDCD5 with CCTs. a**, Structure of PDCD5 (38–112 a.a.) in complex with the equatorial domain of CCT3-CCT1-CCT4 predicted by AlphaFold-Multimer. The predicted model was coloured by per-residue confidence score (pLDDT). The predicted model was fitted into the open TRiC map shown in the side view. **b**, The predicted alignment error (PAE) plot for the model in (**a**). **c**, Structural overlay of the predicted PDCD5 structure in (a) with the experimentally determined structure (PDB 2K68). **d**, Structural overlay of predicted CCT3, CCT1 and CCT4 with experimental structure (PDB 7NVO). **e**, PDCD5 (38-112 a.a.) structure predicted in (a) was fitted into the segmented density in the open TRiC map. **f**, AlphaFold-Multimer prediction of PDCD5 with different combinations of CCT. Five predicted models of each combination were overlayed to evaluate the consistency.

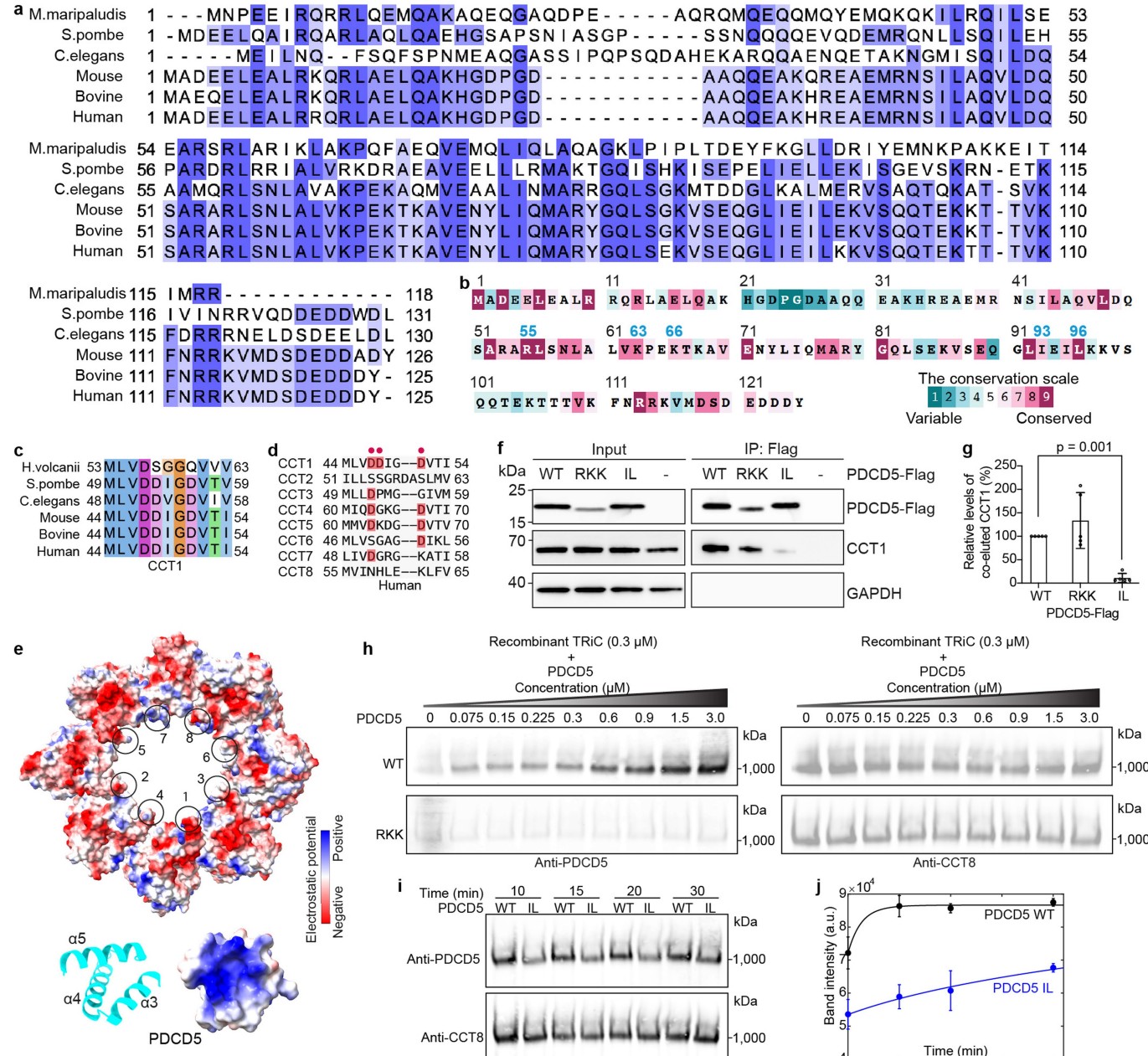

**Extended Data Fig. 6 | Sequence alignments of CCTs and PDCD5. a**, Sequence alignments of PDCD5 from various organisms using ClustalO in Jalview. **b**, Evolutionary conservation analysis of PDCD5 using ConSurf Web Server. **c**, Sequence alignments of the stem loops of CCT1 from various organisms. **d**, Sequence alignments of the stem loops of human CCT1-CCT8. **e**, Open TRiC (PDB 7NVO) and PDCD5 (helices α3 to α5, residues Q50 to T103) was shown as surface coloured by electrostatic potential. Black circles highlighted the stem loops of CCT1-CCT8. **f**, Binding of PDCD5 to TRiC was measured by co-immunoprecipitation (co-IP) from HEK 293F cells transfected with PDCD5-Flag constructs (WT: wild-type; IL: I93G, L96G; RKK: R55A, K63A, K66A). The experiment was repeated independently five times with similar results. **g**, Quantification of co-eluted CCT1 with PDCD5-Flag in (**f**). The percentage was calculated with $IP_{CCT1}/IP_{PDCD5}$ and normalized to WT (set at 100%). The data

represent the mean ± SD of five biologically independent experiments. Statistical analysis was performed using two-tailed unpaired t-tests. Significantly different (P < 0.05). **h**, Native gels show western blots analysis of the binding of increasing amounts of recombinant WT PDCD5 (top) and RKK PDCD5 (bottom) to 300 nM TRiC. Interaction was measured after 20 min of interaction at 25 °C in buffer (50 mM Tris pH 7.4, 100 mM KCl, 5mM MgCl₂, 10% glycerol, and 1mM TCEP). left panel: anti-PDCD5, right panel: anti-CCT8. The experiment was repeated independently two times with similar results. **i**, Native gels show the binding of PDCD5 WT or IL (3 µM) to TRiC (300 nM). The interaction was measured in ATPase buffer at 25 °C after 10 to 30 min. **j**, Quantification of the band intensity in (**i**). n = 2 biologically independent experiments. Data are presented as mean values ± standard error of the mean (SEM).

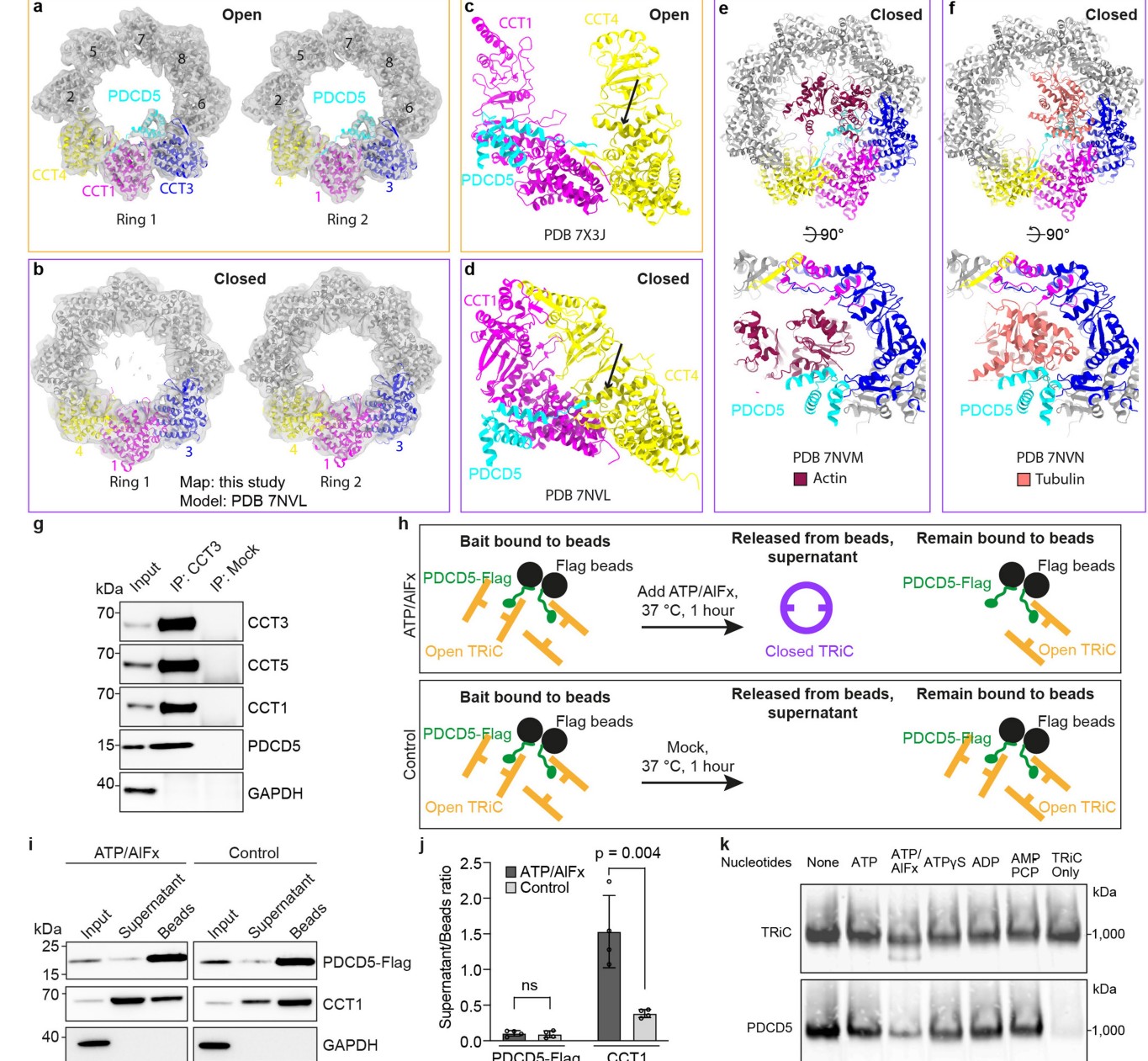

**Extended Data Fig. 7 | PDCD5 was not associated with closed TRiC. a**, The atomic model of PDCD5-CCT3-CCT1-CCT4 (AlphaFold-Multimer) was fitted into two rings of the open TRiC map (C1 symmetry). PDB 7NVO, grey. **b**, The closed TRiC map (C1 symmetry) was fitted with equatorial domains of closed TRiC (PDB 7NVL). **c,d**, PDCD5-CCT3-1-4 (AlphaFold-Multimer) was superimposed with open TRiC (PDB 7X3J) and closed TRiC (PDB 7NVL). The black arrow indicates a potential clash between the C-terminus of PDCD5 and the helix in CCT4 (N394 to V416) in the closed TRiC but not in the open TRiC. **e,f**, PDCD5-CCT3-1-4 (AlphaFold-Multimer) was overlaid with closed TRiC associated with actin (PDB 7NVM) and tubulin (PDB 7NVN). **g**, CCT3 antibody (rabbit) pulldown endogenous PDCD5 in HEK293F cells. Rabbit IgG (mock) as a control. **h**, The schematic of induction of TRiC closure during co-IP in two conditions. **i**, As illustrated in (**h**), beads bound with TRiC-PDCD5-Flag (from co-IP) were incubated in buffer with ATP/AlFx or without ATP/AlFx (control) for 1 h at 37 °C.

The supernatant (containing released TRiC and PDCD5) and beads (bound with TRiC-PDCD5-Flag) were detected by western blotting. The experiment was repeated independently four times with similar results. **j**, The ratio of PDCD5 (left two columns) in supernatant compared to PDCD5 remaining bound to beads after ATP/AlFx incubation in (**i**). The ratio of TRiC (right two columns) in supernatant compared to TRiC bound to beads after ATP/AlFx incubation in (**i**). The data represent the mean ± SD of four biologically independent experiments. Statistical analysis was performed using two-tailed unpaired t-tests. Significantly different (P < 0.05). ns, not significant. (**k**) Native gels of the interaction of 300 nM TRiC to 3 µM WT PDCD5 in buffer containing 1 mM of different ATP analogues, which induce different TRiC conformational states, analysed by immunoblotting with anti PDCD5 or CCT8 antibodies. The experiment was repeated two times with similar results.

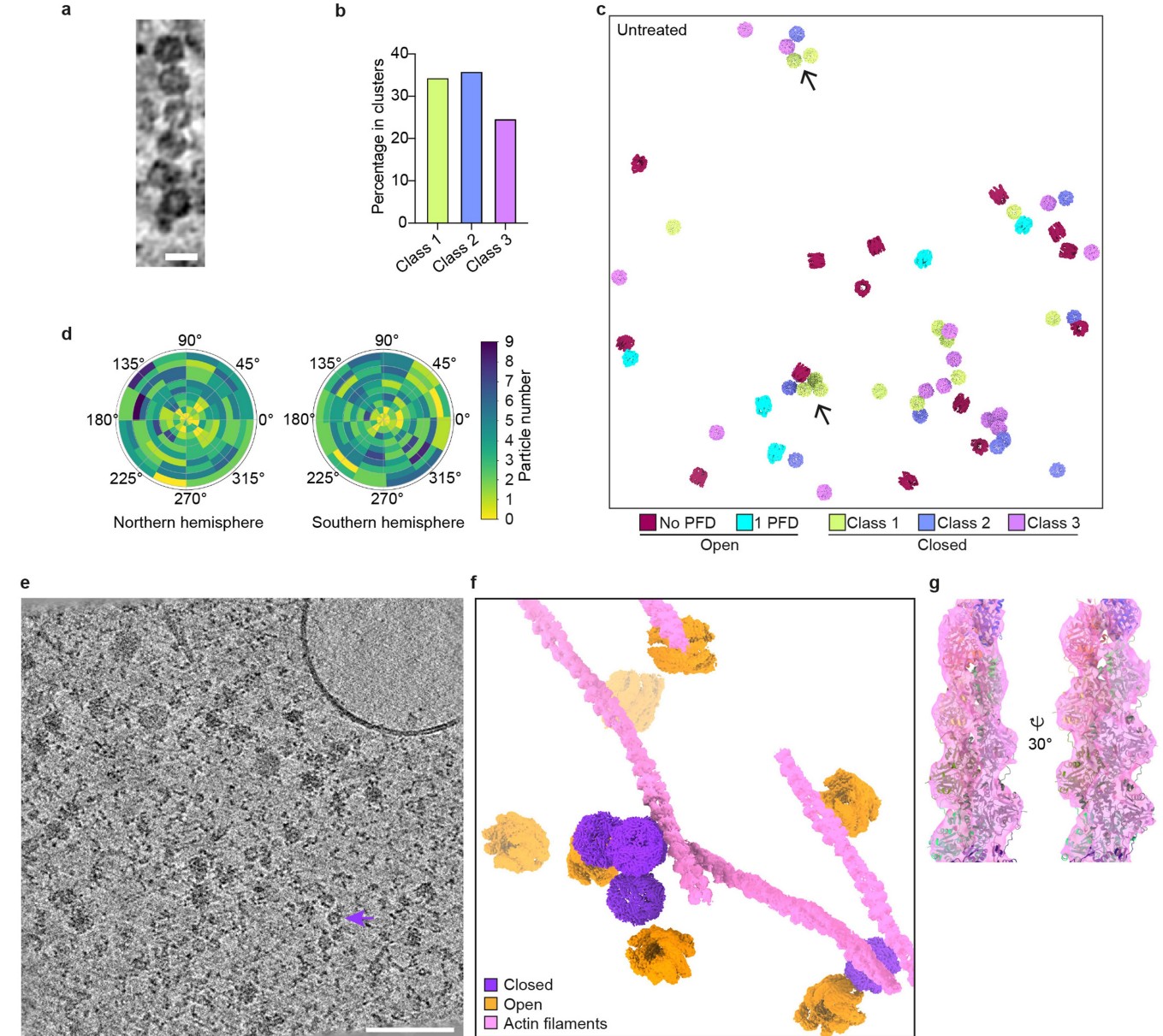

**Extended Data Fig. 8 | Analysis of TRiC neighbours in situ. a**, A tomographic slice showing a cluster of closed TRiC particles. Scale bar, 20 nm. The micrograph is representative of 360 tomograms. **b**, Percentages of class 1 to 3 of closed TRiC within clusters. **c**, TRiC states in a tomogram from an untreated cell. The black arrow highlights different classes of closed TRiC within the cluster. **d**, Angular distribution of the closed TRiC relative to its nearest neighbour in the clusters (20 nm threshold). The divided area represents particles with cone rotation of vector (0, 0, 1), projected on the northern hemisphere for vectors with z coordinate > 0 and the southern hemisphere for vectors with z coordinate <= 0 using stereographic projection. **e**, A tomographic slice from an untreated cell. Scale bar, 100 nm. The purple arrow indicates a TRiC cluster. The micrograph is representative of 360 tomograms. **f**, Snapshots of TRiC and actin filaments. Actin filaments and TRiC were mapped back to the tomogram after subtomogram averaging. **g**, The subtomogram averaging map of actin filament fitted with atomic model (PDB 8F8P).

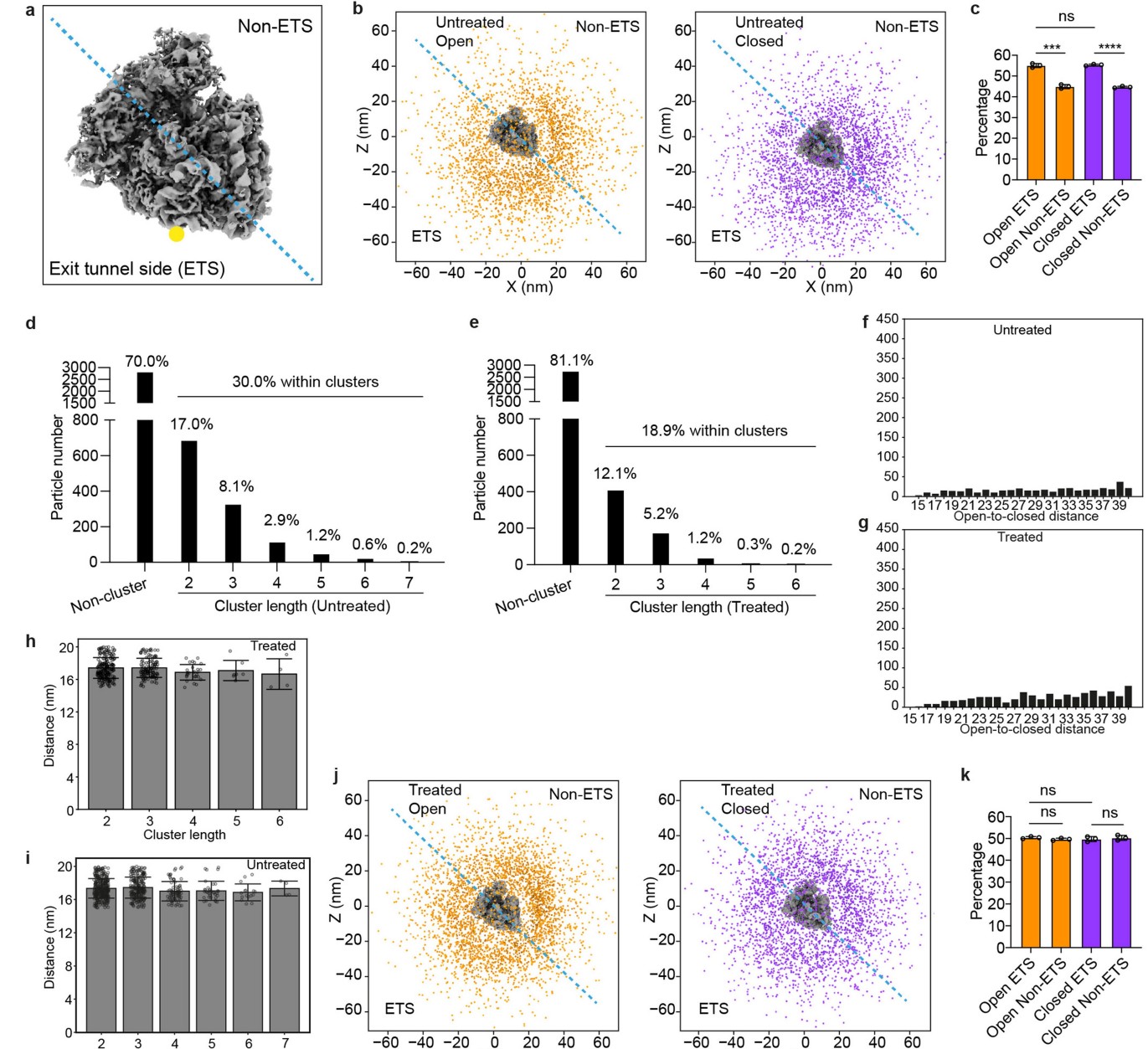

**Extended Data Fig. 9 | TRiC clusters analysis in untreated and treated datasets. a**, Illustration of ribosome exit tunnel side (ETS) and non-ETS for the analysis in (**b**,**j**). The position of exit tunnel was highlighted by a yellow dot. **b**, Distribution of the nearest neighbouring TRiC of ribosomes within 70 nm (TRiC centre to ribosome centre) in untreated cells. **c**,**k**, Percentages of open and closed TRiC in the ETS of ribosomes. The data represents mean ± SD of three independent data collection sessions. 2,716 open and 2,900 closed TRiC in untreated cells (**c**). 3,453 open TRiC and 2,924 closed TRiC in treated cells (**k**). Statistical analysis was conducted using two-tailed unpaired t-tests. ns, not significant (p > 0.05); ***P = 0.0002; ****P < 0.0001. **d**,**e**, The abundance of the closed TRiC within clusters and non-clusters in untreated (**d**) and HHT-treated cells (**e**). The percentage of TRiC in cluster was calculated as closed-TRiC in

clusters (1,215 particles for untreated, 647 for treated) divided by all closed TRiC (4,054 for untreated, 3,418 for treated). The cluster length means the particle number per TRiC cluster. **f**,**g**, The number of open TRiC with neighbouring closed TRiC at the indicated distance in untreated (**f**) and treated (**g**) cells. Open TRiC particles with neighbouring closed TRiC at various distances (15 to 40 nm) were counted. **h**,**i**, Centre-to-centre distance of TRiC pairs in clusters of varying lengths. The numbers (N) are N2 (cluster length = 2) = 326 particles, N3 = 218 particles, N4 = 74 particles, N5 = 35 particles, N6 = 16 particles and N7 = 4 particles in untreated cells, and N2 = 195 particles, N3 = 116 particles, N4 = 27 particles, N5 = 7 particles and N6 = 4 particles in treated cells. The data represent the mean ± SD (Methods). **j**, The analysis of ribosomes and TRiC in HHT-treated cells, similar to (**b**).

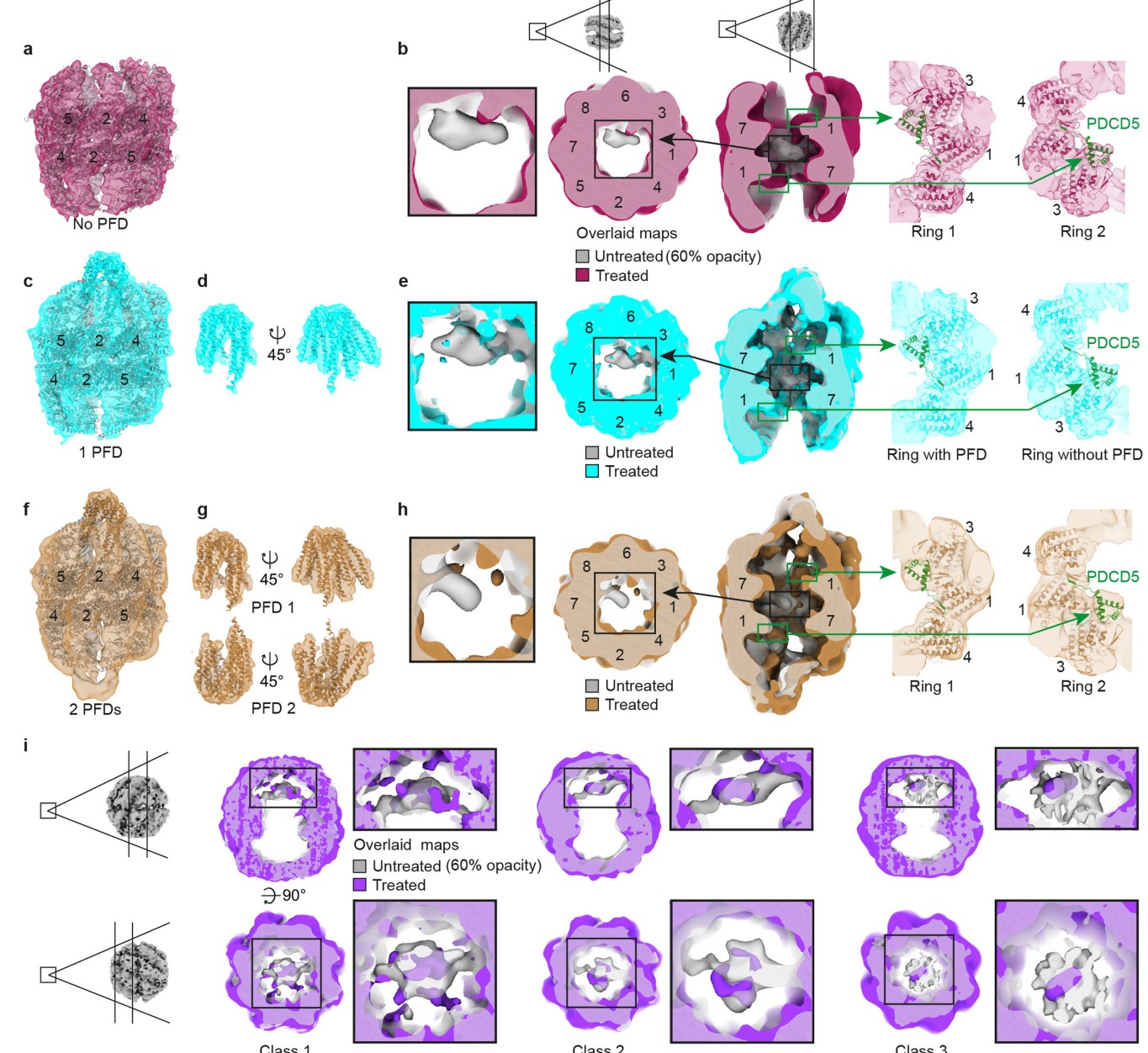

**Extended Data Fig. 10 | Atomic models were fitted into TRiC structures in HHT-treated cells. a**, Cryo-ET map of open TRiC without PFD, fitted with PDB 7X3J in treated cells. **b**, Overlaid maps (at similar contour level) of untreated and treated TRiC without PFD. The opacity of the untreated map was set at 60%. The predicted model of PDCD5-CCT3-CCT1-CCT4 was fitted into the map in (**a**). **c**, PDB 7WU7 was fitted into the open TRiC structure bound with 1 PFD. **d**, PDB 7WU7 was fitted into the PFD density segmented from (**c**). **e**, Overlaid maps (at similar contour level) of untreated and treated TRiC with one PFD. The densities inside the TRiC chamber were Gaussian-filtered (sDev = 4) in (**b**) and (**e**) for visualization. The predicted model of PDCD5-CCT3-1-4 was fitted into the map in (**c**). **f**, The open TRiC structure bound with 2 PFDs was fitted with PDB 7WU7. **g**, PFD densities segmented from (**f**) were fitted with PDB 7WU7. **h**, Overlaid maps (at similar contour level) of untreated and treated TRiC with two PFDs. The predicted model of PDCD5-CCT3-CCT1-CCT4 was fitted into the map in (**f**). **i**, Overlaid maps (at similar contour level) of untreated and treated TRiC at the closed conformation. The opacity of untreated maps was set at 60%.

Judith Frydman

# Reporting Summary

## Statistics

For all statistical analyses, confirm that the following items are present in the figure legend, table legend, main text, or Methods section.

| n/a | Confirmed | |
|---|---|---|
| ☐ | ☒ | The exact sample size (*n*) for each experimental group/condition, given as a discrete number and unit of measurement |
| ☐ | ☒ | A statement on whether measurements were taken from distinct samples or whether the same sample was measured repeatedly |
| ☐ | ☒ | The statistical test(s) used AND whether they are one- or two-sided *Only common tests should be described solely by name; describe more complex techniques in the Methods section.* |
| ☒ | ☐ | A description of all covariates tested |
| ☒ | ☐ | A description of any assumptions or corrections, such as tests of normality and adjustment for multiple comparisons |
| ☐ | ☒ | A full description of the statistical parameters including central tendency (e.g. means) or other basic estimates (e.g. regression coefficient) AND variation (e.g. standard deviation) or associated estimates of uncertainty (e.g. confidence intervals) |
| ☐ | ☒ | For null hypothesis testing, the test statistic (e.g. *F*, *t*, *r*) with confidence intervals, effect sizes, degrees of freedom and *P* value noted *Give P values as exact values whenever suitable.* |
| ☒ | ☐ | For Bayesian analysis, information on the choice of priors and Markov chain Monte Carlo settings |
| ☒ | ☐ | For hierarchical and complex designs, identification of the appropriate level for tests and full reporting of outcomes |
| ☒ | ☐ | Estimates of effect sizes (e.g. Cohen's *d*, Pearson's *r*), indicating how they were calculated |

*Our web collection on statistics for biologists contains articles on many of the points above.*

## Software and code

Policy information about availability of computer code

| Data collection | Tilt series were collected using SerialEM version 4.0.1. |
|---|---|
| Data analysis | For subtomogram averaging and classification, we used IMOD 4.11.0, Warp 1.0.9 and RELION 3.1. For structure and tomogram visualization, we used UCSF ChimeraX-1.6.1 and napari (version 0.4.16). We used cryoCAT package for the spatial analysis of TRiC (https://github.com/turonova/cryoCAT). Protein structure prediction was performed with AlphaFold-Multimer 2.2.0. Data plotting and statistical analysis were performed using GraphPad Prism (version 10, GraphPad Software). Protein sequence alignment was performed with Clustal Omega (https://www.ebi.ac.uk/jdispatcher/msa/clustalo) or ClustalO in Jalview6 (Version: 2.11.4.0). |

For manuscripts utilizing custom algorithms or software that are central to the research but not yet described in published literature, software must be made available to editors and reviewers. We strongly encourage code deposition in a community repository (e.g. GitHub). See the Nature Portfolio guidelines for submitting code & software for further information.

## Data

Cryo-ET maps have been deposited to the Electron Microscopy Data Bank (EMDB) under accession numbers EMD-18921 (Open TRiC in untreated and HHT-treated cells), EMD-18913 (Closed TRiC in untreated and HHT-treated cells, C1 symmetry), EMD-18914 (Closed TRiC in untreated and HHT-treated cells, D8 symmetry), EMD-18922 (Open TRiC in untreated cells), EMD-18923 (Open TRiC without PFD in untreated cells), EMD-18924 (Open TRiC with one PFD in untreated cells), EMD-18925 (Open TRiC with two PFDs in untreated cells), EMD-18926 (Closed TRiC in untreated cells, C1 symmetry), EMD-18927 (Closed TRiC in untreated cells, D8 symmetry), EMD-18928 (Closed TRiC-class 1 in untreated cells), EMD-18929 (Closed TRiC-class 2 in untreated cells), EMD-18930 (Closed TRiC-class 3 in untreated cells), EMD-18931 (Open TRiC in HHT-treated cells), EMD-18932 (Open TRiC without PFD in HHT-treated cells), EMD-18933 (Open TRiC with one PFD in HHT-treated cells), EMD-18934 (Open TRiC with two PFDs in HHT-treated cells), EMD-18936 (Closed TRiC in HHT-treated cells, C1 symmetry), EMD-18937 (Closed TRiC in HHT-treated cells, D8 symmetry), EMD-18938 (Closed TRiC-class 1 in HHT-treated cells), EMD-18939 (Closed TRiC-class 2 in HHT-treated cells) and EMD-18940 (Closed TRiC-class 3 in HHT-treated cells). Maps and atomic models used from previous studies were downloaded from the EMDB (EMD-12606, EMD-12607 and EMD-40461) and the PDB (2K6B, 7X3J, 7NVN, 7NVO, 7NVL, 7NVM, 8F8P and 7WU7). The model of PDCD5 was from the AlphaFold Protein Structure Database (AF-O14737-F1). Saccharomyces Genome Database is available at https://www.yeastgenome.org/. Protein sequences were from UniProt: CCT1-8 (UniProt: P17987, P78371, P49368, P50991, P48643, P40227, Q99832 and P50990), PDCD5 (UniProt: M.maripaludis_A9A8D7, S.pombe_O13929, C.elegans_Q93408, Mouse_P56812, Bovine_Q2HJH9, Human_O14737) and CCT1 (UniProt: H.volcanii_O30561, S.pombe_O94501, C.elegans_P41988, Mouse_P11983, Bovine_Q32L40, Human_P17987). Source data (raw gels and blots) to Fig.2c, Extended Data Figs. 6f,h,i and 7g,i,k , and Supplementary Fig 7b,e,f are provided with this paper.

## Research involving human participants, their data, or biological material

| | |
|---|---|
| Reporting on sex and gender | N/A |
| Reporting on race, ethnicity, or other socially relevant groupings | N/A |
| Population characteristics | N/A |
| Recruitment | N/A |
| Ethics oversight | N/A |

# Field-specific reporting

Please select the one below that is the best fit for your research. If you are not sure, read the appropriate sections before making your selection.

☒ Life sciences  ☐ Behavioural & social sciences  ☐ Ecological, evolutionary & environmental sciences

# Life sciences study design

All studies must disclose on these points even when the disclosure is negative.

| | |
|---|---|
| Sample size | The cryo-ET sample sizes were not predetermined and were limited by the availability of microscopy time. These cryo-ET data were sufficient to support our conclusions based on the resolution of the cryo-ET maps. Additionally, the datasets used in this study represent one of the largest cryo-ET datasets of human cells at this pixel size. The number of images collected is indicated in Extended Data Fig. 2 and Supplementary Fig. 1. Sample sizes for the cryo-ET map obtained in this study: Structure of the open TRiC in untreated and HHT-treated cells was determined from 7138 particles. Structure of the closed TRiC- C1 symmetry in untreated and HHT-treated cells was determined from 7472 particles. Structure of the closed TRiC- D8 symmetry in untreated and HHT-treated cells was determined from 7472 particles. Structure of the open TRiC in untreated cells was determined from 3353 particles. Structure of the open TRiC without PFD in untreated cells was determined from 2395 particles. Structure of the open TRiC with one PFD in untreated cells was determined from 875 particles. Structure of the open TRiC with two PFDs in untreated cells was determined from 83 particles. Structure of the closed TRiC-C1 symmetry in untreated cells was determined from 4054 particles. Structure of the closed TRiC-D8 symmetry in untreated cells was determined from 4054 particles. Structure of the closed TRiC-class 1 in untreated cells was determined from 1170 particles. Structure of the closed TRiC-class 2 in untreated cells was determined from 905 particles. Structure of the closed TRiC-class 3 in untreated cells was determined from 1979 particles. Structure of the open TRiC in HHT-treated cells was determined from 3785 particles. Structure of the open TRiC without PFD in HHT-treated cells was determined from 2334 particles. Structure of the open TRiC with one PFD in HHT-treated cells was determined from 1287 particles. Structure |

of the open TRiC with two PFDs in HHT-treated cells was determined from 164 particles. Structure of the closed TRiC-C1 symmetry in HHT-treated cells was determined from 3418 particles. Structure of the closed TRiC-D8 symmetry in HHT-treated cells was determined from 3418 particles. Structure of the closed TRiC-class 1 in HHT-treated cells was determined from 767 particles. Structure of the closed TRiC-class 2 in HHT-treated cells was determined from 748 particles. Structure of the closed TRiC-class 3 in HHT-treated cells was determined from 1903 particles. Biochemical sample sizes were not predetermined but were determined after completing three independent replicates and evaluating statistical significance to ensure reproducibility.

**Data exclusions**  In cryo-ET image processing, the elimination of particles erroneously identified as valid peaks is achieved through 3D classification procedures. This step is a standard practice in cryo-EM image processing.

**Replication**  Cells grow on the EM grids in four independent cell culture dishes before plunge freezing for the untreated dataset. Cells grow on the EM grids in four independent cell culture dishes before plunge freezing for the HHT-treated dataset. The pipeline used in this study is reproducible.

**Randomization**  Randomization was not required because all data were used for the analysis in this study.

**Blinding**  This study focuses on structural and spatial analysis of a cellular machine, which does not involve any experiments related to blinding.

# Reporting for specific materials, systems and methods

We require information from authors about some types of materials, experimental systems and methods used in many studies. Here, indicate whether each material, system or method listed is relevant to your study. If you are not sure if a list item applies to your research, read the appropriate section before selecting a response.

## Materials & experimental systems

| n/a | Involved in the study |
|---|---|
| ☐ | ☒ Antibodies |
| ☐ | ☒ Eukaryotic cell lines |
| ☒ | ☐ Palaeontology and archaeology |
| ☒ | ☐ Animals and other organisms |
| ☒ | ☐ Clinical data |
| ☒ | ☐ Dual use research of concern |
| ☒ | ☐ Plants |

## Methods

| n/a | Involved in the study |
|---|---|
| ☒ | ☐ ChIP-seq |
| ☒ | ☐ Flow cytometry |
| ☒ | ☐ MRI-based neuroimaging |

## Antibodies

**Antibodies used**  mouse anti-FLAG M2 (Sigma-Aldrich, F1804, 1:2,000), rabbit anti-PDCD5 (abcam, ab126213, 1:1,000), rabbit anti-CCT1 (abcam, ab240903, 1:10,000), rabbit anti-CCT2 (abcam, ab92746, 1:10,000), rabbit anti-CCT3 (proteintech, 10571-1-AP, 1:30,000), rabbit anti-CCT4 (proteintech, 21524-1-AP, 1:5,000), rabbit anti-CCT5 (proteintech, 11603-1-AP, 1:3,000), rabbit anti-CCT6 (proteintech, 19793-1-AP, 1:1,000), rabbit anti-CCT7 (abcam, ab240566, 1:30,000), rabbit anti-CCT8 (proteintech, 12263-1-AP, 1:2,000), rabbit anti-GAPDH (proteintech, 10494-1-AP, 1:15,000), mouse anti-actin (Invitrogen, AM4302, 1:3,000), mouse anti-tubulin (Sigma, T5168, 1:3,000), anti-rabbit IgG (Cell signaling, 7074, 1:10,000), anti-mouse IgG + IgM (Jackson ImmunoResearch, 115-035-044, 1:10,000), rabbit anti-PDCD5 (proteintech, 12456-1-AP, 1:1,000), mouse anti-CCT8 (Santa Cruz Biotechnology, sc-377261, 1:250), rabbit anti-CCT5 (abcam, ab129016,1:10,000)

**Validation**  mouse anti-FLAG M2 (Sigma-Aldrich, F1804), https://www.sigmaaldrich.com/DE/en/product/sigma/f1804?=kr&srsltid=AfmBOoqeCslWG5-kRE-f0WnSaQYNp5wxVrHDkOJMfNEkgca97KGtzCR8
rabbit anti-PDCD5 (abcam, ab126213), https://www.abcam.com/en-us/products/primary-antibodies/pdcd5-antibody-ab126213
rabbit anti-CCT1 (abcam, ab240903), https://www.abcam.com/en-us/products/primary-antibodies/tcp1-alpha-ccta-antibody-91a-ab240903
rabbit anti-CCT2 (abcam, ab92746), https://www.abcam.com/en-us/products/primary-antibodies/cct2-antibody-epr4084-ab92746
rabbit anti-CCT3 (proteintech, 10571-1-AP), https://www.ptglab.com/products/CCT3-Antibody-10571-1-AP.htm
rabbit anti-CCT4 (proteintech, 21524-1-AP), https://www.ptglab.com/products/CCT4-Antibody-21524-1-AP.htm
rabbit anti-CCT5 (proteintech, 11603-1-AP), https://www.ptglab.com/products/CCT5-Antibody-11603-1-AP.htm
rabbit anti-CCT6 (proteintech, 19793-1-AP), https://www.ptglab.com/products/CCT6A-Specific-Antibody-19793-1-AP.htm
rabbit anti-CCT7 (abcam, ab240566), https://www.abcam.com/en-us/products/primary-antibodies/tcp1-eta-antibody-ab240566
rabbit anti-CCT8 (proteintech, 12263-1-AP), https://www.ptglab.com/products/CCT8-Antibody-12263-1-AP.htm
rabbit anti-GAPDH (proteintech, 10494-1-AP)), https://www.ptglab.com/products/GAPDH-Antibody-10494-1-AP.htm
mouse anti-actin (Invitrogen, AM4302), https://www.thermofisher.com/antibody/product/beta-Actin-Antibody-clone-AC-15-Monoclonal/AM4302
mouse anti-tubulin (Sigma, T5168), https://www.sigmaaldrich.com/DE/en/product/sigma/t5168
rabbit anti-PDCD5 (proteintech, 12456-1-AP), https://www.ptglab.com/products/PDCD5-Antibody-12456-1-AP.htm
mouse anti-CCT8 (Santa Cruz Biotechnology, sc-377261), https://www.scbt.com/p/tcp-1-theta-antibody-e-7?srsltid=AfmBOoo6lhdT8i2Hq_gFUwDuwfwbaFbPgRgnBU-9JAwbmJtuUuFCX7yL
rabbit anti-CCT5 (abcam, ab129016), https://www.abcam.com/en-us/products/primary-antibodies/tcp1-epsilon-cct5-antibody-epr7562-ab129016

# Eukaryotic cell lines

Policy information about cell lines and Sex and Gender in Research

| | |
|---|---|
| Cell line source(s) | HEK Flp-In T-Rex 293 (Invitrogen), HEK293F (Thermo Fisher), Wide-type HEK293T (abcam, ab255449), PDCD5 knockout HEK 293T cells (abcam, ab266229) |
| Authentication | No additional authentication was conducted for commercially available cell lines. |
| Mycoplasma contamination | The cells were tested negative for mycoplasma contamination. |
| Commonly misidentified lines<br>(See ICLAC register) | No commonly misidentified cell lines were used in the study. |

# Plants

| | |
|---|---|
| Seed stocks | N/A |
| Novel plant genotypes | N/A |
| Authentication | N/A |

