## [Peer Review file · Nature]

In situ analysis reveals TRiC duty cycle and PDCD5 as open state cofactor

Corresponding Author: Professor Martin Beck

Version 1:

Reviewer comments:

Referee #1

(Remarks to the Author)

Xing et al analyze previously acquired tomograms and present the first detailed structural analysis of the TRiC chaperonin in cells. This is an exciting area of study, and it is likely that much will be learnt by characterizing the function of chaperones in their native environment. The manuscript makes three main claims:

1. In cells, almost all TRiC is substrate-engaged.
2. Although the closed state of TRiC dominates in the presence of ATP in vitro, the open state is abundant in cells.
3. Open (but not closed) TRiC is almost always bound by PDCD5.

More data are required to establish the veracity of these claims, as detailed below. Furthermore, although this is conceptually a very exciting piece of work, a fundamental limitation is that it does not substantially advance our understanding of the mechanism of TRiC. Although the observations regarding PDCD5 are potentially important, very little is done to follow up on these ideas. As this aspect is still very preliminary, it is not possible to tell whether additional data establishing the mechanism of PDCD5 would result in a study of general interest.

Below are some suggestions to improve the clarity of the manuscript and more rigorously establish the main claims.

Major points

1. The authors show that ~55% of TRiC is symmetrically closed in cells and claim that this is different to the in vitro situation. Stated in line 24: "Although the closed TRiC state dominates the ATP-driven substrate folding cycle in vitro...". However, no quantitative evidence is given for this claim. What proportion of TRiC is open/closed in vitro, exactly? What is expected at different ATP concentrations, based on in vitro experiments, and how does this compare to the ATP/ADP ratio in the cytosol?
2. The "headline" claim of the manuscript is that PDCD5 is essentially constitutively bound to open TRiC. This claim requires additional support:
 - a. When describing the fitting of PDCD5 into the 'extra' density of open TRiC, the authors show the fit from the top (looking down into the TRiC chamber) but it would be helpful to show side views as well.
 - b. The authors should fit other known TRiC binders (e.g. Phosducin-like proteins) into the density they assign to PDCD5 and show the differences in fit quality (for instance with a correlation coefficient).
 - c. For the PDCD5 pulldown, show that the entire TRiC complex is pulled down, not just CCT1. It would also be good to perform the reciprocal pulldown, by immunoprecipitating TRiC and looking for PDCD5.
3. AlphaFold is used to generate models of the TRiC-PDCD5 complex, and the predicted interface is shown in detail in Fig 2. To make specific claims about the interface, the prediction must be tested by mutagenesis of PDCD5.
4. Another key claim regarding PDCD5 is that it does not bind closed TRiC. This claim is based on a pulldown experiment using flag-tagged PDCD5. However, the experimental design has several issues which interfere with interpreting the result.
 - a. The authors need to show that, in their experiment, the treatment of cell lysate with ATP/AIFx results in symmetrically closed TRiC.
 - b. As the authors state, "The absence of a PDCD5 density in the structure of closed TRiC, could in principle also be explained by PDCD5 becoming too flexible for structural detection." If PDCD5 is encapsulated in closed TRiC, it would not be accessible for Flag pulldown. How can the authors discriminate between loss of PDCD5 and encapsulation within PDCD5?
5. In the manuscript text, the authors are appropriately conservative about the possible proximity of TRiC to the ribosome

exit. In this context, Fig 3e/h is misleading, as it suggests to the reader that this arrangement of TRiC and the ribosome represents a specific complex rather than just one of many possible orientations, with no way to distinguish between actual complexes and simple stochastic proximity. I suggest removing these panels.

6. In lines 98-100 authors state that nearly all open TRiC binds substrate, and they show the binding in Fig 1 b and Extended Data Fig 3 a,c,e,h. How can the authors be confident that the density between the rings corresponds to substrate and not the N/C terminal tails of TRiC?

7. In lines 163-165, the authors state that density for both PDCD5 and substrate can be observed in the same maps. Can the authors show both entities in one structure/map? Could they comment on their assignment of that density as substrate rather than e.g. a cofactor or cochaperone?

8. Previous CryoET analyses showed that TRiC forms clusters in cells (<https://doi.org/10.1016/j.cell.2017.12.030>). This work should be cited and discussed.

Minor points

1. In lines 46, 55 and 56, the authors refer to a single 'central chamber' as well as 'two opposite chambers'. This should be rephrased for clarity.

2. When describing the interface between PDCD5 and TRiC, the authors consistently swap positively charged and negatively charged residues (e.g. l. 142, 143 and 146; Figure 2 and Extended Figure 6).

3. In lines 180 and 182, the authors refer to Extended Data Fig. 6 instead of Extended Data Fig. 7.

4. In lines 203-205, the authors state that TRiC clusters have 'a tendency to be organized in a one-dimensional fashion', either linear or circular. Could this observation be a result of the very thin lamellae produced by FIB milling? Moreover, would a circle not be two dimensional?

5. The reference to published tomograms of *Dictyostelium discoideum* (lines 215-217 and Extended Data Fig. 8i) would lend more support to the authors' claims if it included a more in-depth analysis. A single image is not the most convincing

6. In the paragraph starting in line 219, the authors could consider adding more detail regarding the approach to determine the positioning of TRiC relative to the exit tunnel.

7. In line 255, the authors state that 'the densities corresponding to the substrate position were reduced'. The word 'reduced' may be confusing as it could refer to a lower number of particles exhibiting the density, or a decreased volume of the density itself.

8. The maps of the untreated and HHT-treated cells in Extended Data Figs. 3 and 1- would benefit from being shown side-by-side or overlaid (with the same threshold) for a clearer comparison.

9. Figure 4ef show that HHT treatment leads to the complete loss of substrate specifically in PFD-free open TRiC, leading to the appearance of a species that is virtually non-existent in untreated cells. This seems like an interesting result which would benefit from being discussed in the main text.

10. Line 283. The comparison with condensates seems unnecessary – there is no reason to think that a complex/cluster of just a few molecules is related to a condensate.

11. In line 314, 'a mutually exclusively manner' should read 'a mutually exclusive manner'.

12. The electrostatic scale in Fig. 2 shows aspartate residues as positively charged.

13. In Fig. 2, the positioning of panel c (and insets) next to the maps in panels a and b gives the impression that the interface is clearly resolved in the experimental structure. This is not the case as the resolution of the reconstruction would not allow side chain building. The authors should make it clear in the visual itself that the interface is an AlphaFold prediction that has not been verified (e.g. by mutagenesis) but it consistent with the positioning of PDCD5 in the experimental map.

14. In Extended Data Fig. 6e, the electrostatic scale is likely swapped as the main text refers to lysine residues in PDCD5.

15. In Extended Data Fig. 8h, there is no need for a chart containing only one bar. The value can just be mentioned in text.

16. In Extended Fig 4e, the "top 10" genes associated with PDCD5 do not match the current DepMap analysis. This should be updated. Furthermore, can the authors comment on PDCD5 having a strong association with prefoldin genes, as well as tubulin folding cofactors TBCA and TBCB?

17. In line 93 authors state "About one third of open TRiC was found without PFD" this is confusing, as based on Extended Data Fig 2 it is a third of all TRiC and ~70% of open TRiC that has no prefoldin bound. Same applies for the percentages in lines 95-96.

Referee #2

(Remarks to the Author)

In this manuscript, Xing et al presented a comprehensive in-situ structural characterization of the chaperonin TRiC/CCT, which is an ATP-driven protein-folding machine critical for folding many essential proteins such as actin and tubulin. Although the complex has been extensively characterized in vitro, its in-situ structure and assembly remain poorly understood until the studies presented in this manuscript. The authors developed a highly sophisticated cryo-electron tomography workflow to visualize the in-situ structures, conformational dynamics, and distribution of the chaperonin TRiC in human 293 cells at the resolution up to 1.0nm. Furthermore, the authors observed that the programmed cell death protein 5 (PDCD5) only interacts with the open TRiC conformation although the closed TRiC conformation is also abundantly observed in cells. Overall, this is a potentially high-impact work due to an innovative approach and impressive results. However, the structures and models derived from the in-situ studies should be further validated and experimentally tested. Otherwise, the studies presented here are rather descriptive.

Specific comments:

1. The authors used "template matching" to pick 1000 particles per tomogram, resulting in 360,000 raw particles from the untreated dataset and 352,000 raw particles from treated dataset. It's clear that most particles are "junks" because only 3,353 open and 4,054 closed TRiC particles were used to generate two class averages after 30 iterations of 3D classification. It's

- important to present detailed procedures and classification results to explain how less than 2% of the particles are selected.
2. In Figure 1, the author showed different densities occupied in the chamber. More detailed classification and analysis should be beneficial.
 3. In Line 190-191: "PDCD5 may dissociate during in vitro purification or structural determination procedures". In reference 37, in vitro cryo-EM structure shows that PDCD5 interacts with the open TRiC conformation.
 4. Classification results are critical for the spatial distribution of TRiC in different states. The authors should take multiple rounds of independent classification to figure out the most possible state for each particle.

Referee #3

(Remarks to the Author)

The authors use electron microscopy to assess the distribution of the TRiC complex in cells and propose that a density situated at the equator of the chaperonin chamber is in fact PDCD5. The authors assess open and closed forms of the chaperonin and its general distribution in the cytosol.

The work is original and of high significance to the molecular chaperone field but is of less significance to a broader audience.

Data is clearly presented but further work is needed to ensure the validity of approaches.

The manuscript requires further work in order to ensure that robust conclusions can be made.

Major Concern:

The biochemical assessment of PDCD5 interactions with CCT1 are insufficient to support that PDCD5 is interacting with intact TRiC oligomer. The approach of transfecting cells with PDCD5-FLAG then immunoprecipitating via the FLAG tag directly from cell lysate is not sufficiently stringent to select for PDCD5 that is bound to intact TRiC oligomers and thus cannot exclude that the interaction seen is not instead an interaction with a sub-assembled or monomeric CCT1. At the very least western blot analysis of several other TRiC subunits needs to be performed. The authors should indicate the recovery/efficiency of the immunoprecipitation experiments with regard to both the tagged PDCD5 and the TRiC subunits.

The work should also be supported by cross-linking MS analysis to confirm that the PDCD5 is indeed in close proximity to the three TRiC subunits suggested in the model.

In the discussion the statement regarding large proteins being substrates should be amended as showing an interaction with TRiC should not be sufficient to consider the protein to be a genuine folding substrate.

Minor concern:

On the whole the manuscript is well written and figures/data clearly presented. In figure 1c the labelling of the subunits in the rotated image should be checked.

Version 2:

Reviewer comments:

Referee #1

(Remarks to the Author)

The authors have thoroughly addressed my concerns. The rephrased introduction much better conveys the significance of the results, and the key findings are better articulated. Importantly, the fascinating PDCD5 finding is now well supported with additional data.

I support publication, and have only two minor comments for the authors to consider.

1. Line 283. "Our finding that about 1/3 of all particles resemble open TRiC without PFD (Fig. 1b), now strongly argues for a PFD-independent step indeed taking place in cells"

I'm not sure I understand the logic here. The PFD-TRiC complexes are presumably in dynamic equilibrium. Could Pfd not deliver the substrate to TRiC then leave again? In other words, the authors capture the end state not the pathway. Does observing a TRiC-substrate complex without PFD mean that PFD was not involved in delivering the substrate initially? Perhaps referring to the affinity/kinetics of PFD interaction with TRiC would clarify.

2. Thermal profiling experiments, line 209. "These experiments revealed substrate stabilization in PDCD5 KO cells."

This seems like an important result, but I had to dig through the methods to understand it. Could this result be stated more explicitly in the main text? Also, it seems counter-intuitive that knocking out a TRiC cofactor would result in higher levels of soluble actin. Are actin filaments pelleted here along with aggregates?

Referee #2

(Remarks to the Author)

The authors addressed the previous concerns very well. The revised manuscript is exceptional.

Point-to-point response to the reviewers' comments:

We thank all reviewers for their overall positive comments and constructive feedback, which helped us to significantly improve our manuscript.

One point that has been raised more than once regards the broader impact of our work. We therefore would like to briefly highlight the following biological insights and technical advances:

1. In situ structural biology based on cryo electron tomography presently triggers a lot of excitement. But, to date, it also has a major short-coming: in order to confidently identify a target of interest in the tomograms, high cellular abundance and fairly high molecular weight are required. For this reason, the vast majority of papers published are focused on ribosomes, nuclear pores or proteasomes. Further, experimental model systems that are more easily accessible to cryo electron tomography are often used, i.e. because of their reduced dimensions such as e.g. Mycoplasma cells or isolated microsomes (see e.g. Xue et al, Nature 2022 or Gemmer et al, Nature 2023). Here, we demonstrate that it is feasible to obtain in situ structures of functional states of a much less abundant and only moderate molecular weight target (TRiC) in human tissue culture cells. To briefly illustrate this: We initially picked 350,000 particles from ~350 tomograms of FIB-milled HEK293 cells. The vast majority of those particles were not TRiC. We established the workflow to detect and classify a subset of only ~7,000 particles corresponding to TRiC and determined its structure to sub-nanometer resolution. This corresponds to only 2% of particles even before splitting into the different functional states (open/closed/PFD-bound etc). While inarguably a technical advance, this is also very exciting because it opens up avenues to investigate various other biological processes by cryo-ET.
2. The manner by which we identify PDCD5 is conceptually attractive for future in situ structural biology. The structure solved from the inside of human cells contains an additional density that is not explained by any published in vitro structures. A systematic fitting analysis of AlphaFold models of previously published interactors of TRiC points to PDCD5 as the only candidate that can explain this density.

AlphaFold models covering all rotational states of TRiC subsequently highlighted that the interface with PDCD5 is formed only in one out of the eight rotational possibilities – exactly the state in which we identified the additional density in our structure. Importantly, we are able to confirm this model for PDCD5 binding biochemically. Moreover, the fact that PDCD5 is present in this position in virtually all open TRiC, although it has never been observed bound to TRiC in vitro, is also very surprising.

3. TRiC is critical for maintaining cellular homeostasis and is involved in cancer and neurodegenerative disease^{1,2}. It is an ATP-dependent chaperone that is much less well understood than the distantly related and very different prokaryotic GroEL. Previous in vitro analysis has shown that ATP hydrolysis is required for TRiC's transition from the open into the closed state. In those experiments, however, the nucleotide derivative ATP/AlFx was used to artificially drive TRiC closure³⁻⁶, and thus, the structural intermediates of the actual folding cycle of TRiC under physiological conditions have never been observed. Even the most basic question of whether the two back-to-back rings undergo a symmetric or asymmetric cycle of closure (i.e. one ring closed and one ring open as is the case for bacterial chaperonins) was not known. We now provide a compelling overview of the intermediates in the ATP-driven cycle of closure as it happens in the actual cell, and show only symmetrically open and closed states are populated in vivo. We feel that this point has so far been possibly under-appreciated by the reviewers.
4. In addition, we discover a number of unexpected cellular features of TRiC: the fact that it functions at capacity to fold substrates in a translation-dependent manner, and that closed TRiC, but not open TRiC, forms distinctive clusters in the native functional condition also in a translation-dependent manner. Such a differential structural arrangement of the two TRiC states in cells has not been reported to our knowledge and reveals a higher level of cellular organization of chaperonin function.

We think these findings have very broad implications and are of high interest to an interdisciplinary readership.

A second major point of concern was the degree of additional insight into the functional role of PDCD5 binding to TRiC. Here, we performed two sets of additional experiments:

Cell-based experiments:

1. With mutational analysis and reciprocal pulldown, we validate the interface of PDCD5's interaction with TRiC.
2. Native PAGE analysis of cell extracts confirms our finding that PDCD5 binds to TRiC.
3. Analysis of PDCD5 knockout cells supports a functional link to actin and tubulin folding.

Biochemical reconstitution experiments:

1. We reconstituted PDCD5-TRiC complex formation and confirm it binds to the open state of TRiC, but not to the ATP-AIFx induced closed state.
2. We generated PDCD5 mutants and validate the interface of PDCD5 with TRiC through biochemical analyses.
3. We show that PDCD5 does not affect the ATPase activity of TRiC.

Our experiments conclusively identify PDCD5 as a novel, previously unrecognized, *major* co-factor for the eukaryotic chaperonin, that is very abundantly and specifically bound to one intermediate of the TRiC duty cycle, and has been previously overlooked. We think that those insights are very striking and will certainly be the basis for further defining TRiC's precise molecular function, which will clearly require more studies beyond what is possible for this manuscript. PDCD5 can be knocked out without loss of cell viability, in fact its down-regulation promotes tumorigenesis, and it is thus not required for TRiC's essential function to fold proteins in the cell. On the other hand, genetic, protein-based and DEPMAP data show PDCD5 functions within the TRiC-PFD network and being embryonic lethal in mouse models ⁷, some functions are clearly important. At this point, we can only speculate that it may play a regulatory role linking TRiC function to sensing signalling or cell death pathways. It is indeed interesting that this structural in situ study unveils a new unexpected aspect of biology, but it may take years to unravel this pathway.

We will elucidate this in more detail in the following point-by-point response.

Referee #1 (Remarks to the Author):

Xing et al analyze previously acquired tomograms and present the first detailed structural analysis of the TRiC chaperonin in cells. This is an exciting area of study, and it is likely that much will be learnt by characterizing the function of chaperones in their native environment. The manuscript makes three main claims:

1. In cells, almost all TRiC is substrate-engaged.
2. Although the closed state of TRiC dominates in the presence of ATP in vitro, the

open state is abundant in cells.

3. Open (but not closed) TRiC is almost always bound by PDCD5.

More data are required to establish the veracity of these claims, as detailed below. Furthermore, although this is conceptually a very exciting piece of work, a fundamental limitation is that it does not substantially advance our understanding of the mechanism of TRiC. Although the observations regarding PDCD5 are potentially important, very little is done to follow up on these ideas. As this aspect is still very preliminary, it is not possible to tell whether additional data establishing the mechanism of PDCD5 would result in a study of general interest.

We thank the reviewer for the overall positive comments and detailed suggestions!

However, as pointed out in the general response above, we would like to respectfully disagree with the reviewer regarding the perceived advance in mechanistic understanding of TRiC function. In contrast to previous work on the TRiC folding cycle *in vitro*, we now assess this inside of live cells and show that:

- PFD-independent binding of substrates to open TRiC occurs. While this had only been speculated on before, we now quantitatively demonstrated it *in vivo*^{8,9}
- The rings in the TRiC complex are either symmetrically open or symmetrically closed. No asymmetric intermediates are detected: this key allosteric property of TRiC is in sharp contrast to the “bullet” intermediates of GroEL-ES.
- Differently shaped substrates can bind to different regions of the closed TRiC chamber. In fact, this is the first demonstration that the closed TRiC chamber encapsulates substrates in cells.
- About half of the TRiC complexes in cells are in the closed state, independently of whether or not active translation provides nascent substrates. This suggests that the *in vivo* ATP cycling of TRiC is not affected by substrate binding
- Under conditions of active translation, closed TRiC can form beads on a string-like clusters
- PDCD5 is a constitutive binding partner of the majority of open TRiC, this occurs in an unanticipated position inside of the chamber and is not mutually exclusive with PFD binding. Our new biochemical analyses suggest this interaction likely plays a regulatory role.

One additionally important conclusion of our study is that under conditions of ongoing translation, TRiC is almost fully engaged with substrate. This finding provides potential answers to two important questions in cellular proteostasis:

- (i) How much idle capacity do chaperones have in the cell? For TRiC, apparently not much.
- (ii) Why is TRiC not stress inducible, unlike other chaperones? Our findings that TRiC substrates are newly translated proteins explain this paradox, as translation is downregulated during stress.

Below are some suggestions to improve the clarity of the manuscript and more rigorously establish the main claims.

Major points

1. The authors show that ~55% of TRiC is symmetrically closed in cells and claim that this is different to the *in vitro* situation. Stated in line 24: “Although the closed TRiC state dominates the ATP-driven substrate folding cycle *in vitro*...”. However, no quantitative evidence is given for this claim. What proportion of TRiC is open/closed *in vitro*, exactly? What is expected at different ATP concentrations, based on *in vitro* experiments, and how does this compare to the ATP/ADP ratio in the cytosol?

As outlined in more detail below, we based our predictions on previous cryo-EM and SAXS studies showing the closed/open ratio of human TRiC with or without ATP *in vitro*, as well as the reported ATP concentrations in cells.

Previous single-particle work has shown that without ATP addition to the TRiC sample before plunge freezing, ~100% TRiC particles are in the open conformation^{3,5,9}. When ATP/AIFx is used to drive TRiC into the closed conformation, 40-80% of closed TRiC is observed³⁻⁶. Representative experiments have used 1 mM ATP and 5 mM MgCl₂ at 37 °C for 30 s before plunge freezing and found 27% of TRiC in the closed state³. Additionally, SAXS analyses of TRiC in the presence of different nucleotide states at 1 mM showed that TRiC adopts the same radius of gyration in the presence of ATP-AIFx and ATP, which led to the conclusion that TRiC most probably only transiently samples open states in the presence of hydrolysable ATP *in vivo*¹⁰. Since the reported ATP concentration of 3-5 mM in the cytoplasm of HEK 293 cells¹¹ and other

mammalian cells^{12,13} in a similar range, we had expected to observe a minority of closed TRiC. We however agree that this has not been obvious and revised the text for clarity. In any case, this discussion shows that we have observed something that had not been known in a physiological setting.

2. The “headline” claim of the manuscript is that PDCD5 is essentially constitutively bound to open TRiC. This claim requires additional support:

a. When describing the fitting of PDCD5 into the ‘extra’ density of open TRiC, the authors show the fit from the top (looking down into the TRiC chamber) but it would be helpful to show side views as well.

As suggested, we have now included the side views in Fig. 2 and Extended Data Fig. 5a.

Fig. 2

a, A PDCD5-CCT3-CCT1-CCT4 model predicted by AlphaFold-Multimer was fitted into the open TRiC map and explains the additional density observed proximate to CCT1, 3 and 4 in situ. *b*, The side view of open TRiC fitted with the PDCD5-CCT3-CCT1-CCT4 model in (a).

Extended Data Fig. 5a, side view:

a, Structure of PDCD5 (38-112 a.a.) in complex with the equatorial domain of CCT3-CCT1-CCT4 predicted by AlphaFold-Multimer. The predicted model was colored by per-residue confidence score (pLDDT). The predicted model was fitted into the open TRiC map shown in the side view.

b. The authors should fit other known TRiC binders (e.g. Phosducin-like proteins) into the density they assign to PDCD5 and show the differences in fit quality (for instance with a correlation coefficient).

As suggested, we have fitted other known TRiC interactors (Phosducin-like proteins and TRiC substrates) into the density and calculated their correlations as suggested. The results of this analysis are shown in Supplementary Fig. 6 and none of those interactors can explain the observed density.

Supplementary Fig. 6: The additional density from open TRiC fitted with TRiC binders.

A, Atomic model of PhLP1, PhLP2A, PhLP3, actin, tubulin and Gβ5 were fitted into the additional density segmented from the open TRiC map. PhLP1 (79-277 a.a. out of full length 301 a.a.), PhLP2A (91-232 a.a. out of full length 239 a.a.), PhLP3 (68-209 a.a. out of full length 226 a.a.) were used to achieve the best fitting for calculating the correlation. **B**, Calculation of correlation of the fitting shown in (a) in ChimeraX. The map simulated from atoms of the PDB model is at the resolution of 8 Å, which is similar to our map resolution. The fitting of PDCD5 was shown in Extended Data Fig. 5e.

Extended Data Fig. 5e:

In addition, we would like to emphasize that there are several lines of evidence that support the identity of PDCD5. (i) the density fitting discussed above; (ii) the AlphaFold models show an agreement with PDCD5 being the respective density in only the one of the eight rotational possibilities in which density is observed; (iii) our own biochemical data, which we have strengthened as suggested by the reviewer (see

below). (iv) mutational analysis that we have done as additional experiments during the revisions (see below).

c. For the PDCD5 pulldown, show that the entire TRiC complex is pulled down, not just CCT1. It would also be good to perform the reciprocal pulldown, by immunoprecipitating TRiC and looking for PDCD5.

We thank the reviewer for the suggestion. We added several additional experiments to support this conclusion. First, we can detect all eight TRiC subunits in the PDCD5 pulldown sample and now added this result to the manuscript (Fig. 2c).

Fig. 2c, PDCD5-Flag pulldown TRiC in HEK 293F cells.

We have furthermore performed the reciprocal pulldowns that the reviewer suggested. As expected, CCT3 pulled down PDCD5 (Extended Data Fig. 7g). In addition, we carried out native gel analyses of cell extracts that demonstrate that the cellular endogenous PDCD5 comigrates with the endogenous TRiC complex at the characteristic 1 MDa band (Supplementary Fig. 7b).

Extended Data Fig. 7g, CCT3 antibody (rabbit) pulldown endogenous PDCD5 in HEK 293F cells. Rabbit IgG (mock) as a control.

Supplementary Fig. 7b, Native gel analysis of cell lysis from HEK cell. WT, wild-type. KD, knockdown PDCD5. The arrow head shows PDCD5 comigrated with TRiC.

3. Alphafold is used to generate models of the TRiC-PDCD5 complex, and the predicted interface is shown in detail in Fig 2. To make specific claims about the interface, the prediction must be tested by mutagenesis of PDCD5.

We have analyzed two mutants as suggested both *in vivo* and *in vitro*. The first one targets the charged part of interface (RKK), and did not express well in HEK293 cells, possibly because of its critical importance. The second one targets the hydrophobic part of the interface (IL) and behaves as expected: binding of PDCD5 to TRiC is reduced (see below, Extended Data Fig. 6f,g). Since the expression (input for co-IP) of PDCD5-RKK was dramatically lower than PDCD5-WT, the co-IP experiment could not be used to reliably compare the PDCD5 WT and RKK interactions with TRiC. To

overcome this, we purified the PDCD5 WT and RKK proteins from *E. coli* and performed titration binding combined with native gel experiments by incubating purified TRiC complex with PDCD5 WT or RKK at different concentrations. In this *in vitro* setting mutation of RKK abrogated binding of PDCD5 to TRiC (see below, Extended Data Fig. 6h). Together the *in vivo* and *in vitro* experiments validate our structural model.

These data have been included into the manuscript:

Extended Data Fig. 6, f, Binding of PDCD5 to TRiC was measured by co-immunoprecipitation (co-IP) from HEK 293F cells transfected with PDCD5-Flag constructs (WT: wild-type; IL: I93G, L96G; RKK: R55A, K63A, K66A). g, Quantification of co-eluted CCT1 with PDCD5-Flag in (f).

Extended Data Fig. 6h, Native gels show western blots analysis of the binding of increasing amounts of recombinant WT PDCD5 (top) and RKK PDCD5 (bottom) to 300nM TRiC. Interaction was measured after 20 minutes of interaction at 25°C in

buffer (50mM Tris pH 7.4, 100mM KCl, 5mM MgCl₂, 10% glycerol, and 1mM TCEP).
left panel: anti-PDCD5, right panel: anti-CCT8.

4. Another key claim regarding PDCD5 is that it does not bind closed TRiC. This claim is based on a pulldown experiment using flag-tagged PDCD5. However, the experimental design has several issues which interfere with interpreting the result.

We have added an extensive set of experiments to support this conclusion, as described below at the end of point 4b, where we respond to comments regarding our previous experiments. We want to point out this claim is supported by several lines of evidence, beyond the pulldown experiment:

(i) the PDCD5 density was resolved in the cryo-ET map of the open TRiC, while PDCD5 was absent in the closed TRiC map (Extended Data Fig. 7a,b).

Extended Data Fig. 7a,b

(ii) The structural superimposition shows the closed conformation leads to a clash between the C-terminal region of PDCD5 and the CCT4 helix (N394 to V416), whereas it does accommodate PDCD5 binding in the open TRiC conformation (Extended Data Fig. 7c,d).

Extended Data Fig. 7c,d

(iii) The actin and tubulin binding sites within the closed TRiC cavity clash with the region occupied by PDCD5 (Extended Data Fig. 5e,f).

Extended Data Fig. 5e,f

a. The authors need to show that, in their experiment, the treatment of cell lysate with ATP/AlFx results in symmetrically closed TRiC.

We want to clarify that, other than the reviewer might assume, we did not treat the cell lysate with ATP/AlFx, but the beads with the purified PDCD5-TRiC complexes. As a matter of fact, these conditions are the same as in previous in vitro analysis, which had convincingly established that they are sufficient for TRiC to close³⁻⁶. To make this more transparent we added the information in the 'Method' section of our manuscript as follows:

“Without adding ATP in the TRiC sample before plunge freezing, ~100% TRiC particles are at open conformation based on the single-particle analysis^{3,5,9}. With extra ATP/AlFx in TRiC solution before plunge freezing, a portion of TRiC particles were closed, although different papers show different closed/open ratios with ATP/AlFx at different conditions. Closed/open ratio: ~1.7 in buffer (1 mM ATP, 5 mM MgCl₂, 5 mM Al(NO₃)₃, and 30 mM NaF) from reference³. ~5.1 in buffer (1 mM ATP, 1 mM Al₃(NO₃)₃, 6 mM NaF, 10 mM MgCl₂ 50 mM KCl) from reference⁶. ~0.6 in buffer with ATP-AlFx from reference⁵. ~2.2 in buffer (1 mM

ATP, 5 mM MgCl₂, and AlF_x (5 mM Al(NO₃)₃ and 30 mM NaF) from reference⁴. In our experimental settings (Extended Data Fig. 7i), we use the conditions from reference³ (1 mM ATP, 5 mM MgCl₂, 5 mM Al (NO₃)₃, and 30 mM NaF).”

b. As the authors state, “The absence of a PDCD5 density in the structure of closed TRiC, could in principle also be explained by PDCD5 becoming too flexible for structural detection.” If PDCD5 is encapsulated in closed TRiC, it would not be accessible for Flag pulldown. How can the authors discriminate between loss of PDCD5 and encapsulation within PDCD5?

Our rationale for this experiment is the following: We bind PDCD5-Flag/TRiC complex to Flag beads in the absence of nucleotide and wash the beads. Now all TRiC should be open. We subsequently incubate the beads with ATP/AlF_x which should drive TRiC closure. According to our model, the closed TRiC would then be released into the supernatant, as it does not bind PDCD5, while PDCD5 itself should remain bound to the beads. The respective experiment shows this is the case: Compared to the control, the TRiC abundance is increased in the supernatant after ATP/AlF_x treatment. PDCD5 however, remains on the beads.

Extended Data Fig. 7h,i,j :

Extended Data Fig. 7h, The schematic of induction of TRiC closure during co-immunoprecipitation (co-IP) in two conditions. **i**, As illustrated in (h), beads bound with TRiC-PDCD5-Flag (from co-IP) were incubated in buffer with ATP/AIFx or without ATP/AIFx (control) for 1 hour at 37°C. The supernatant (containing released TRiC and PDCD5) and beads (bound with TRiC-PDCD5-Flag) were detected by western blotting. **j**, The ratio of PDCD5 (left two columns) in supernatant compared to PDCD5 remaining bound to beads after ATP/AIFx incubation in (i). The ratio of TRiC (right two columns) in supernatant compared to TRiC bound to beads after ATP/AIFx incubation in (i). The data represent the mean \pm SD of four independent experiments. Statistical analysis was performed using two-tailed unpaired t-tests. Significantly different ($P < 0.05$). ns, not significant.

We have rephrased the respective text as follows to explain this better:

‘We immunoprecipitated the TRiC-PDCD5-Flag complex in buffer without ATP/AIFx (Methods) and subsequently induced TRiC closure by incubating the precipitated complex on beads with ATP/AIFx (1 h, 37°C)^{3–6}. Upon closure, we observe CCT1 enrichment (TRiC) in

the supernatant, indicating a release of TRiC from bead-bound PDCD5 (Extended Data Fig. 7h-j).

We have included the following additional data into the revised version to further support that PDCD5 does not bind the closed state of TRiC:

Since the symmetrically closed state of TRiC is stabilized by ATP-AIFx incubation, we carried out a series of experiments using purified TRiC complexes reconstituted with PDCD5. In these experiments the preformed binary TRiC-PDCD5 complexes were incubated with or without ATP-AIFx (or ATP which permits cycling between open and closed states) and then analyzed by native gel followed by immunoblot detection of PDCD5 comigration with TRiC (Extended Data Fig. 7k). It is very clear that TRiC closure dissociated PDCD5 from the complex. In fact, there is a small difference in migration between the closed and open TRiC and when overlaying the blots, it becomes apparent the residual PDCD5 in the ATP-AIFx condition is bound to the open TRiC in the sample. These experiments conclusively show that PDCD5 is bound to TRiC in the open state.

Extended Data Fig. 7k, Native gels of the interaction of 300nM TRiC to 3mM WT PDCD5 in buffer containing 1mM of different ATP analogs, which induce different TRiC conformational states, analyzed by immunoblotting with anti PDCD5 (bottom) or CCT8 (top) antibodies.

5. In the manuscript text, the authors are appropriately conservative about the possible proximity of TRiC to the ribosome exit. In this context, Fig 3e/h is misleading, as it suggests to the reader that this arrangement of TRiC and the ribosome represents a specific complex rather than just one of many possible orientations, with no way to distinguish between actual complexes and simple stochastic proximity. I suggest removing these panels.

We agree with the reviewer and have removed the panels from Fig. 3. We have updated Fig. 3 as below and moved the previous Fig. 3e-j to Supplementary Fig. 8a-f for transparency.

6. In lines 98-100 authors state that nearly all open TRiC binds substrate, and they show the binding in Fig 1 b and Extended Data Fig 3 a,c,e,h. How can the authors be confident that the density between the rings corresponds to substrate and not the N/C terminal tails of TRiC?

The N/C terminal amino acids exist in both untreated and HHT-treated cells. However, the densities between the rings of open TRiC almost disappeared in HHT-treated cells only (Figs. 1b and 4d and Extended Data Fig. 10), suggesting that these densities are likely not the N/C terminal region of TRiC. In addition, HHT inhibits translation and causes a reduction of translated proteins including TRiC substrates, which would be in line with the reduced density volume we observe inside the TRiC chamber in HHT-treated cells. Taken together we conclude that the densities are more likely substrates than N/C terminal tails of TRiC. This conclusion is supported by many single particle cryo-EM studies of TRiC: the N- and C-tail density is never observed in substrate-free TRiC, and only in the presence of substrate such density becomes visible^{3-6,14,15}.

7. In lines 163-165, the authors state that density for both PDCD5 and substrate can

be observed in the same maps. Can the authors show both entities in one structure/map? Could they comment on their assignment of that density as substrate rather than e.g. a cofactor or cochaperone?

We have now included densities in the same map with insets in Extended Data Fig. 3.

Extended Data Fig. 3: TRiC states fitted with atomic models in untreated cells.

a, The cryo-ET map of open TRiC without PFD bound in untreated cells. The densities within the TRiC cavity were Gaussian-filtered ($sDev = 4$). **b**, Different views of the map in (a) showing both PDCD5 and potential substrate densities. The predicted model of PDCD5-CCT3-CCT1-CCT4 was fitted into the map in (a). **c**, The open TRiC structure from EMDB (EMD-13754). The map was Gaussian-filtered ($sDev = 4$) for visualization. **d**, Different views of EMD-13754 showing potential substrate densities but no PDCD5 density. The AlphaFold-Multimer predicted model of PDCD5-CCT3-CCT1-CCT4 was fitted into the open TRiC map in (c). **e**,

Atomic model from PDB 7WU7 was fitted into the open TRiC structure associated with 1 PFD. The chamber densities were Gaussian-filtered (sDev = 4). f, PFD (PDB 7WU7) was fitted into the corresponding densities segmented from (e). g, Different views of the map in (e). The predicted model of PDCD5-CCT3-1-4 was fitted into the structure in (e). h, The open TRiC bound with 2 PFDs was fitted with PDB 7WU7. i, PFD densities segmented from (h) were fitted with PDB 7WU7. j, Different views of the map in (h). The predicted model of PDCD5-CCT3-CCT1-CCT4 was fitted into the map in (h).

As also discussed above (comment 2), the density volume inside the TRiC chamber was largely reduced in HHT-treated cells, where translation is inhibited (Extended Data Fig. 10), supporting that the densities correspond to substrates. However, we cannot ultimately exclude that this density may also correspond to other cofactors and have added a respective sentence to the main text:

“Notably, the density observed likely corresponds to the substrate because of its position within the TRiC chamber. However, we cannot ultimately exclude binding of cofactors, cochaperones or mixtures thereof in this position.”

8. Previous CryoET analyses showed that TRiC forms clusters in cells (<https://doi.org/10.1016/j.cell.2017.12.030>). This work should be cited and discussed.

We thank the reviewer for pointing this out. We agree that the figures of the publication show TRiC clusters, although the text does not specifically refer to them as clusters. We want to point out that those clusters were observed under a very specific condition and subcellular localization, i.e. upon transduction of neurons with toxic and aggregation-prone Poly-GA and were located at the aggregate. This is an important difference to our work, where the clusters were abundantly observed in the cytosol of non-stressed cells.

We now refer to this work in the discussion, as detailed below.

Notably, localized accumulation of TRiC complexes was observed at the periphery of Poly-GA aggregates in rat neurons expressing (GA)175-GFP¹⁶, while our findings strongly suggest that formation of TRiC clusters occurs as a more general phenomenon, likely related to the presence of substrate.

Minor points

1. In lines 46, 55 and 56, the authors refer to a single 'central chamber' as well as 'two opposite chambers'. This should be rephrased for clarity.

For clarity of nomenclature, we have removed the term 'central chamber' from the manuscript and rephrased lines 55 and 56 as below:

Recent in vitro studies reported binding of recombinant human PhLP2A, a co-chaperone, within the chamber of one ring of closed TRiC.

2. When describing the interface between PDCD5 and TRiC, the authors consistently swap positively charged and negatively charged residues (e.g. I. 142, 143 and 146; Figure 2 and Extended Figure 6).

We apologize for the mistake and thank this reviewer for pointing this out. We have corrected all related text and figures (Fig. 2d and Extended Data Fig. 6e).

Fig. 2d, bottom panel:

Negative Positive

Extended Data Fig. 6e:

3. In lines 180 and 182, the authors refer to Extended Data Fig. 6 instead of Extended Data Fig. 7.

We have corrected it.

4. In lines 203-205, the authors state that TRiC clusters have ‘a tendency to be organized in a one-dimensional fashion’, either linear or circular. Could this observation be a result of the very thin lamellae produced by FIB milling? Moreover, would a circle not be two dimensional?

Indeed, a circle is two-dimensional. We have removed the description ‘one-dimensional fashion’ and used ‘circular and linear organizations’ in the main text. The diameter of the TRiC particle is ~15 nm, but the lamella thickness is 100-200 nm. Thus, the organization (circular or linear) of TRiC was likely not limited by the lamella thickness.

5. The reference to published tomograms of *Dictyostelium discoideum* (lines 215-217 and Extended Data Fig. 8i) would lend more support to the authors’ claims if it included a more in-depth analysis. A single image is not the most convincing

We have decided that an in-depth analysis of Dictyostelium would not add to the main conclusions in a degree that would justify the significant amount of time and resources that would be necessary to do this. Therefore, we have removed the respective image.

6. In the paragraph starting in line 219, the authors could consider adding more detail regarding the approach to determine the positioning of TRiC relative to the exit tunnel.

We have added a separate section of the approach in Methods (see below) and cited the 'Methods' in the main text as follows: To explore this in situ, we analyzed the spatial interplay between ribosomes and TRiC (Methods)...

Spatial relation between ribosomes and TRiC in cells

The spatial distribution of TRiC near the ribosome exit tunnel was investigated. The coordinates of ribosome, 60S and 40S determined by subtomogram averaging were used to localize the particles in the tomograms¹⁷. The ribosome was rotated to a reference position (zero rotation) through an inverse rotation, which means it was rotated by $(-\psi, -\theta, -\varphi)_{\text{ribosome}}$. Subsequently, TRiC underwent rotation by its respective angles $(\varphi, \theta, \psi)_{\text{TRiC}}$, followed by another rotation of $(-\psi, -\theta, -\varphi)_{\text{ribosome}}$, thus aligning the ribosome-TRiC within a standard rotation frame (zero rotation of the ribosome), while maintaining their original angular relationship. The coordinates of the ribosome exit tunnel were subtracted from both the ribosome exit tunnel coordinates (setting it to zero) and TRiC coordinates. The new TRiC coordinates were rotated by $(-\psi, -\theta, -\varphi)_{\text{ribosome}}$ to illustrate their positioning relative to the zero rotation of the ribosome. For the spatial analysis of ribosome and TRiC, ribosome particles were more abundant than TRiC particles. As a result, the same TRiC can be the nearest neighbour of several ribosomes. Our analysis focused on the ribosomes that acted as the nearest neighbours of TRiC. The mean \pm SD in (Extended Data Fig. 9c,k) were as follows: untreated open TRiC in the ribosome exit tunnel side (ETS), $55.1\% \pm 0.8\%$; untreated closed

TRiC in the ETS, 55.3% ± 0.3%; untreated open TRiC in the non-ETS, 44.9% ± 0.8%; untreated closed TRiC in the non-ETS, 44.7% ± 0.3%; treated open TRiC in the ETS, 50.4% ± 0.4%; treated closed TRiC in the ETS, 49.7% ± 1.0%; treated open TRiC in the non-ETS, 49.6% ± 0.4%; treated closed TRiC in the non-ETS, 50.3% ± 1.0%.

7. In line 255, the authors state that ‘the densities corresponding to the substrate position were reduced’. The word ‘reduced’ may be confusing as it could refer to a lower number of particles exhibiting the density, or a decreased volume of the density itself.

Technically speaking, it could be both, we thus chose a term that leaves it undefined. We did not succeed in subclassifying the particles. We think it is not very likely that substrates become smaller. But they could become more flexible. Therefore, it appears appropriate to simply describe the observation transparently.

‘...less density was observed in the substrate position...’

8. The maps of the untreated and HHT-treated cells in Extended Data Figs. 3 and 1- would benefit from being shown side-by-side or overlaid (with the same threshold) for a clearer comparison.

As suggested, we now show the overlaid maps of untreated and HHT-treated TRiC at similar thresholds in Extended Data Fig. 10b,e,h. In addition, we have shown the untreated TRiC maps at similar thresholds as HHT-treated TRiC maps in Extended Data Fig. 3 without opacity.

Extended Data Fig. 10: Atomic models were fitted into TRiC structures in HHT-treated cells.

a, Cryo-ET map of open TRiC without PFD, fitted with PDB 7X3J in treated cells. **b**, Overlaid maps (at similar contour level) of untreated and treated TRiC without PFD. The opacity of the untreated map was set at 60%. The predicted model of PDCD5-CCT3-CCT1-CCT4 was fitted into the map in (a). **c**, PDB 7WU7 was fitted into the open TRiC structure bound with 1 PFD. **d**, PDB 7WU7 was fitted into the PFD density segmented from (c). **e**, Overlaid maps (at similar contour level) of untreated and treated TRiC with one PFD. The densities inside the TRiC chamber were Gaussian-filtered ($sDev = 4$) in (b) and (e) for visualization. The predicted model of PDCD5-CCT3-1-4 was fitted into the map in (c). **f**, The open TRiC structure bound with 2 PFDs was fitted with PDB 7WU7. **g**, PFD densities segmented from (f) were fitted with PDB 7WU7. **h**, Overlaid maps (at similar contour level) of untreated and treated TRiC with two PFDs. The predicted model of PDCD5-CCT3-CCT1-

CCT4 was fitted into the map in (f). i, Overlaid maps (at similar contour level) of untreated and treated TRiC at the closed conformation. The opacity of untreated maps was set at 60%.

9. Figure 4ef show that HHT treatment leads to the complete loss of substrate specifically in PFD-free open TRiC, leading to the appearance of a species that is virtually non-existent in untreated cells. This seems like an interesting result which would benefit from being discussed in the main text.

We agree that this is interesting. We have added the following text at the end of the first paragraph in the discussion section of the manuscript.

Upon HHT treatment, we observed an absence of substrate densities in PFD-free open TRiC (Fig. 4e,f). This finding may indicate that substrates in the chamber of open PFD-free TRiC reflect newly translated proteins, which would be absent upon translation inhibition.

10. Line 283. The comparison with condensates seems unnecessary – there is no reason to think that a complex/cluster of just a few molecules is related to a condensate.

We have removed the sentence about condensates.

11. In line 314, ‘a mutually exclusively manner’ should read ‘a mutually exclusive manner’.

We have corrected it to ‘a mutually exclusive manner’.

12. The electrostatic scale in Fig. 2 shows aspartate residues as positively charged.

We have corrected the Fig. 2d.

Fig. 2d, bottom panel:

13. In Fig. 2, the positioning of panel c (and insets) next to the maps in panels a and b gives the impression that the interface is clearly resolved in the experimental structure. This is not the case as the resolution of the reconstruction would not allow side chain building. The authors should make it clear in the visual itself that the interface is an AlphaFold prediction that has not been verified (e.g. by mutagenesis) but it consistent with the positioning of PDCD5 in the experimental map.

We have added the text 'AlphaFold-Multimer model' in the top of Fig. 2d and described in the figure legend, as detailed below.

Fig. 2d, The interface between PDCD5 and the stem loop of CCT1 predicted by AlphaFold-Multimer...

14. In Extended Data Fig. 6e, the electrostatic scale is likely swapped as the main text refers to lysine residues in PDCD5.

We have corrected the scale in Extended Data Fig. 6e and the related main text.

Extended Data Fig. 6e:

15. In Extended Data Fig. 8h, there is no need for a chart containing only one bar. The value can just be mentioned in text.

We have removed the figure and cited 'Methods' section describing how the value was calculated.

16. In Extended Fig 4e, the "top 10" genes associated with PDCD5 do not match the current DepMap analysis. This should be updated. Furthermore, can the authors comment on PDCD5 having a strong association with prefoldin genes, as well as tubulin folding cofactors TBCA and TBCB?

We thank the reviewer for the information. We now extend the discussion of the genetic interactions of PDCD5 (which DepMap essentially is) as well as the very similar genetic interactions in yeast for the PDCD5 homologue SDD2. Basically, both networks are very similar, and both firmly place PDCD5 at the center of the TRiC chaperone system. We have updated the Extended Data Fig. 4e to Supplementary Fig. 7c as follows.

Supplementary Fig. 7c, Top 15 genes associated with PDCD5 based on the analysis in DepMap (<https://depmap.org>). Genes involved in protein folding were colored in black. PFDN4 (prefoldin subunit 4), VBP1 (prefoldin subunit 3), PFDN1 (prefoldin subunit 1), PFDN5 (prefoldin subunit 5), TXNDC9 (thioredoxin domain-containing protein 9), TCP1 (CCT1), TBCA (Tubulin-specific chaperone A), TBCB (Tubulin-folding cofactor B). The y-axis represents the Pearson correlation.

The Pearson correlation co-efficients displayed refer to CRISPR knockout screens from the Achilles project assessing cancer cell lines for genetic co-dependency. Thus, these values indicate a potential functional relationship of the genes and/or the pathways they act in. Without further experiments, definitive causal interactions cannot be determined and we can only speculate on potential functional interactions. Prefoldin subunits are the top genetic interactors with PDCD5. PFD is proposed to deliver unfolded substrates into the TRiC chamber^{8,9}. Based on our observation that PFD, PDCD5 and potential substrates can co-exist in the open TRiC structure (Extended Data Fig. 3g), these genetic associations become conceivable.

TBCA and TBCB are cofactors involved in tubulin folding. Tubulin is one of the well-known and abundant TRiC substrates. The genetic interactions between TBCA/TBCB and PDCD5 may reflect this functional relationship.

17. In line 93 authors state “About one third of open TRiC was found without PFD” this is confusing, as based on Extended Data Fig 2 it is a third of all TRiC and ~70% of open TRiC that has no prefoldin bound. Same applies for the percentages in lines 95-96.

We thank the reviewer for pointing this out. We have corrected them to ‘one third of all TRiC’ and ‘12 % of all TRiC’ in previous lines 93-96.

Referee #2 (Remarks to the Author):

In this manuscript, Xing et al presented a comprehensive in-situ structural characterization of the chaperonin TRiC/CCT, which is an ATP-driven protein-folding machine critical for folding many essential proteins such as actin and tubulin. Although the complex has been extensively characterized in vitro, its in-situ structure and assembly remain poorly understood until the studies presented in this manuscript. The authors developed a highly sophisticated cryo-electron tomography workflow to visualize the in-situ structures, conformational dynamics, and distribution of the chaperonin TRiC in human 293 cells at the resolution up to 1.0nm. Furthermore, the authors observed that the programmed cell death protein 5 (PDCD5) only interacts with the open TRiC conformation although the closed TRiC conformation is also abundantly observed in cells. Overall, this is a potentially high-impact work due to an innovative approach and impressive results. However, the structures and models derived from the in-situ studies should be further validated and experimentally tested. Otherwise, the studies presented here are rather descriptive.

We thank this reviewer for the overall positive comments and the appreciation of our technical approach. We have provided more details about the image processing workflow and validated the classification by multiple independent runs, as detailed below. In addition, we have performed several in vitro and cell-based experiments to further support our conclusions that we draw from the structural analyses.

Specifically, we have:

- validated the interaction of PDCD5 with the full TRiC complex by IP (Fig. 2c, Extended Data Fig. 7g and Supplementary Fig. 7a) and native gels
- validated the interaction interface of PDCD5 with TRiC by mutating key residues
- performed additional experiments to show that PDCD5 only binds to closed TRiC
- reconstituted the complex formation TRiC-PDCD5 (WT and mutants) with purified components, and examined its properties, including assayed the effect on ATP hydrolysis.

Specific comments:

1. The authors used “template matching” to pick 1000 particles per tomogram, resulting in 360,000 raw particles from the untreated dataset and 352,000 raw particles from treated dataset. It’s clear that most particles are “junks” because only 3,353 open and 4,054 closed TRiC particles were used to generate two class averages after 30 iterations of 3D classification. It’s important to present detailed procedures and classification results to explain how less than 2% of the particles are selected.

We thank this reviewer for expressing this concern and have added more information in Extended Data Fig. 2 (untreated dataset), Supplementary Fig. 1 (HHT-treated dataset) and figure legends to make it more clear how TRiC particles were classified, as detailed below.

We want to point out that detection and classification of particles inside the complex cellular environment is more complicated than for in vitro single-particle samples. To include all potential TRiC particles in our dataset (we do not know a priori how many actual TRiC particles or how many conformations of TRiC to expect per tomogram), we used a strategy similar to an oversampling approach, meaning that we intentionally include many more particles than are likely present. We then perform several rounds of classifications (also see the following comment 4) to select the true TRiC particles. This is validated subtomogram averaging that ultimately results in subnanometer structures showing clear structural features of TRiC.

Extended Data Fig. 2: Data processing workflow of TRiC in untreated cells.

a, Diagrams of TRiC image-processing in the untreated dataset. Tomograms were reconstructed with IMOD at bin4. Initial TRiC candidates were generated through template matching using STOPGAP. Subtomogram extraction was carried out in Warp. 3D classifications were executed to remove false positive particles and identify TRiC particles in RELION 3.1 (Methods). TRiC particles were mapped back into the tomogram for assessing the workflow (Extended Data Fig. 1i). Further classification and refinement allowed us to determine different open and closed TRiC states. ~2% TRiC particles were highlighted in brown square and all percentages in the workflow were shown as the percent of 360,000 initial

particles for clarity. **b**, FSC curves of corresponding TRiC states and the resolution were displayed (FSC = 0.143).

Supplementary Fig. 1: Data processing of TRiC in HHT-treated cells.

a, The image-processing approach of TRiC in the HHT-treated dataset was the same as the untreated dataset (Extended Data Fig. 2a and Methods). ~2% TRiC particles were highlighted in brown square and all percentages in the workflow were shown as the percent of 352,000 initial particles for clarity. **b**, FSC curves of corresponding TRiC states and the resolution were displayed (FSC = 0.143).

2. In Figure 1, the author showed different densities occupied in the chamber. More detailed classification and analysis should be beneficial.

We agree with the reviewer that further classifications could be beneficial and we have extensively performed classifications with different parameters (class 4, $T = 1, 3, 5, 7, 9$, different mask shapes, different mask sizes, 30-50 iterations). Unfortunately, none of those allowed us to credibly subclassify the dataset further, which may be due to current methodological limitations or the heterogeneity of the densities inside the chamber in the cellular environment. As TRiC not only folds a single substrate or one stable state of the substrate in the cellular context, but assists in folding $\sim 10\%$ of the eukaryotic proteome, densities in the TRiC chamber are highly dynamic, potentially explaining why we cannot classify a meaningful structure. Nevertheless, we are including the results and parameters of what we have attempted (Supplementary Figs. 4 and 5), as they may be useful for the readers and community.

Supplementary Fig. 4: Classification of densities in the open TRiC chamber.

a-c, Multiple runs of focused classification with a mask covering the densities in the TRiC chamber. The curves show the change in the particle number of each class over 50 iterations. **a**, open TRiC without PFD bound. **b**, open TRiC with 1 PFD bound. **c**, open TRiC with 2 PFDs bound. Run 1 ($T = 3$, 3 classes, 50 iterations), Run 2 ($T = 1$, 3 classes, 50 iterations), Run 3 ($T = 5$, 3 classes, 50 iterations) and Run 4 ($T = 7$, 3 classes, 50 iterations) in (**a-c**).

Supplementary Fig. 5: Classification of densities in the closed TRiC chamber.

a-c, Multiple runs of focused classification with a mask covering the densities in the TRiC chamber. The curves show the change in the particle number of each class over 50 iterations. **a**, Closed TRiC-class 1. **b**, Closed TRiC-class 2. **c**, Closed TRiC-class 3. Run 1 ($T = 3$, 3 classes, 50 iterations), Run 2 ($T = 1$, 3 classes, 50 iterations), Run 3 ($T = 5$, 3 classes, 50 iterations) and Run 4 ($T = 7$, 3 classes, 50 iterations) in (**a-c**).

3. In Line 190-191: “PDCD5 may dissociate during in vitro purification or structural determination procedures”. In reference 37, in vitro cryo-EM structure shows that PDCD5 interacts with the open TRiC conformation.

In the specific section, we were referring to native PDCD5-TRiC complexes purified from cells. The cryo-EM structure (in reference 37) was determined from a sample reconstituted in vitro ¹⁸: ‘PDCD5-CCT complexes were prepared by mixing purified components in a 10:1 PDCD5/CCT ratio.’ It assigned PDCD5 into an entirely different position outside of the chamber based on a 25 Å resolution map (See Fig. 6B of this study¹⁸).

To make our description more precise, we have rephrased the sentence in the main text to ‘*This may indicate that PDCD5 associated with TRiC in cells dissociates during purification*’.

4. Classification results are critical for the spatial distribution of TRiC in different states. The authors should take multiple rounds of independent classification to figure out the most possible state for each particle.

We thank this reviewer for the suggestion. As detailed below, we have performed four more independent classifications to validate the open and closed TRiC in untreated and HHT-treated datasets (Supplementary Fig. 2). We have also carried out four more independent classifications for PFD-bound classes in untreated (Supplementary Fig. 3) and HHT-treated cells (Supplementary Fig. 9). For the classification of the densities in the TRiC chamber, please also refer to comment 2. Unfortunately, our additional efforts did not yield any further subclassification that was credible.

Supplementary Fig. 2: Validation of open and closed TRiC classification in two datasets.

a, Output of four runs to classify open and closed TRiC from 7,407 TRiC particles in the untreated dataset (classes = 4, $T = 0.5$, iterations = 30, C1 symmetry, without mask). **b**, In each run in (**a**), the total number of the classified open and closed TRiC was summarized. **c**, The curves show the change in the particle number of each class over 30 iterations in (**a**). **d**, Output of four runs to classify open and closed TRiC from 7,203 TRiC particles in the HHT-treated dataset (classes = 4, $T = 0.5$, iterations = 30, C1 symmetry, without mask). **e**, In each run in (**d**), the total number of the classified open and closed TRiC was summarized. **f**, The curves show the change in the particle number of each class over 30 iterations in (**d**).

Supplementary Fig. 3: Validation of classification of PFD-bound TRiC particles in untreated cells.

a, Output of four runs to classify TRiC without PFD, with 1 PFD and 2 PFDs from 3,353 open TRiC particles with a sphere mask focused on the top PFD region or bottom PFD region (classes = 3, $T = 3$, iterations = 50, $C1$ symmetry). The curves show the change in the particle number of each class over 50 iterations. PFD-bound class was labeled and empty classes were not labeled on the curve for virilization. The TRiC classes without PFD, with 1 PFD and 2 PFDs were sorted by finding shared and unique particles from focused classifications with the top and bottom masks (Methods). **b**, In each run in (a), the total number of TRiC without PFD and with PFD was summarized.

Supplementary Fig. 9: Validation of classification of PFD-bound TRiC particles in HHT-treated cells.

a, Output of four runs to classify TRiC without PFD, with 1 PFD and 2 PFDs from 3,785 open TRiC particles with a sphere mask focused on the top PFD or bottom PFD region (classes = 3, $T = 3$, iterations = 50, $C1$ symmetry). The curves show the change in the particle number of each class over 50 iterations. PFD-bound class was labeled and empty classes were not labeled on the curve. The TRiC classes without PFD, with 1 PFD and 2 PFDs were sorted by finding shared and unique particles from focused classifications with the top and bottom masks (Methods). **b**, In each run in (a), the total number of TRiC without PFD and with PFD was summarized.

Referee #3 (Remarks to the Author):

The authors use electron microscopy to assess the distribution of the TRiC complex in cells and propose that a density situated at the equator of the chaperonin chamber is in fact PDCD5. The authors assess open and closed forms of the chaperonin and its general distribution in the cytosol.

The work is original and of high significance to the molecular chaperone field but is of less significance to a broader audience.

We thank the reviewer for recognizing the high significance of our work. However, as also noted in our general response to all reviewers above, we respectfully disagree with the assessment that our study is of less interest to a broader audience.

Data is clearly presented but further work is needed to ensure the validity of approaches. The manuscript requires further work in order to ensure that robust conclusions can be made.

We have now added several experiments to further support our conclusions. These are detailed in our initial response to all reviewers, as well as in the specific points below.

Major Concern:

The biochemical assessment of PDCD5 interactions with CCT1 are insufficient to support that PDCD5 is interacting with intact TRiC oligomer. The approach of transfecting cells with PDCD5-FLAG then immunoprecipitating via the FLAG tag directly from cell lysate is not sufficiently stringent to select for PDCD5 that is bound to intact TriC oligomers and thus cannot exclude that the interaction seen is not instead an interaction with a sub-assembled or monomeric CCT1. At the very least western blot analysis of several other TriC subunits needs to be performed.

We thank the reviewer for this suggestion, which in our view was similar to reviewer #1's comment 2c. We performed the suggested additional experiments and now show that PDCD5 is interacting with all TRiC complex components through both co-IP. We have added this result to the manuscript (Fig. 2c) as follows.

Fig. 2c, PDCD5-Flag pulldown TRiC in HEK 293F cells.

We have furthermore performed the reciprocal pulldowns that the reviewer suggested. As expected, CCT3 pulled down PDCD5 (Extended Data Fig. 7g). In addition, we carried out native gel analyses of cell extracts that demonstrate that most of the cellular endogenous PDCD5 comigrates with the endogenous TRiC complex at the characteristic 1 MDa band (Supplementary Fig. 7b).

Extended Data Fig. 7g, CCT3 antibody (rabbit) pulldown endogenous PDCD5 in HEK 293F cells. Rabbit IgG (mock) as a control.

Supplementary Fig. 7b, Native gel analysis of cell lysis from HEK cell. WT, wild-type. KD, knockdown PDCD5. The arrow head shows PDCD5 comigrated with TRiC.

The authors should indicate the recovery/efficiency of the immunoprecipitation experiments with regard to both the tagged PDCD5 and the TRiC subunits.

We have added the immunoprecipitation quantification to Supplementary Fig. 7a and added more details in Methods.

Supplementary Fig. 7a, Efficiency of the co-IP in (Fig. 2c). The percentage was calculated by IP/Input and normalized by the amount of loaded sample (Input 1% and IP 5%) for western. The percentage's variability may depend on how much unassembled CCT subunits and assembled TRiC complexes in the cell lysis (Input). The data represent the mean \pm SD of four independent experiments (Methods).

The work should also be supported by cross-linking MS analysis to confirm that the PDCD5 is indeed in close proximity to the three TRiC subunits suggested in the model.

In principle, we agree with the reviewer that cross-linking mass spectrometry can be a valuable approach. However, in this specific context, we expect that such experiments would not be straight-forward to conduct nor interpret. There are many charged amino acids in PDCD5 and CCT, which may cause difficulty for XL-MS. Depending on the crosslinker's length, XL-MS may also detect CCTs that are not in close proximity.

Therefore, since we now provide several independent lines of evidence for the interaction of PDCD5 with TRiC, we find that XL-MS would not provide the extent of unambiguous additional support for the interaction to warrant attempting such an approach in the scope of this revision. To convince the reviewer, we would like to again briefly summarize those multiple lines of evidence supporting our conclusion that PDCD5 is close to CCT3, CCT1 and CCT4:

1) Open TRiC is rotationally asymmetric, in a very particular arrangement of 8 CCT proteins^{3,5}. Although the eight CCTs have a similar sequence, they have distinct conformations in open TRiC, meaning they open up to a different extent, giving the overall arrangement a very pronounced asymmetry (below, Fig. a). This is clearly resolved in our map, thus previously published atomic model can be fitted with very high confidence (Extended Data Fig. 1c and Supplementary Video 2). The additional density accounted for by PDCD5 is in a defined position close to CCT3, CCT1 and CCT4.

Fig. a:

Fig. a, Different views of open TRiC highlight the asymmetrical arrangement of CCT subunits.

Extended Data Fig. 1c:

Extended Data Fig. 1c, Open TRiC map (C1 symmetry) fitted with PDB 7X3J. Arrows point to the additional density: ring 1 (green); ring 2 (cyan).

2) Related to above point, we did not impose C2 symmetry in our subtomogram averaging approach. Thus, we obtained this result twice, for either of the rings.

3) The AlphaFold predicted PDCD5-CCT3-1-4 structure exhibited high-confidence interactions and fitted well with our cryo-ET map (Fig. 2d and Extended Data Fig. 5a). This accounts not only for the position of PDCD5 but also the exact shape of the structure.

Fig. 2a:

Extended Data Fig. 5a:

4) AlphaFold-Multimer prediction for the seven other possible rotational states did not yield any meaningful results, contrasting this one specific solution that is in agreement with the cryo-EM map and shows very high scores (Extended Data Fig. 5a,b).

Extended Data Fig. 5a,b:

5) This prediction suggested that the core region of PDCD5 was specifically associated with the stem loop of CCT1 and neither of the other subunits (Extended Data Fig. 5f).

Extended Data Fig. 5f:

6) We have done additional experiments during the revisions that mutate two of the respective patches in PDCD5 and tested them in cells and in vitro via reconstitution with purified components. The first mutant targets the charged part of interface (RKK). While this mutant did not express well in cells, we expressed it in vitro and showed it does not bind to purified TRiC. The second mutant targets the hydrophobic part of the interface (IL) and behaves as expected: binding of PDCD5 to TRiC is reduced in cells (Extended Data Fig. 6f,g). In addition, we also performed a suite of native gel analyses

using purified proteins that also show that PDCD5 binds TRiC and that mutation of the interaction interface abolishes this binding (Extended Data Fig. 6h).

Extended Data Fig. 6, f, Binding of PDCD5 to TRiC was measured by co-immunoprecipitation (co-IP) from HEK 293F cells transfected with PDCD5-Flag constructs (WT: wild-type; IL: I93G, L96G; RKK: R55A, K63A, K66A). g, Quantification of co-eluted CCT1 with PDCD5-Flag in (f).

Extended Data Fig. 6h, Native gels show western blots analysis of the binding of increasing amounts of recombinant WT PDCD5 (top) and RKK PDCD5 (bottom) to 300nM TRiC. Interaction was measured after 20 minutes of interaction at 25°C in buffer (50mM Tris pH 7.4, 100mM KCl, 5mM MgCl₂, 10% glycerol, and 1mM TCEP). left panel: anti-PDCD5, right panel: anti-CCT8.

We hope the reviewer agrees that the above evidence sufficiently supports our claim that PDCD5 is close to CCT3-1-4.

In the discussion the statement regarding large proteins being substrates should be amended as showing an interaction with TRiC should not be sufficient to consider the protein to be a genuine folding substrate.

The reviewer is correct, however, several large proteins (beyond the size cutoff to fit inside the chamber) have been shown to be obligate substrates of TRiC, including myosin, EFT2 and many F-box proteins^{19–21}. We have rephrased ‘many larger proteins are reported as TRiC substrates’ to ‘many larger proteins are reported as interactors of TRiC’ in the manuscript.

Minor concern:

On the whole the manuscript is well written and figures/data clearly presented. In figure 1c the labelling of the subunits in the rotated image should be checked.

We thank the reviewer for this comment. We have checked the labels of CCT subunits in Fig. 1c and added the following sentence to explain why we did not label the subunit in our map in the figure legend.

Fig. 1c, classification of closed TRiC according to substrate position. Individual CCT subunits cannot be assigned to our maps of closed TRiC at the given resolution.

References

1. Leroux, M. R. & Hartl, F. U. Protein folding: Versatility of the cytosolic chaperonin TRiC/CCT. *Current Biology* **10**, R260–R264 (2000).
2. Gestaut, D., Limatola, A., Joachimiak, L. & Frydman, J. The ATP-powered gymnastics of TRiC/CCT: an asymmetric protein folding machine with a symmetric origin story. *Curr Opin Struct Biol* **55**, 50–58 (2019).

3. Liu, C. *et al.* Pathway and mechanism of tubulin folding mediated by TRiC/CCT along its ATPase cycle revealed using cryo-EM. *Communications Biology* 2023 6:1 **6**, 1–14 (2023).
4. Kelly, J. J. *et al.* Snapshots of actin and tubulin folding inside the TRiC chaperonin. *Nature Structural & Molecular Biology* 2022 29:5 **29**, 420–429 (2022).
5. Gestaut, D. *et al.* Structural visualization of the tubulin folding pathway directed by human chaperonin TRiC/CCT. *Cell* **185**, 4770–4787.e20 (2022).
6. Park, J. *et al.* A structural vista of phospho-tyrosine-like PhLP2A-chaperonin TRiC cooperation during the ATP-driven folding cycle. *Nature Communications* 2024 15:1 **15**, 1–19 (2024).
7. Li, G. *et al.* Deletion of Pcd5 in mice led to the deficiency of placenta development and embryonic lethality. *Cell Death & Disease* 2017 8:5 **8**, e2811–e2811 (2017).
8. Vainberg, I. E. *et al.* Prefoldin, a Chaperone that Delivers Unfolded Proteins to Cytosolic Chaperonin. *Cell* **93**, 863–873 (1998).
9. Gestaut, D. *et al.* The Chaperonin TRiC/CCT Associates with Prefoldin through a Conserved Electrostatic Interface Essential for Cellular Proteostasis. *Cell* **177**, 751–765.e15 (2019).
10. Meyer, A. S. *et al.* Closing the Folding Chamber of the Eukaryotic Chaperonin Requires the Transition State of ATP Hydrolysis. *Cell* **113**, 369–381 (2003).
11. Yoshida, T., Kakizuka, A. & Imamura, H. BTeam, a Novel BRET-based Biosensor for the Accurate Quantification of ATP Concentration within Living Cells. *Sci Rep* **6**, (2016).
12. Gribble, F. M. *et al.* A Novel Method for Measurement of Submembrane ATP Concentration. *Journal of Biological Chemistry* **275**, 30046–30049 (2000).
13. Greiner, J. V. & Glonek, T. Intracellular atp concentration and implication for cellular evolution. *Biology (Basel)* **10**, (2021).
14. Wang, S. *et al.* Visualizing the chaperone-mediated folding trajectory of the G protein $\beta 5$ β -propeller. *Mol Cell* **83**, 3852–3868.e6 (2023).
15. Han, W. *et al.* Structural basis of plp2-mediated cytoskeletal protein folding by TRiC/CCT. *Sci Adv* **9**, (2023).
16. Guo, Q. *et al.* In Situ Structure of Neuronal C9orf72 Poly-GA Aggregates Reveals Proteasome Recruitment. *Cell* **172**, 696–705.e12 (2018).
17. Xing, H. *et al.* Translation dynamics in human cells visualized at high resolution reveal cancer drug action. *Science* **381**, 70–75 (2023).
18. Tracy, C. M. *et al.* Programmed cell death protein 5 interacts with the cytosolic chaperonin containing tailless complex polypeptide 1 (CCT) to regulate β -tubulin folding. *Journal of Biological Chemistry* **289**, 4490–4502 (2014).
19. Srikakulam, R. & Winkelmann, D. A. Myosin II Folding Is Mediated by a Molecular Chaperonin. *Journal of Biological Chemistry* **274**, 27265–27273 (1999).
20. Yam, A. Y. *et al.* Defining the TRiC/CCT interactome links chaperonin function to stabilization of newly made proteins with complex topologies. *Nature Structural & Molecular Biology* 2008 15:12 **15**, 1255–1262 (2008).
21. Freund, A. *et al.* Proteostatic Control of Telomerase Function through TRiC-Mediated Folding of TCAB1. *Cell* **159**, 1389–1403 (2014).

Point-to-point response to the reviewers' comments – round 2:

We sincerely appreciate the reviewer for positive feedback. Below are our responses to the final comments.

Referees' comments:

Referee #1 (Remarks to the Author):

The authors have thoroughly addressed my concerns. The rephrased introduction much better conveys the significance of the results, and the key findings are better articulated. Importantly, the fascinating PDCD5 finding is now well supported with additional data.

I support publication, and have only two minor comments for the authors to consider.

1. Line 283. “Our finding that about 1/3 of all particles resemble open TRiC without PFD (Fig. 1b), now strongly argues for a PFD-independent step indeed taking place in cells”

I'm not sure I understand the logic here. The PFD-TRiC complexes are presumably in dynamic equilibrium. Could Pfd not deliver the substrate to TRiC then leave again? In other words, the authors capture the end state not the pathway. Does observing a TRiC-substrate complex without PFD mean that PFD was not involved in delivering the substrate initially? Perhaps referring to the affinity/kinetics of PFD interaction with TRiC would clarify.

The reviewer is correct. Observing the PFD-free TRiC does not necessarily mean that PFD is not involved in delivering substrates to the TRiC chamber. We revised the discussion accordingly and removed the following sentence.

‘Our finding that about 1/3 of all particles resemble open TRiC without PFD (Fig. 1b), now strongly argues for a PFD-independent step indeed taking place in cells.’

2. Thermal profiling experiments, line 209. “These experiments revealed substrate stabilization in PDCD5 KO cells.”

This seems like an important result, but I had to dig through the methods to understand it. Could this result be stated more explicitly in the main text? Also, it seems counter-intuitive that knocking out a TRiC cofactor would result in higher levels of soluble actin. Are actin filaments pelleted here along with aggregates?

We included the following sentence into the main text to make this experiment more comprehensible:

“Thereby, cells were exposed to different temperatures to induce the aggregation of proteins to probe their stability.”

We have not assessed if filamentous or monomeric protein is pelleted. However, the WT and KO samples were prepared in parallel, making them comparable at the same conditions. In any case, at this stage interpreting this result mechanistically remains difficult, because we do not know how exactly PDCD5 affects protein folding. We agree with the reviewer, if one would assume that PDCD5 promotes folding, one may have expected reduced actin stability in KO cells. The apparent result nevertheless provides a functional association of the stability of two known substrates of TRiC with PDCD5. We have accounted for this in both, the main text:

“Although we cannot rule out indirect effects, these results are in line with a model in which PDCD5 is functionally associated with TRiC activity.”

.. and the discussion:

“Although we do not yet elucidate the precise molecular effect that PDCD5 binding has on TRiC function, the apparently exclusive interaction with the large majority of the cellular open TRiC suggests a role in modulating TRiC substrate recruitment and the folding cycle. This would be supported by the observation that actin and tubulin are stabilized in PDCD5 KO cells. However, we did not observe that PDCD5 affects TRiC ATPase activity in vitro

(Supplementary Fig. 10). More importantly, while TRiC function is essential, PDCD5 KO cells are viable. Thus, PDCD5 may have a regulatory function. Given its reported role in initiating apoptosis³⁵, it is also tempting to speculate that PDCD5 could potentially act as a sensor for TRiC activity, where the abundance of free PDCD5 would indicate the level of closed and thus occupied TRiC in a cell.”

Referee #2 (Remarks to the Author):

The authors addressed the previous concerns very well. The revised manuscript is exceptional.